# VisJudge-Bench: Aesthetics and Quality Assessment of Visualizations

**Yupeng Xie**[1], **Zhiyang Zhang**[1], **Yifan Wu**[1], **Sirong Lu**[1], **Jiayi Zhang**[1],
**Zhaoyang Yu**[2], **Jinlin Wang**[2], **Sirui Hong**[2], **Bang Liu**[3], **Chenglin Wu**[2], **Yuyu Luo**[1]*

[1]The Hong Kong University of Science and Technology (Guangzhou),
[2]DeepWisdom, [3]Université de Montréal & Mila

## Abstract

Visualization, a domain-specific yet widely used form of imagery, is an effective way to turn complex datasets into intuitive insights, and its value depends on whether data are faithfully represented, clearly communicated, and aesthetically designed. However, evaluating visualization quality is challenging: unlike natural images, it requires simultaneous judgment across data encoding accuracy, information expressiveness, and visual aesthetics. Although multimodal large language models (MLLMs) have shown promising performance in aesthetic assessment of natural images, no systematic benchmark exists for measuring their capabilities in evaluating visualizations. To address this, we propose VisJudge-Bench, the first comprehensive benchmark for evaluating MLLMs' performance in assessing visualization aesthetics and quality. It contains 3,090 expert-annotated samples from real-world scenarios, covering single visualizations, multiple visualizations, and dashboards across 32 chart types. Systematic testing on this benchmark reveals that even the most advanced MLLMs (such as GPT-5) still exhibit significant gaps compared to human experts in judgment, with a Mean Absolute Error (MAE) of 0.553 and a correlation with human ratings of only 0.428. To address this issue, we propose VisJudge, a model specifically designed for visualization aesthetics and quality assessment. Experimental results demonstrate that VisJudge significantly narrows the gap with human judgment, reducing the MAE to 0.421 (a 23.9% reduction) and increasing the consistency with human experts to 0.687 (a 60.5% improvement) compared to GPT-5. The benchmark is available at `https://github.com/HKUSTDial/VisJudgeBench`.

## 1 Introduction

Visualization serves as an effective approach for transforming complex datasets into intuitive insights (Shen et al., 2023; Qin et al., 2020; Ye et al., 2024; Li et al., 2024a; Zhu et al., 2025b; Li et al., 2025b;a). The value of a high-quality visualization depends on whether its data is faithfully presented (Fidelity), whether information is clearly communicated (Expressiveness), and whether the design is aesthetically well-presented (Aesthetics), as shown in Figure 1. These three dimensions are closely interconnected and indispensable, posing challenges for visualization quality assessment.

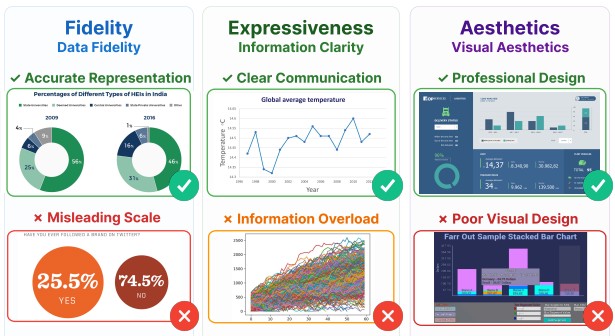

Figure 1: The "Fidelity, Expressiveness, and Aesthetics" evaluation framework.

---

*Corresponding author: Yuyu Luo (E-mail: yuyuluo@hkust-gz.edu.cn)

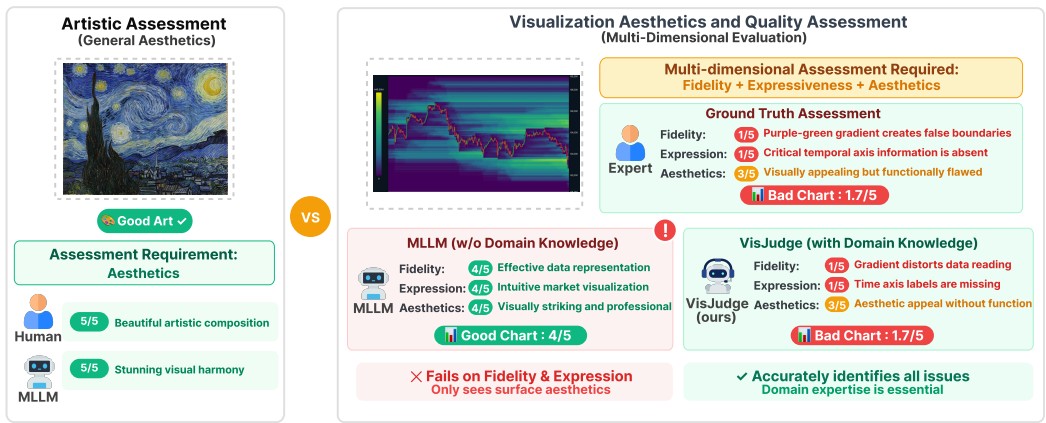

Figure 2: From natural images to visualization: the need for specialized visualization assessment. Green and red denote positive and negative assessments, respectively, highlighting the contrast between MLLMs' capabilities in general aesthetics versus visualization-specific evaluation.

Although Multimodal Large Language Models (MLLMs) have shown potential in aesthetic evaluation of natural images (Murray et al., 2012; Cao et al., 2025), applying them to visualization evaluation faces unique challenges. Unlike natural images, visualization evaluation requires simultaneous judgment of data encoding accuracy, information communication effectiveness, and visual design appropriateness, as shown in Figure 2. However, existing MLLM benchmarks are insufficient for such comprehensive evaluation, as detailed in Table 1. First, chart question answering benchmarks (e.g., ChartInsights (Wu et al., 2024)) evaluate models' ability to understand chart information, rather than their overall design quality. Second, natural image aesthetic evaluation benchmarks (e.g., ArtiMuse (Cao et al., 2025)) focus on assessing aesthetics, but ignore the core purpose of visualization to effectively communicate data. Finally, existing visualization evaluation benchmarks (e.g., VisEval (Chen et al., 2024b)) mainly evaluate natural language to visualization (NL2VIS) tasks (Luo et al., 2021c), with the focus on assessing whether generated visualizations accurately reflect natural language queries, rather than the aesthetics and quality of visualizations. This leads to a critical research gap: we lack a systematic framework to measure MLLMs' comprehensive capabilities in evaluating visualization aesthetics and quality.

To address this challenge, we construct VISJUDGE-BENCH, the first comprehensive benchmark based on the "Fidelity, Expressiveness, and Aesthetics" principles to assess MLLMs' capabilities in visualization aesthetics and quality evaluation. It contains 3,090 expert-scored samples from real-world scenarios, covering single visualizations, multiple visualizations, and dashboards across 32 chart types. Using this benchmark, we conduct extensive testing on 12 representative MLLMs, including GPT-5, finding that even the most advanced models show significant differences from human experts (MAE as high as 0.553, correlation only 0.428). This finding clearly demonstrates that general MLLMs cannot automatically acquire specialized evaluation capabilities in the visualization domain, making the development of specialized optimization models necessary.

Based on this, we propose VISJUDGE, a model specifically designed for visualization aesthetics and quality assessment, aimed at improving the consistency between general MLLMs and human expert evaluation standards. Experimental results prove the effectiveness of this approach: VISJUDGE significantly improves consistency with human experts, achieving a 23.9% reduction in MAE (to 0.421) and a 60.5% improvement in correlation (to 0.687) compared to GPT-5, performing best among all tested models.

In summary, our main contributions are: (1) We construct VISJUDGE-BENCH, a comprehensive benchmark based on "Fidelity, Expressiveness, and Aesthetics" principles to evaluate MLLMs' capabilities in visualization assessment. (2) We systematically evaluate representative MLLMs, revealing notable gaps with human expert standards. (3) We propose VISJUDGE, an optimized model that significantly outperforms existing models and better aligns with human expert judgment.

## 2 RELATED WORK

**Data Visualization Quality Assessment.** Assessing the quality of data visualizations is a core problem in visualization generation and recommendation tasks.

Table 1: Comparison of related benchmarks across key evaluation dimensions.

| Types | Benchmark | Input | Data Types | Evaluation Dimensions | | |
|---|---|---|---|---|---|---|
| | | | | Fidelity | Expressiveness | Aesthetics |
| Aesthetic Evaluation | AVA (Murray et al., 2012) | Images | General Images | × | × | ✓ |
| | ArtiMuse (Cao et al., 2025) | Images | General Images | × | × | ✓ |
| Chart Understanding | ChartQA (Masry et al., 2022) | Chart, Question | Single Vis | × | ✓ | × |
| | PlotQA (Methani et al., 2020) | Chart, Question | Single Vis | × | ✓ | × |
| | ChartInsights (Wu et al., 2024) | Chart, Question | Single Vis | × | ✓ | × |
| Visualization Evaluation | VisEval (Chen et al., 2024b) | Chart, NL, Data | Single Vis | × | ✓ | × |
| | VIS-Shepherd (Pan et al., 2025) | Chart, NL, Data | Single Vis | × | ✓ | × |
| | **VisJudge-Bench (Ours)** | **Chart** | **Single Vis, Multi Vis, Dashboard** | ✓ | ✓ | ✓ |

In *visualization recommendation* tasks, the goal is to enumerate and recommend the best (top-$k$) visualizations for a given dataset. To achieve this, existing methods fall into two main categories. The first is *rule-based approaches*, such as Voyager (Wongsuphasawat et al., 2017), Draco (Moritz et al., 2019), and CoInsight (Li et al., 2024b), which use heuristic scoring based on established design principles. However, their rules are often hard-coded and lack flexibility. The second category is *learning-based methods*, like VizML (Hu et al., 2019), DeepEye (Luo et al., 2018; 2022), and HAIChart (Xie et al., 2024). These methods train models on large annotated datasets to predict user preferences but are limited by simplistic evaluation dimensions and expensive annotated data.

In *NL2VIS* tasks, the goal is to generate corresponding visualizations based on user-provided natural language queries (Luo et al., 2021c). Representative works include ncNet (Luo et al., 2021c), Deep-VIS (Shuai et al., 2025), ChartGPT (Tian et al., 2023), and LLM4Vis (Wang et al., 2023). To assess how accurately these methods translate natural language into visualizations, several benchmarks have been proposed, including nvBench (Luo et al., 2021b;a), nvBench 2.0 (Luo et al., 2025), and MatPlotAgent (Yang et al., 2024). However, these methods and their related evaluations primarily focus on the model's ability to "write" code rather than "judge" the quality of visualizations.

**MLLM as a Judge.** Recently, MLLMs have shown significant potential in emulating human expert judgment, a paradigm known as "*MLLM-as-a-Judge*" (Zheng et al., 2023; Chen et al., 2024a; Bian et al., 2025). As summarized in Table 1, these works fall into three categories: (1) *general visual aesthetics assessment* (e.g., AVA (Murray et al., 2012), ArtiMuse (Cao et al., 2025)), which evaluates the artistic quality of photographs but overlooks information communication efficiency in data visualization; (2) *chart understanding tasks* (e.g., ChartQA (Masry et al., 2022), ChartInsights (Wu et al., 2024)), which focus on understanding and interpreting chart content but neglect overall design quality; and (3) *visualization evaluation* (e.g., VisEval (Chen et al., 2024b), VIS-Shepherd (Pan et al., 2025)), which evaluates whether generated visualizations accurately reflect natural language queries, but lacks a comprehensive assessment of intrinsic design quality. To address this gap, we introduce VISJUDGE-BENCH, the first comprehensive benchmark designed to systematically evaluate the capabilities of MLLMs as "*visualization quality judges*".

## 3  VISJUDGE-BENCH: DESIGN AND CONSTRUCTION

To systematically evaluate the capability boundaries of MLLMs in visualization evaluation, we design VISJUDGE-BENCH. As shown in Figure 3, its construction follows a three-stage methodology: (1) data collection and processing; (2) adaptive question generation; and (3) expert annotation and quality control. We detail the specific implementation of each stage below.

### 3.1  BENCHMARK CONSTRUCTION PIPELINE

#### 3.1.1  DATA COLLECTION AND PREPROCESSING

**Corpus Construction.** To evaluate the performance of MLLMs across different visualization types, we construct a corpus covering three main categories: single visualizations, multiple visualizations, and dashboards. To ensure the authenticity and diversity of our corpus, we collect visualization samples from search engines using web crawling methods with diverse query keywords (see Appendix A.1 for detailed crawling architecture and keyword generation strategy).

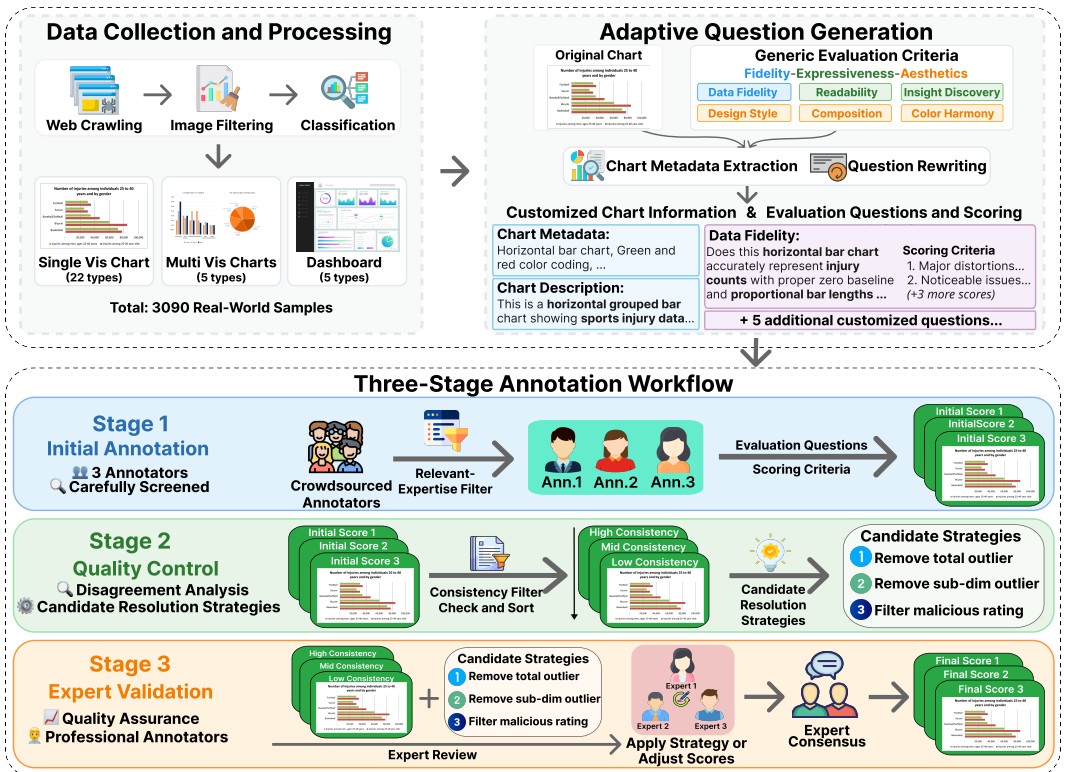

Figure 3: VISJUDGE-BENCH construction framework.

Table 2: VISJUDGE-BENCH statistical information (Dash. = Dashboard).

| Vis Type | Count | #-Subtype | Subtype Details (Count) | | | | | |
|---|---|---|---|---|---|---|---|---|
| Single Vis | 1,041 | 22 | Bar Chart | 176 | Pie Chart | 129 | Line Chart | 100 |
| | | | Area Chart | 75 | Heatmap | 55 | Scatter Plot | 49 |
| | | | Histogram | 48 | Donut Chart | 47 | Funnel Chart | 45 |
| | | | Treemap | 62 | Sankey Diagram | 61 | Bubble Chart | 29 |
| | | | *... 10 more subcategories* | | | | | |
| Multi Vis | 1,024 | 5 | Comparison Views | 670 | Small Multiples | 195 | Coordinated Views | 97 |
| | | | Other Multi View | 59 | Overview Detail | 3 | | |
| Dashboard | 1,025 | 5 | Analytical Dash. | 743 | Operational Dash. | 122 | Interactive Dash. | 91 |
| | | | Strategic Dash. | 62 | Other Dash. | 7 | | |

**Data Preprocessing Pipeline.** We design a three-stage data filtering process to curate the benchmark from over 300,000 initial images. (1) Initial Filtering: We employ automated scripts and perceptual hash algorithms to eliminate non-visualization content and duplicates, yielding 80,210 candidate images (detailed algorithms in Appendix A.1). (2) Automated Classification: We leverage GPT-4o for visualization type classification and quality filtering, resulting in 13,220 valid visualization samples after human verification (classification prompts and criteria in Appendix A.1). (3) Stratified Sampling: We apply stratified random sampling to select the final 3,090 samples, ensuring balanced distribution across categories. As shown in Table 2, the final corpus contains 1,041 single visualizations, 1,024 multiple visualizations, and 1,025 dashboards, covering 32 distinct subtypes. Detailed dataset statistics and distribution analyses are provided in Appendix A.2.

### 3.1.2 THE "FIDELITY, EXPRESSIVENESS, AND AESTHETICS" EVALUATION FRAMEWORK

To enable fine-grained visualization assessment, this stage first establishes a multi-dimensional evaluation framework, then implements an adaptive question generation process based on this framework (as illustrated in the upper-right panel of Figure 3).

**The "Fidelity, Expressiveness, and Aesthetics" Framework Design.** To systematically evaluate visualization quality, we construct a multi-dimensional evaluation framework. This framework draws inspiration from classical translation theory principles of "Fidelity, Expressiveness, and Aesthetics" (illustrated with positive-negative examples in Figure 1), combined with established theories in graphical perception (Cleveland & McGill, 1984), information visualization design (Munzner, 2025), and aesthetic evaluation (Cao et al., 2025). We operationalize this core concept into six measurable evaluation dimensions (as shown in Figure 3):

- **Fidelity** focuses on **Data Fidelity**. This dimension draws from Tufte's design principles (Tufte, 1983) for avoiding "graphical lies" and recent research on visualization misleadingness issues (Nguyen et al., 2013; Szafir, 2018; Lan & Liu, 2024; McNutt et al., 2020). Since source data for web-collected visualizations is typically unavailable, this dimension evaluates data presentation accuracy at the visual level. It assesses whether visual encodings accurately reflect the displayed values, examining visually detectable distortions such as improper axis settings, scale distortions, truncated baselines, disproportional encodings, or other misleading design patterns.

- **Expressiveness** focuses on the effectiveness of information communication. This dimension evaluates how effectively visualizations convey information to users. It includes two progressive sub-dimensions: First, (1) **Semantic Readability** evaluates the clarity of basic information encoding, assessing whether users can unambiguously decode visual elements in charts (Pan et al., 2025; Tang et al., 2026; Wang et al., 2026). Building on chart readability, (2) **Insight Discovery** further evaluates the analytical value in revealing deep data patterns, trends, or outliers, helping users transition from "reading information" to "gaining insights" (Chen et al., 2024b; 2025).

- **Aesthetics** focuses on **Aesthetic Quality** of visual design, integrating visualization perception theory (Ware, 2019) with design practice. This dimension consists of three sub-dimensions that collectively influence the overall visual experience: (1) **Design Style** evaluates the innovation and uniqueness of design, measuring the degree of novel visual elements and distinctive style (Dibia, 2023; Brath & Banissi, 2016); (2) **Visual Composition** focuses on the rationality of spatial layout, evaluating the balance and order of element positioning, size proportions, and spacing arrangements (Wu et al., 2023); and (3) **Color Harmony** evaluates the coordination and functionality of color combinations, ensuring color palette choices balance aesthetics with effective information communication (Harrower & Brewer, 2003; Gramazio et al., 2017).

In addition, this evaluation framework offers flexibility, with evaluation questions and scoring criteria adaptively tailored to each visualization's type (such as single visualizations (Chen et al., 2024b), multiple visualizations (Chen et al., 2020), and dashboards (Bach et al., 2023)) and its specific visual characteristics. Complete evaluation rules and customization details are provided in Appendix C.

**Adaptive Question Generation Mechanism.** Based on the evaluation framework, we have devised an adaptive question generation process (detailed workflow shown in Figure 3). This process begins by leveraging GPT-4o to extract metadata from the chart, such as its type and visual elements. Subsequently, it rewrites questions by populating predefined templates based on this metadata, generating highly customized questions and their corresponding five-point scoring criteria for the six evaluation sub-dimensions. For instance, under the *data fidelity* dimension, the generated question specifically targets the chart's data presentation accuracy, asking whether the "horizontal bar chart accurately represents injury counts with a proper zero baseline and proportional bar lengths." The scoring criteria focus on visually detectable issues: a score of 1 identifies "major distortions, such as truncated axes or misleading scales that exaggerate differences," whereas a score of 5 confirms a "highly faithful representation where bar lengths are strictly proportional to the displayed values." This approach ensures that the evaluation questions are closely aligned with the specific visualization content. For more detailed examples, please refer to Appendix C.1.

### 3.1.3 Expert Annotation and Quality Control

To build reliable human ground truth, VISJUDGE-BENCH adopts a rigorous three-stage annotation and quality control workflow (bottom panel of Figure 3) informed by benchmark construction approaches (Rein et al., 2024; Liu et al., 2025; Zhu et al., 2024; Shen et al., 2026). This systematic process ensures high-fidelity and consistent scoring through careful review and expert judgment.

**Stage 1: Initial Annotation.** We recruited 603 highly qualified crowdsourcing workers through the CloudResearch platform (Hartman et al., 2023). To ensure annotation quality, we set strict

screening criteria (see Appendix A.3.1 for details) and designed a dedicated annotation interface (Appendix A.3.3). Crucially, we embedded validation checks to identify and filter out inattentive responses (examples in Appendix A.3.4). Each of the 3,090 samples was scored by three independent annotators across six evaluation dimensions (task design details in Appendix A.3.2), generating an initial scoring matrix.

**Stage 2: Quality Control.** To address scoring disagreements among annotators, we designed a systematic conflict identification and resolution mechanism based on established crowdsourcing quality control and statistical evaluation theory (Gadiraju et al., 2015; Rousseeuw & Leroy, 2003; Brennan, 1992). The system first identifies high-disagreement samples by analyzing score variance, then algorithmically generates candidate resolution strategies including outlier removal, malicious scoring detection, and sub-dimensional bias correction. These algorithm-generated suggestions are processed and ranked before being submitted to the expert team for final review (complete algorithmic details and parameters in Appendix A.3.4).

**Stage 3: Expert Validation.** Three experts with visualization analysis experience independently reviewed all samples using a dedicated interface (Appendix A.3.5), selecting, modifying, or rejecting algorithm-generated candidate solutions. For complex cases, the team reached consensus through discussion. Through this rigorous process, we built a high-quality human scoring benchmark for all 3,090 samples as the gold standard for model evaluation. Detailed annotation quality analysis is provided in Appendix A.4.

## 4 VisJudge: A Specialized Model for Visualization Evaluation

The primary goal of VisJudge-Bench is to systematically evaluate MLLMs' capabilities in visualization aesthetics and quality assessment. Building on this benchmark, we explore a key question: can domain-specific fine-tuning improve models' visualization evaluation capabilities? To answer this question, we develop VisJudge by fine-tuning multiple open-source multimodal architectures at different parameter scales. In Section 5, we conduct unified evaluation of these fine-tuned models alongside closed-source and open-source baselines on the same test set.

**Training Setup.** We use VisJudge-Bench's human-annotated data with a 70%/10%/20% train/-validation/test split (2,163/279/648 samples) via stratified sampling to maintain consistent visualization type distribution across all splits. Training data is kept separate from baseline evaluation to prevent contamination.

**Model Training.** We fine-tuned four representative open-source multimodal models as base models: Qwen2.5-VL (3B/7B-Instruct) (Bai et al., 2025), InternVL3-8B (Zhu et al., 2025a), and Llava-v1.6-mistral-7B (Liu et al., 2024). These models span different architectures and parameter scales (3B to 8B), enabling comprehensive evaluation of our training approach's cross-architecture generalization (Lin et al., 2025). All models are trained to generate quality scores (1.0–5.0) and rationales aligned with human expert judgments. We employ reinforcement learning with the GRPO algorithm (Shao et al., 2024; Liang et al., 2026; Wu et al., 2026), using a composite reward function combining accuracy reward (minimizing prediction error) and format reward (ensuring structured outputs) (Shi et al., 2025; Wu et al., 2025). Formal reward definitions are detailed in Appendix D.2. For parameter-efficient fine-tuning, we adopted Low-Rank Adaptation (LoRA) (Hu et al., 2022). All models were trained using consistent hyperparameter configurations across architectures. Training used 5 epochs with a learning rate of 1e-5. Detailed configurations are in Appendix D.3.

## 5 Experiments

### 5.1 Experimental Settings

To evaluate existing MLLMs in visualization quality assessment and validate our VisJudge, we conduct comprehensive experiments on VisJudge-Bench.

**Evaluation Setup.** We evaluate 12 representative MLLMs: GPT-5, GPT-4o, Claude-4-Sonnet, Claude-3.5-Sonnet, Gemini-2.0-Flash, Gemini-2.5-Pro, Qwen2.5-VL (3B/7B/32B/72B-Instruct) (Bai et al., 2025), InternVL3-8B (Zhu et al., 2025a), Llava-v1.6-mistral-7B (Liu et al., 2024), and their corresponding fine-tuned variants, which we collectively refer to as VisJudge, on a balanced test set of 648 samples (see Appendix A.2.7 for distribution details). Each model provides 1-to-5 scores with justifications based on our evaluation framework. Following human

Table 3: Overall performance of MLLMs and the VISJUDGE on VISJUDGE-BENCH across different evaluation metrics and dimensions.

| Metric | Type | Model | Overall | Fidelity | Expressiveness | | Aesthetics | | |
| | | | | | Readability | Insight | Design Style | Composition | Color |
|---|---|---|---|---|---|---|---|---|---|
| **MAE (↓)** | Closed-source | Claude-3.5-Sonnet | 0.824 | 0.978 | 0.904 | 1.154 | 0.783 | 0.940 | 0.862 |
| | | Claude-4-Sonnet | 0.622 | 0.841 | 0.757 | 0.832 | 0.679 | 0.734 | 0.785 |
| | | Gemini-2.0-Flash | 0.682 | 0.829 | 0.912 | 0.820 | 0.638 | 0.729 | 0.798 |
| | | Gemini-2.5-Pro | 0.662 | 1.243 | 0.945 | 0.900 | 0.840 | 0.918 | 0.980 |
| | | GPT-4o | 0.610 | 0.988 | 0.806 | 0.744 | 0.609 | 0.695 | 0.657 |
| | | GPT-5 | 0.553 | 0.862 | 0.781 | 0.778 | 0.649 | 0.699 | 0.682 |
| | Open-source | Qwen2.5-VL-3B-Instruct | 0.821 | 1.085 | 1.258 | 1.088 | 0.723 | 0.727 | 0.808 |
| | | Qwen2.5-VL-7B-Instruct | 0.847 | 1.171 | 1.296 | 0.858 | 0.756 | 0.812 | 0.772 |
| | | Qwen2.5-VL-32B-Instruct | 0.703 | 0.909 | 0.987 | 0.801 | 0.678 | 0.761 | 0.719 |
| | | Qwen2.5-VL-72B-Instruct | 0.702 | 0.999 | 0.926 | 0.811 | 0.663 | 0.735 | 0.717 |
| | | InternVL3-8B | 0.793 | 1.234 | 1.193 | 0.870 | 0.679 | 0.759 | 0.753 |
| | | Llava-v1.6-mistral-7B | 0.724 | 0.929 | 1.124 | 0.939 | 0.801 | 0.814 | 0.818 |
| | VisJudge (Ours) | InternVL3-8B | 0.541 | 0.797 | 0.769 | 0.691 | 0.608 | 0.645 | 0.616 |
| | | Llava-v1.6-mistral-7B | 0.496 | 0.767 | 0.695 | 0.775 | 0.643 | 0.574 | **0.598** |
| | | Qwen2.5-VL-3B-Instruct | 0.491 | 0.705 | 0.721 | 0.696 | 0.625 | 0.571 | 0.616 |
| | | Qwen2.5-VL-7B-Instruct | **0.421** | **0.661** | **0.648** | **0.677** | **0.580** | **0.545** | 0.604 |
| **MSE (↓)** | Closed-source | Claude-3.5-Sonnet | 1.009 | 1.577 | 1.306 | 1.989 | 0.994 | 1.467 | 1.198 |
| | | Claude-4-Sonnet | 0.603 | 1.182 | 0.973 | 1.147 | 0.774 | 0.934 | 1.037 |
| | | Gemini-2.0-Flash | 0.720 | 1.182 | 1.326 | 1.120 | 0.673 | 0.925 | 1.057 |
| | | Gemini-2.5-Pro | 0.677 | 2.291 | 1.478 | 1.371 | 1.112 | 1.325 | 1.460 |
| | | GPT-4o | 0.577 | 1.562 | 1.063 | 0.921 | 0.627 | 0.823 | 0.729 |
| | | GPT-5 | 0.486 | 1.219 | 0.989 | 0.970 | 0.720 | 0.862 | 0.810 |
| | Open-source | Qwen2.5-VL-3B-Instruct | 1.028 | 1.872 | 2.400 | 1.852 | 0.868 | 0.903 | 1.046 |
| | | Qwen2.5-VL-7B-Instruct | 1.045 | 2.051 | 2.413 | 1.177 | 0.938 | 1.093 | 0.996 |
| | | Qwen2.5-VL-32B-Instruct | 0.756 | 1.361 | 1.528 | 1.067 | 0.728 | 0.988 | 0.880 |
| | | Qwen2.5-VL-72B-Instruct | 0.762 | 1.626 | 1.373 | 1.092 | 0.713 | 0.932 | 0.870 |
| | | InternVL3-8B | 0.934 | 2.240 | 2.088 | 1.219 | 0.747 | 0.981 | 0.968 |
| | | Llava-v1.6-mistral-7B | 0.817 | 1.513 | 1.960 | 1.403 | 1.007 | 1.071 | 1.076 |
| | VisJudge (Ours) | InternVL3-8B | 0.437 | 1.068 | 0.951 | 0.822 | 0.638 | 0.736 | 0.655 |
| | | Llava-v1.6-mistral-7B | 0.383 | 0.919 | 0.752 | 0.952 | 0.642 | 0.541 | 0.619 |
| | | Qwen2.5-VL-3B-Instruct | 0.377 | 0.855 | 0.850 | 0.822 | 0.620 | 0.539 | 0.603 |
| | | Qwen2.5-VL-7B-Instruct | **0.286** | **0.747** | **0.690** | **0.756** | **0.545** | **0.498** | **0.578** |
| **Corr. (↑)** | Closed-source | Claude-3.5-Sonnet | 0.395 | 0.325 | 0.492 | 0.365 | 0.455 | 0.137 | 0.259 |
| | | Claude-4-Sonnet | 0.465 | 0.393 | 0.550 | 0.452 | 0.421 | 0.163 | 0.228 |
| | | Gemini-2.0-Flash | 0.395 | 0.372 | 0.459 | 0.417 | 0.459 | 0.157 | 0.209 |
| | | Gemini-2.5-Pro | 0.265 | 0.178 | 0.379 | 0.353 | 0.445 | 0.193 | 0.208 |
| | | GPT-4o | 0.482 | 0.381 | 0.539 | 0.442 | 0.471 | 0.277 | 0.363 |
| | | GPT-5 | 0.428 | 0.255 | 0.439 | 0.382 | 0.463 | 0.276 | 0.295 |
| | Open-source | Qwen2.5-VL-3B-Instruct | 0.272 | 0.199 | 0.222 | 0.275 | 0.338 | 0.130 | 0.155 |
| | | Qwen2.5-VL-7B-Instruct | 0.341 | 0.341 | 0.352 | 0.281 | 0.357 | 0.149 | 0.155 |
| | | Qwen2.5-VL-32B-Instruct | 0.435 | 0.348 | 0.468 | 0.408 | 0.449 | 0.200 | 0.268 |
| | | Qwen2.5-VL-72B-Instruct | 0.440 | 0.331 | 0.479 | 0.416 | 0.435 | 0.165 | 0.251 |
| | | InternVL3-8B | 0.409 | 0.323 | 0.407 | 0.344 | 0.419 | 0.216 | 0.170 |
| | | Llava-v1.6-mistral-7B | 0.180 | 0.201 | 0.137 | 0.160 | 0.188 | 0.064 | 0.076 |
| | VisJudge (Ours) | InternVL3-8B | 0.660 | 0.533 | 0.594 | 0.545 | 0.499 | 0.391 | **0.420** |
| | | Llava-v1.6-mistral-7B | 0.605 | 0.226 | 0.536 | 0.443 | 0.432 | 0.406 | 0.403 |
| | | Qwen2.5-VL-3B-Instruct | 0.648 | 0.533 | 0.581 | **0.579** | 0.504 | 0.490 | 0.402 |
| | | Qwen2.5-VL-7B-Instruct | **0.687** | **0.574** | **0.628** | 0.576 | **0.568** | **0.513** | 0.385 |

annotation procedures, we run each model three times and average the results. All inference uses vLLM on four NVIDIA A6000 (48GB) GPUs with bfloat16 precision and a temperature of 0.8.

**Evaluation Metrics.** We assess model performance through correlation analysis using the Pearson coefficient and error metrics (MAE and MSE) compared to human scores. We also analyze score distributions to identify systematic biases. Metrics are computed for each sub-dimension and aggregated across the three main evaluation dimensions.

## 5.2 EXPERIMENTAL RESULTS AND ANALYSIS

### 5.2.1 CAN MLLMS ASSESS VISUALIZATION AESTHETICS AND QUALITY LIKE HUMANS?

Table 3 presents a comprehensive performance comparison of 12 representative models including the latest GPT-5 across our evaluation framework, revealing significant capability differences and systematic limitations in current MLLMs for visualization assessment.

**Hierarchical Capability Structure.** Current MLLMs exhibit a hierarchical performance pattern: "Fidelity" is relatively strong, "Expressiveness" is moderate, and "Aesthetics" is the most challenging, with average MAE around 0.755 and correlations of only 0.177–0.408 across the three aesthetic sub-dimensions, reflecting the difficulty of abstract design principles and cultural context for cur-

Figure 4: Distribution and bias analysis of MLLM scores. Score distribution density curves showing the rating patterns of different models compared to human experts on the 1–5 scale.

rent models. Models also exhibit distinct evaluation characteristics: GPT-5 performs consistently across dimensions; GPT-4o is relatively strong in Color Harmony (MAE 0.657); Claude-4-Sonnet excels in Semantic Readability (MAE 0.757); and Gemini-2.0-Flash leads in Data Fidelity (MAE 0.829). Among open-source models, Qwen2.5-VL-72B-Instruct achieves the best overall performance (MAE 0.702, Corr. 0.440), approaching commercial model capabilities.

**Domain-Specific Fine-tuning Effectiveness.** VISJUDGE (based on Qwen2.5-VL-7B-Instruct) achieves the best overall performance with MAE 0.421 and correlation 0.687, representing a 23.9% MAE reduction over GPT-5 and a 42.5% correlation improvement over GPT-4o. Domain-specific fine-tuning consistently yields 30–40% error reductions across different backbone architectures: Qwen2.5-VL-3B-Instruct achieves a 40.2% reduction (from 0.821 to 0.491), InternVL3-8B 31.8% (0.793 to 0.541), and Llava-v1.6-mistral-7B 31.5% (0.724 to 0.496), demonstrating robust generalization across diverse model families. In subsequent analyses, VISJUDGE refers to this best-performing variant unless otherwise specified. For analysis of how training data scale affects these results, see Appendix E.1.

### 5.2.2 Do MLLMs Exhibit Human-like Scoring Behaviors?

To analyze systematic biases in model evaluation behavior, we examine score distribution patterns across representative baseline models. Figure 4 reveals significant bias issues in current MLLMs compared to human experts ($\mu = 3.13$).

**Systematic Biases in Current Models.** Most models exhibit score inflation with rightward-shifted distributions. Qwen2.5-VL-7B-Instruct and Claude-3.5-Sonnet show the most severe inflation ($\mu = 3.89$ and $\mu = 3.87$), while Gemini-2.0-Flash, GPT-4o, Claude-4-Sonnet, and GPT-5 demonstrate moderate inflation ($\mu = 3.64$, $\mu = 3.53$, $\mu = 3.56$, and $\mu = 3.36$ respectively). Notably, GPT-5 shows relatively better control compared to other inflated models. Conversely, Gemini-2.5-Pro exhibits overly conservative behavior ($\mu = 3.02$). Additionally, models like Qwen2.5-VL-7B-Instruct, Claude-3.5-Sonnet, and Gemini-2.0-Flash exhibit sharp peaks around 4.0, indicating excessive score concentration that limits discriminative capability.

**Effective Bias Correction through Fine-tuning.** Our VISJUDGE achieves near-perfect alignment with human scoring patterns ($\mu = 3.11$) and maintains a broader, more balanced distribution. This demonstrates that domain-specific fine-tuning effectively corrects both inflation and concentration issues, achieving human-like evaluation behaviors.

### 5.2.3 How Does Visualization Complexity Affect Model Performance?

To understand model robustness across varying complexity, we analyze representative baseline models on three visualization types: single visualizations, multiple visualizations, and dashboards. Figure 5 shows the main trends.

**Performance Degradation with Complexity.** All models show consistent performance degradation: single visualizations > multiple visualizations > dashboards. VISJUDGE achieves the best performance across all types with correlations of 0.577 (single visualizations), 0.565 (multiple visualizations), and 0.375 (dashboards), significantly outperforming baselines. This demonstrates the effectiveness of domain-specific fine-tuning for complex multi-element interactions.

**Stability in Complex Scenarios.** Baseline models show significant instability in complex scenarios. For dashboards, most baselines experience substantial correlation drops, with Claude-3.5-

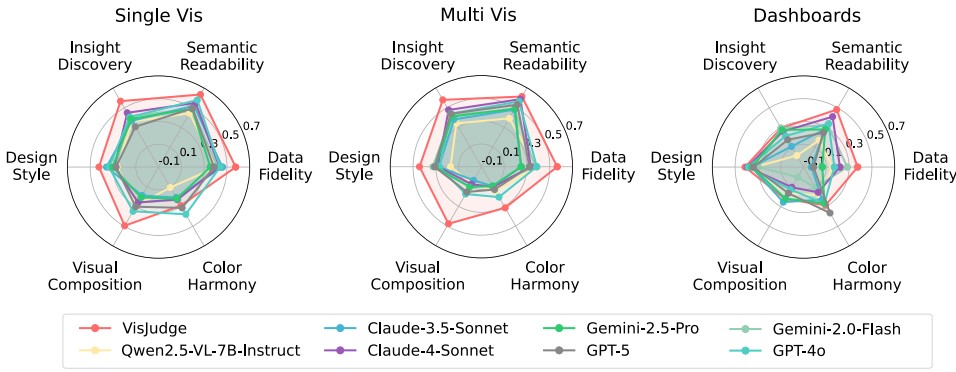

Figure 5: Model–Human rating correlation across visualization types.

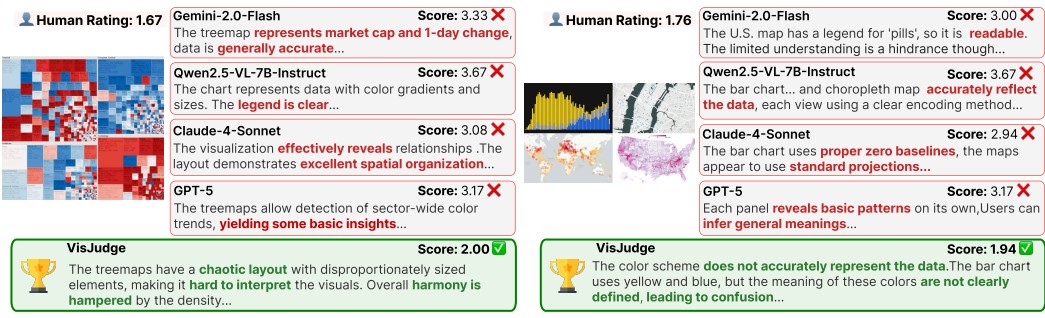

Figure 6: Model evaluation examples on low-quality visualizations.

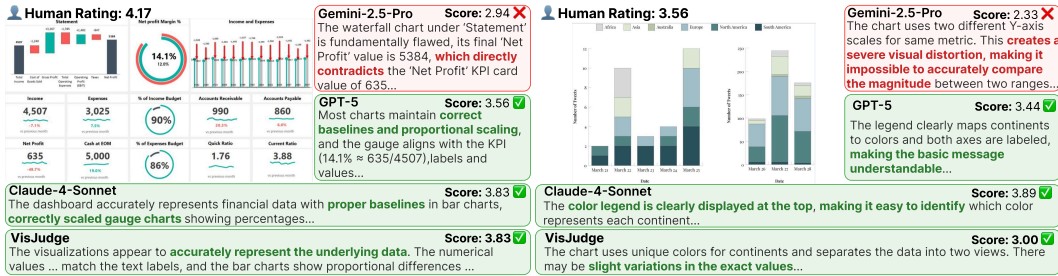

Figure 7: Case study highlighting the conservative bias of Gemini-2.5-Pro.

Sonnet and GPT-5 even showing negative correlations in Data Fidelity (-0.031 and -0.013), while VISJUDGE maintains consistency (0.224–0.482). Functional dimensions (Data Fidelity, Semantic Readability) remain stable across types, but aesthetic dimensions struggle with complex layouts, particularly Visual Composition in dashboards (most models <0.2). These findings highlight the critical importance of specialized training for robust visualization evaluation across diverse complexity levels. For detailed error analysis of multi-visualization and dashboard cases, including systematic bias patterns and failure modes across different models, see Appendix B.6.

### 5.2.4 HOW DO MODEL EVALUATION BEHAVIORS DIFFER IN PRACTICE?

To qualitatively analyze model evaluation behaviors, we conduct case studies on representative baseline models to reveal two common biases: "score inflation" and "overly conservative" assessments.

Figure 6 illustrates score inflation on low-quality visualizations. For a chaotic treemap (human rating: 1.67), baseline models give inflated scores. For instance, Qwen2.5-VL-7B-Instruct (3.67) praises its "clear legend" while ignoring the confusing layout, and Claude-4-Sonnet (3.08) incorrectly highlights "excellent spatial organization". In contrast, VISJUDGE's score of 2.00 aligns with human judgment, correctly identifying the "chaotic layout" that impairs interpretation.

Conversely, Figure 7 highlights the overly conservative bias of Gemini-2.5-Pro. For a high-quality dashboard rated 4.17 by humans, Gemini-2.5-Pro gives a disproportionately low score of 2.94, fo-

Table 4: MatPlotAgent quality improvement with different feedback models.

| Generation Model | Direct Decoding (Baseline) | Qwen2.5-VL-7B as Feedback | Qwen2.5-VL-72B as Feedback | VISJUDGE as Feedback |
|---|---|---|---|---|
| GPT-5 | 67.15 | 66.49 | 69.80 | **72.07** |
| GPT-4o | 62.71 | 58.49 | 63.53 | **71.62** |
| Gemini-2.5-Pro | 68.10 | 65.04 | 66.12 | **72.20** |
| Claude-4-Sonnet | 64.50 | 67.25 | 65.20 | **72.23** |
| Claude-3.5-Sonnet | 63.89 | 59.23 | 63.14 | **69.04** |
| Qwen2.5-VL-72B-Instruct | 61.18 | 58.70 | 60.30 | **67.32** |
| Qwen2.5-VL-7B-Instruct | 49.76 | 45.38 | 44.94 | **55.29** |

cusing on a single data inconsistency while overlooking the chart's overall effectiveness. Similarly, for another chart (human rating: 3.56), it scores only 2.33 due to the use of dual Y-axes. While other models like GPT-5 and Claude-4-Sonnet provide scores closer to human ratings, VISJUDGE also demonstrates more balanced evaluations (3.83 and 3.00, respectively). Additional case studies across different quality levels and comprehensive model error analysis are provided in Appendix B.

### 5.2.5 CAN VISJUDGE GENERALIZE TO REAL-WORLD APPLICATIONS?

To validate practical generalization, we integrated VISJUDGE into two real-world visualization systems with significant distribution shifts from our training data (detailed setup in Appendix E.2).

**Visualization Generation.** VISJUDGE provides feedback to MatPlotAgent (Yang et al., 2024) for iterative quality improvement. Table 4 shows consistent improvements across seven generation models (+6.07 points average). Notably, base models without domain fine-tuning often degrade performance (7B: -2.39, 72B: -0.61 average). This degradation stems from their systematic evaluation biases (Section 5.2.2): score inflation and poor discriminative capability lead to misleading feedback that misdirects generation models toward suboptimal outputs. In contrast, VISJUDGE's accurate quality assessment consistently improves all generators. Even state-of-the-art models like GPT-5 and Claude-4-Sonnet achieve +4.92 and +7.73 improvements, demonstrating that domain-specific fine-tuning is essential for providing effective feedback.

**Visualization Recommendation.** VISJUDGE as a reward model in HAIChart (Xie et al., 2024) improves recommendation accuracy on the VizML dataset: +4.40% (Data Queries Hit@1), +2.50% (Overall Hit@1), +5.30% (Overall Hit@3). While 7B and 72B base models show minimal to moderate improvements, VISJUDGE achieves consistently larger gains across all metrics, outperforming both base models and the baseline (detailed results in Appendix E.2). This validates VISJUDGE's broad applicability across generation and recommendation tasks.

## 6 CONCLUSION

This paper constructs VISJUDGE-BENCH and fine-tunes VISJUDGE to validate the effectiveness of domain-specific training. Our research finds that existing MLLMs (including GPT-5) show significant gaps with human experts in visualization evaluation, exhibiting issues like scoring bias. VISJUDGE effectively mitigates these problems, achieving 23.9% MAE reduction and 60.5% correlation improvement over GPT-5. VISJUDGE-BENCH provides a standardized evaluation platform for the community, while VISJUDGE's success demonstrates that domain-specific training is a viable approach for improving MLLMs' evaluation capabilities, supporting future work on finer evaluation and higher-quality visualization generation.

## 7 LIMITATIONS AND FUTURE WORK

VISJUDGE-BENCH has several limitations: (1) it focuses on static visualizations, with limited coverage of dynamic and interactive content; (2) web-collected samples typically lack raw data, so evaluation relies on visual presentation alone, limiting data fidelity verification; and (3) point-estimate scoring may not fully capture the diversity of human preferences. Future work includes expanding dynamic visualization samples with corresponding evaluation criteria, exploring ways to obtain samples with raw data for more thorough fidelity assessment, and investigating distribution-based evaluation methods to better reflect human preference diversity.

ETHICS STATEMENT

The VisJudge-Bench framework presented in this work aims to improve multimodal large language models' capabilities in visualization quality assessment and promote the development of automated visualization evaluation technology. We believe this work will not produce direct negative social impacts, but recognize that the framework should be used with caution and ethical oversight when applied to sensitive domains or potentially harmful models. Although VisJudge-Bench aims to objectively assess visualization quality, the base models it relies on (such as Qwen2.5-VL-7B) or the datasets used to construct the benchmark may inadvertently reflect biases. Future work could investigate the fairness implications of these evaluation features across different populations, cultural backgrounds, and visualization styles. We particularly focus on the following ethical considerations: (1) strict compliance with copyright and usage terms during data collection; (2) ensuring fair compensation and voluntary participation for expert annotators; (3) avoiding content that may reinforce stereotypes or biases in benchmark design; (4) open-source release aimed at promoting community development rather than commercial monopoly. All research was conducted in strict compliance with ICLR ethics guidelines.

REPRODUCIBILITY STATEMENT

To ensure reproducibility, we provide comprehensive documentation and resources. The complete VisJudge-Bench construction process is detailed in Appendix A, covering data collection, filtering, and expert annotation protocols. The six-dimensional evaluation framework and VisJudge implementation are described in Appendices C and D, respectively. We release the complete dataset with 3,090 expert-annotated samples across three categories (`single_vis`, `multi_vis`, `dashboard`) at `https://github.com/HKUSTDial/VisJudgeBench`, where each sample includes visualization images, six-dimensional scores, and evaluation prompts. All experimental configurations, evaluation metrics (MAE, MSE, Pearson correlation), and human consistency analysis methods are fully documented to enable result reproduction.

ACKNOWLEDGMENTS

This paper was supported by the NSF of China (62402409); Youth S&T Talent Support Programme of Guangdong Provincial Association for Science and Technology (SKXRC2025461); the Young Talent Support Project of Guangzhou Association for Science and Technology (QT-2025-001); Guangdong Basic and Applied Basic Research Foundation (2023A1515110545); Guangzhou Basic and Applied Basic Research Foundation (2025A04J3935); and Guangzhou-HKUST(GZ) Joint Funding Program (2025A03J3714).

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

# Appendix Contents

## LLM Usage Statement

We used Claude-4-Sonnet for English grammar polishing and consulted Claude-4-Sonnet for suggestions on figure layout and color design. During dataset construction, we used GPT-4o for automated adaptive question generation and chart metadata extraction. All code was written, reviewed, and verified by the authors. All prompts contained no private or sensitive data. Large language models did not provide any novel algorithmic ideas or academic claims; the authors take full responsibility for the content. Large language models are not authors of this paper.

## A  Dataset Construction Details and Statistical Analysis

### A.1  Data Collection and Construction Pipeline

To build a large-scale and diverse visualization dataset, we designed and implemented a systematic web crawling and data filtering pipeline. This process aims to collect a wide range of visualizations from the web, spanning from poorly designed examples to professional exemplars, while ensuring that all collected data is of high relevance and quality. The entire pipeline consists of three core stages: keyword generation, a high-throughput crawling architecture, and multi-stage filtering.

**Keyword Generation Strategy.**  The foundation of our data collection is a meticulously designed keyword generation strategy to ensure broad coverage across visualization types, quality levels, and application domains.

- **Base Keyword Lexicon:** We first established a base lexicon of over 200 professional visualization terms, such as "professional bar chart design," "clean line graph visualization," and "business intelligence dashboard."
- **Visualization Type Expansion:** Building on this, we systematically incorporated over 30 different chart types, covering basic charts (e.g., bar, line, pie charts), advanced visualizations (e.g., Sankey diagrams, treemaps, radar charts), and interactive systems (e.g., interactive dashboards, animated charts).
- **Quality Modifier Combination:** To intentionally capture charts of varying quality levels, we programmatically combined chart types with high-quality modifiers (e.g., "professional," "clean," "effective," "well-designed") and low-quality modifiers (e.g., "poor," "confusing," "cluttered," "misleading").
- **Domain-Specific Terminology:** We also integrated professional terminology from over 20 application domains (e.g., business, finance, healthcare, education) to generate context-specific search queries, such as "financial dashboard," "sales performance chart," and "COVID cases chart."

This automated strategy ultimately generated over 2,000 unique, high-quality search keywords, laying a solid foundation for our large-scale data crawling efforts.

**High-Throughput Crawling and Preliminary Filtering**  To efficiently collect a vast number of candidate images from the web, we developed a high-throughput crawling architecture based on Bing Image Search. This architecture utilizes multi-threaded, asynchronous requests to fetch up to 10 pages of search results for each keyword, maximizing data recall. High-resolution image URLs were reliably extracted by parsing JSON data embedded within the web pages. During the crawling phase, we implemented an initial round of automated preliminary filtering:

- **Size Filtering:** We strictly filtered images by size, requiring a minimum width of 400 pixels, a minimum height of 300 pixels, and a total area of at least 150,000 pixels. This effectively eliminated low-resolution thumbnails and icons.
- **Heuristic Content Pre-screening:** We conducted a rapid pre-assessment of image content using programmatic analysis techniques. By employing an edge detection algorithm and color complexity analysis (counting the number of unique colors), we discarded a significant number of images that were either too simple (e.g., solid-color backgrounds, blank images) or too complex (e.g., real-world photographs), as these typically do not represent data visualizations.

This stage yielded a large-scale preliminary dataset containing tens of thousands of candidate images, laying the groundwork for subsequent fine-grained refinement.

**Fine-Grained Filtering and Hierarchical Classification via Multimodal LLMs**  To precisely filter high-quality, relevant visualizations from the preliminary dataset and organize them into a structured classification, we designed a fine-grained filtering pipeline centered around an MLLM.

- **Perceptual Hash Deduplication:** Before semantic analysis, we first employed a Perceptual Hashing (pHash) algorithm to deduplicate all candidate images. This technique identifies visually identical or highly similar images, regardless of differences in size, format, or compression. By setting a strict similarity threshold (Hamming distance < 5), we effectively ensured the diversity of the final dataset and eliminated redundancy.

- **Prompt-Based AI Semantic Filtering:** We utilized an advanced MLLM (e.g., GPT-4o) as our core classifier. We engineered a highly restrictive system prompt that defined the model's primary task as that of a *strict filter* rather than a simple classifier. This prompt compelled the model to adhere to the following top-priority rules:

  1. **Reject Non-Screenshot Images:** Any image appearing to be a photograph, containing tilted perspectives or distortions, or including real-world environments (e.g., monitor bezels, keyboards, desks) was immediately classified as non-compliant (`non_visualization`).
  2. **Reject Images with People:** Any image containing human figures (including cartoons) or body parts (e.g., hands, fingers) was strictly filtered out.
  3. **Reject Work-in-Progress and Development Interfaces:** We mandated that only "finished" visualizations be retained. Any screenshot depicting the visualization creation process—such as those including software UI elements (menus, toolbars, property panels), code editors (like Jupyter Notebooks), or configuration windows—was also classified as non-compliant.

  The full content of this prompt is detailed in the "Prompt Template" box below.

- **Hierarchical Content Classification:** Only images that passed all the stringent screening criteria and were identified as "clean, front-facing, person-free visualization screenshots" proceeded to the classification stage. The model then categorized them into a hierarchical system based on their structure and function, primarily including: single visualizations, multiple visualizations, and dashboards.

---

**Prompt Template: Fine-Grained Filtering and Classification via Multimodal LLM**

You are a professional data visualization analysis expert. Your core mission is to strictly filter and accurately classify data visualization images.

WARNING – Highest Priority Principle: Absolutely reject all photographs and any images containing people.
Your primary duty is to act as a rigorous filter. Before evaluating the content of an image, you must first assess its form.

- Is it a photograph? If yes, immediately classify as non_visualization.
- Does it contain people? If yes, immediately classify as non_visualization.
- Is it tilted or in perspective? If yes, immediately classify as non_visualization.

Only when an image perfectly meets the standard of a "clean, front-facing, person-free screenshot" can you proceed to analyze its content.

---

Strict Filtering Criteria (Classify as non_visualization if any condition is met):

1. Reject ALL Photographs:
   - Characteristic: Reject any image that appears to be taken with a camera rather than a direct screenshot.
   - Clues:
     - Tilted Angle/Perspective Distortion: The image is not flat and front-facing.

- Device Bezels: Physical borders of a laptop, monitor, phone, or tablet are visible.
- Real-World Environment: Backgrounds like desks, offices, conference rooms, keyboards, or mice are visible.
- People or Body Parts: Any presence of people, hands, or fingers.
- Reflections or Screen Moire: Reflections of ambient light on the screen.

2. Reject ALL Images with People:

- Characteristic: Absolutely forbidden. Any image containing people in any form (full body, portrait, cartoon) or body parts (hands, fingers) must be rejected.

3. Reject Marketing/Concept Images:

- Characteristic: Images that look like stock photos, promotional materials, website banners, or stylized concept designs. These often have artistic effects, tilted perspectives, or non-data elements and are not genuine analytical tool interfaces.

---

Content Classification Criteria (Applicable only to clean screenshots that pass the above filters):

1. Single Visualization (single_view): A pure, single, complete data chart.

- Ultra-Strict Prerequisites (All must be met):
  (a) Pure Chart: The main subject of the image must be the chart itself, with no external UI elements.
  (b) No Editing Controls: It must absolutely not contain any UI elements for configuring or editing the chart (e.g., toolbars, property panels, formatting panes, pop-up menus). If such elements are present, it must be classified as non_visualization.
  (c) Front-facing, person-free, non-photograph screenshot.

2. Multiple Visualizations (multi_view): A pure composition of multiple data charts for analysis.

- Ultra-Strict Prerequisites:
  (a) Pure Chart Composition: The image must only contain data charts, with absolutely no other types of elements.
  (b) Multiple Charts: It must contain 2 or more independent data charts.
  (c) Front-facing, person-free, non-photograph, non-editing interface screenshot.

3. Dashboard (dashboard): An end-user-facing interactive interface for data exploration and monitoring.

- Ultra-Strict Prerequisites: Must be a front-facing, clean screenshot, absolutely free of any people or device bezels.
- Core Features:
  - Composed of multiple, coordinated data charts and KPI metrics. (A single chart does not constitute a dashboard).
  - Interactive elements are end-user-facing and intended for data consumption (e.g., filtering, drilling down, switching views), not for chart creation or editing.

4. Non-Compliant Image (non_visualization): Any image that is not a pure, finished data visualization product.

- This category serves as a "catch-all" to filter out all visualizations that do not meet the strict criteria.
- Core Judgment: Is this image showing something "in progress" or is it a "finished product"? Anything "in progress" is non-compliant.

---

Decision-Making Process (Strictly Adhered To):

1. Step 1: Check for "Non-Compliant Image" (non_visualization). (This is the highest-priority filter).

2. Step 2: Classify the "pure visualization products" that pass the first step.

3. Step 3: Differentiate between Multiple Visualizations and Dashboard.

Return Format: JSON object with the following structure:

```
{
  "category": "Primary category in English (single_view/multi_view/
    dashboard/non_visualization)",
  "type": "Sub-type in English",
  "confidence": "Confidence score (0-100)",
  "reasoning": "Provide the core reason for the judgment, e.g., 'The
    image is a photograph/contains people/is not front-facing, and is
    therefore a non-compliant image.'"
}
```

This multi-stage pipeline, combining heuristic pre-screening, perceptual hash deduplication, and AI-driven semantic refinement, enables the fully automated construction of a high-quality, structured visualization dataset from vast web-scale data. It significantly reduces the manual annotation burden while ensuring the relevance and quality of the collected data.

## A.2 Dataset Statistics and Distribution Analysis

### A.2.1 Overall Dataset Statistics and Distribution

VisJudge-Bench consists of 3,090 professionally assessed visualizations covering the full range of modern visualization design. It was constructed to ensure broad coverage across visualization types, evaluation dimensions, and quality levels, reflecting practices in business intelligence, academic research, and data journalism. The collected quality scores approximately follow a normal distribution (mean = 3.13, std = 0.72, range = 1.00–4.89; see Figure 8a), capturing a broad range from poor to exemplary designs. Figure 9 presents representative visualization examples across different quality score ranges (from 1–2 to 4–5), showcasing the diversity of visualization types and the clear quality distinctions captured by our evaluation framework. All samples include complete six-dimensional annotations, enabling users to study visualization quality holistically as well as across specific types, subtypes, and evaluation dimensions.

### A.2.2 Visualization Type and Subtype Analysis

**Visualization Classification.** A hierarchical taxonomy organizes visualizations by structural complexity and functional purpose. It includes three major categories: *single visualizations* (1,041 samples, 33.7%), *multiple visualizations* (1,024 samples, 33.1%), and *dashboards* (1,025 samples, 33.2%). These categories further expand into 22, 5, and 5 subtypes respectively, ensuring representation from basic charts (e.g., bar, pie, line) to advanced analytical dashboards. This classification system allows users to study quality differences across visualization types and subtypes. Detailed information can be found in Table 5, and Figure 8 illustrates the quality score distributions across these three major categories, revealing distinct distribution patterns for each visualization type.

For the single visualizations (1,041 samples, 22 subtypes), we observe a clear head-tail distribution. Four basic chart types—bar, pie, line, and area charts—account for 46.1% (480 samples), reflecting their central role in practical visualization design. A second tier of commonly used analytic charts (e.g., treemap, Sankey, heatmap, scatter, histogram, funnel, bubble) contributes 38.0% (396 samples). The remaining 15.9% (165 samples) consists of more specialized forms, where network graphs make up about 2.2% and map-based views (choropleth and point maps) around 3.6%, with the rest spread across other rare, domain-specific encodings. This skewed distribution is in line with (Borkin et al., 2013) empirical studies of real-world visualization usage.

Reflecting real-world usage patterns, the distribution of our multi visualizations samples (1,024 samples, 5 subtypes) is heavily skewed towards Comparison views, which constitute the majority (65.4%, 670 samples). Small multiples (19.0%) and coordinated views (9.5%) follow as key patterns, with the remaining 6.1% covering overview–detail layouts and other forms. This composition

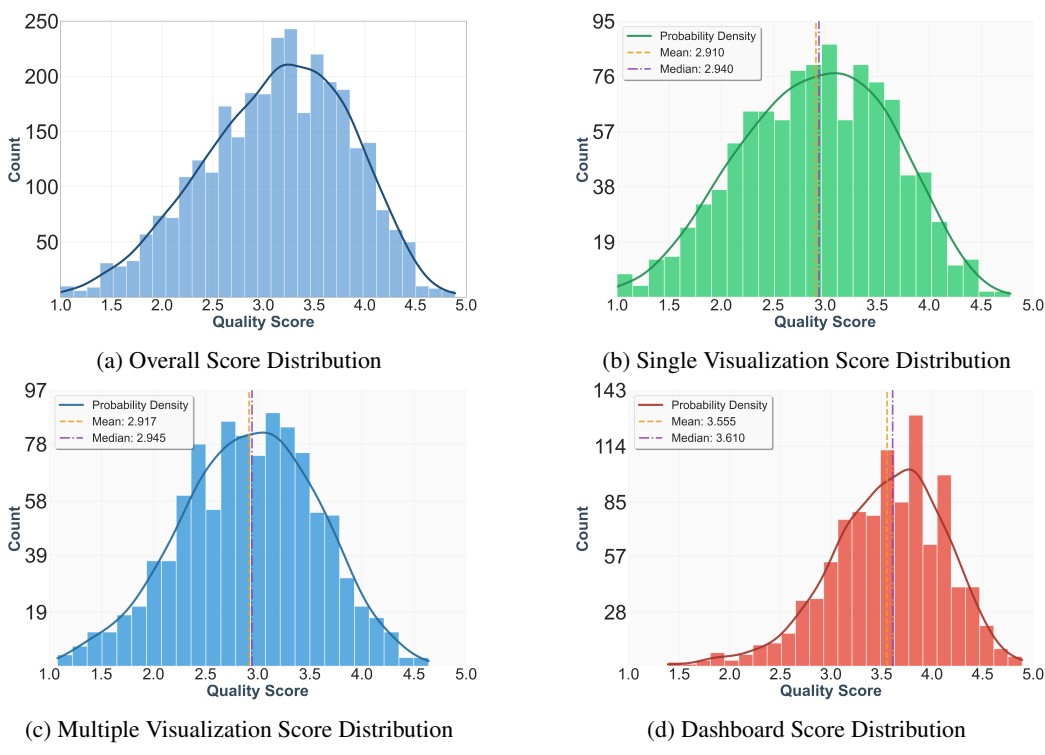

(a) Overall Score Distribution

(b) Single Visualization Score Distribution

(c) Multiple Visualization Score Distribution

(d) Dashboard Score Distribution

Figure 8: Quality score distributions of the dataset across different visualization categories.

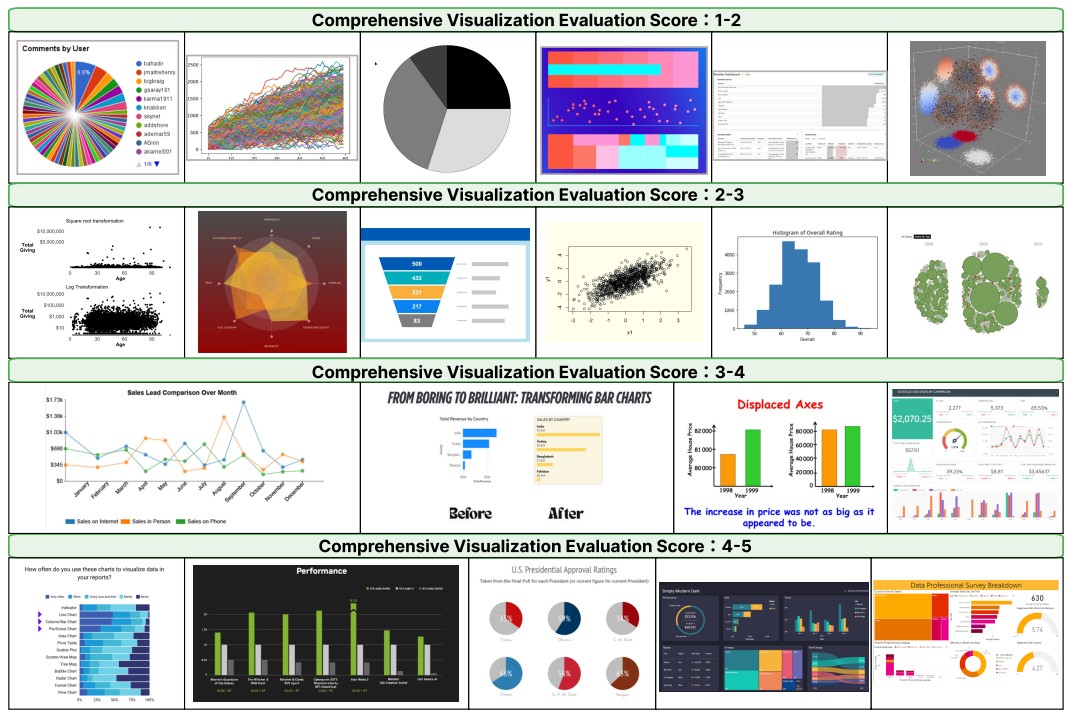

Figure 9: Representative samples from VISJUDGE-BENCH

aligns with established characterizations (e.g., (Chen et al., 2020)), ensuring our benchmark captures a representative cross-section of common multi-view configurations.

Table 5: VISJUDGE-BENCH statistical detailed information.

| Vis Type | Count | Proportion | #-Subtype | Subtype Details | | | |
|---|---|---|---|---|---|---|---|
| Single Vis | 1,041 | 33.7% | 22 | Bar Chart | 176 | Bubble Chart | 29 |
| | | | | Pie Chart | 129 | Choropleth Map | 25 |
| | | | | Line Chart | 100 | Radar Chart | 24 |
| | | | | Area Chart | 75 | Network Graph | 23 |
| | | | | Treemap | 62 | Candlestick Chart | 20 |
| | | | | Sankey Diagram | 61 | Gauge Chart | 20 |
| | | | | Heatmap | 55 | Box Plot | 17 |
| | | | | Scatter Plot | 49 | Point Map | 12 |
| | | | | Histogram | 48 | Word Cloud | 1 |
| | | | | Donut Chart | 47 | Violin Plot | 1 |
| | | | | Funnel Chart | 45 | Other Single View | 22 |
| Multi Vis | 1,024 | 33.1% | 5 | Comparison Views | 670 | Overview Detail | 3 |
| | | | | Small Multiples | 195 | Other Multi View | 59 |
| | | | | Coordinated Views | 97 | | |
| Dashboard | 1,025 | 33.2% | 5 | Analytical Dashboard | 743 | Strategic Dashboard | 62 |
| | | | | Operational Dashboard | 122 | Other Dashboard | 7 |
| | | | | Interactive Dashboard | 91 | | |

Among the dashboard samples (1,025 samples, 5 subtypes), Analytical dashboards are overwhelmingly dominant, constituting 72.5% (743 samples) of the dataset. The remainder consists of Operational (11.9%), Interactive (8.9%), and Strategic (6.0%) types, with a negligible fraction (0.7%) classified as other forms. This distribution, heavily weighted towards analysis- and decision-oriented tools, mirrors the real-world prevalence observed in genre analyses by (Sarikaya et al., 2019) and design surveys by (Bach et al., 2023).

### A.2.3 DESIGN STYLE DISTRIBUTION ANALYSIS

From a quantitative perspective, Figure 10 shows the score distribution for the design style dimension: this dimension has a mean score of 2.97, showing a relatively uniform distribution that indicates the dataset covers diverse design styles ranging from low to high innovation.

To further understand the style diversity in our dataset, we refer to related work on chart design styles (Borkin et al., 2013; Bateman et al., 2010). Prior research has shown that visualization design exhibits different orientations ranging from function-oriented to expression-focused approaches (Borkin et al., 2013), and that different visual styles can influence the aesthetic perception and memory effects of visualizations (Bateman et al., 2010). Based on these studies, we conducted manual inspection of all 3,090 samples in the dataset and identified the following main style categories: (1) Professional/Standard Style (53.3%), adopting clean and standardized designs commonly found in business intelligence tools and academic publications, corresponding to function-oriented design approaches; (2) Creative/Infographic Style (28.4%), demonstrating more creative design approaches prevalent in data journalism and marketing materials, reflecting expression-focused design orientations; (3) Minimalist Style (18.3%), following minimalist design principles by reducing visual elements and using generous whitespace to emphasize data clarity.

We acknowledge that this style classification is coarse-grained and somewhat subjective, as design style itself is multi-dimensional and context-dependent. A more systematic style classification system would require dedicated annotation work, which is beyond the scope of this work. Nevertheless, this preliminary analysis demonstrates that VisJudge-Bench covers diverse design approaches commonly found in real-world practice, ranging from function-oriented standardized designs to visually engaging creative expressions.

### A.2.4 APPLICATION DOMAIN DISTRIBUTION ANALYSIS

We annotated the dataset using high-level categories from the IPTC taxonomy (Li et al., 2025c). About 6.2% of samples were grouped as "other" due to limited context. Since the dataset is con-

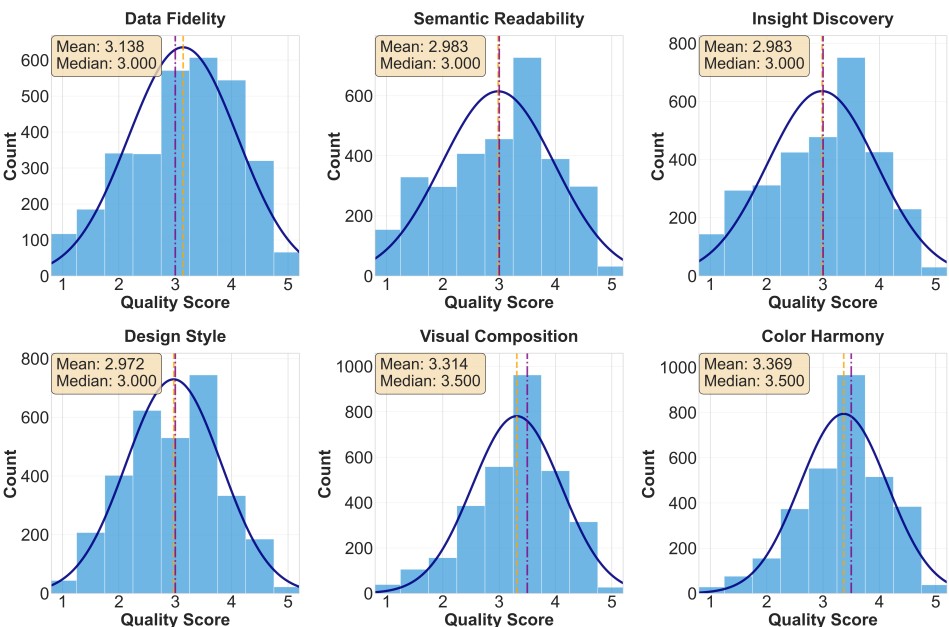

Figure 10: Quality score distributions across six evaluation dimensions.

structed from real-world web sources through our systematic crawling pipeline (see Appendix A.1), its domain distribution naturally reflects actual visualization demand and usage patterns. The annotated subset shows a clear concentration: economy, business and finance (26.3%), science and technology (20.3%), and labor (15.7%) together account for 62.3% of all labels, aligning with high practical demand in these fields. A middle tier includes society (6.6%) and domains like lifestyle and leisure, health, and politics and government (ranging from 2.7% to 4.9%), while topics like crime, law and justice and religion are rare (<1%). This concentration is even more pronounced in dashboards, where the top three categories comprise 78.5% of the data, reflecting the dominant role of business analytics in dashboard applications.

We further analyze the domain co-occurrence matrix to understand how topics are combined within the same visualization or dashboard. Economy, business and finance plays a central role and most often appears together with labor, science and technology, and society, reflecting typical themes such as employment, productivity, and general economic conditions. Science and technology often co-occurs with health, environment, and education, corresponding to topics such as medical research, climate and energy, and science education. In addition, politics and government frequently appears with economy, business and finance and society, while environment is closely linked to both science and technology and weather. These co-occurrence patterns show that many visualizations in our dataset address cross-domain topics rather than a single isolated domain, capturing common interdisciplinary scenarios in practice.

### A.2.5 QUALITY SCORE DISTRIBUTION

**Quality Grade Distribution.** To examine the overall quality of the dataset, we analyze the distribution of quality scores across different visualization categories. Figure 8 presents the histograms with fitted density curves, highlighting both the mean and median values for each category. This analysis allows us to compare quality differences between single visualizations, multiple visualizations, and dashboards.

**Quality Distribution Across Visualization Categories.** The overall quality score distribution (Figure 8a) exhibits a near-normal distribution with scores predominantly ranging from 2.0 to 4.0, indicating balanced representation of both lower and higher quality samples.

Individual visualization categories show distinct patterns. Single visualizations (Figure 8b) and multiple visualizations (Figure 8c) exhibit similar quality levels with means of 2.910 and 2.917 respectively, both displaying broad distributions across the quality spectrum. In contrast, dashboards (Figure 8d) show notably higher scores (mean: 3.555, median: 3.610) with a right-skewed distribution concentrated in higher quality ranges. This difference reflects that published dashboards typically undergo more rigorous design review as polished, production-ready tools, while single and multiple visualizations include more experimental designs with varying execution quality.

**Quality Distribution Across Evaluation Dimensions.** Figure 10 reveals distinct patterns across the six evaluation dimensions, which align with our three-tier framework of *Fidelity*, *Expressiveness*, and *Aesthetics*.

At the *Fidelity* level, *data fidelity* (mean: 3.138, median: 3.000) exhibits a balanced near-normal distribution, indicating varied success in truthful data representation—a fundamental requirement that shows substantial room for improvement across the dataset.

At the *Expressiveness* level, both *semantic readability* (mean: 2.983, median: 3.000) and *insight discovery* (mean: 2.983, median: 3.000) center around 3.0 with broad distributions, reflecting the persistent challenge of effective communication and analytical support. These similar patterns suggest that clarity and insight facilitation remain equally difficult aspects of visualization design.

At the *Aesthetics* level, we observe a gradient in achievement: *color harmony* (mean: 3.369, median: 3.500) and *visual composition* (mean: 3.314, median: 3.500) show the highest scores with right-skewed distributions, benefiting from well-established design guidelines and tool support. In contrast, *design style* (mean: 2.972, median: 3.000) shows the lowest average with broader spread, reflecting its subjective nature and the varying emphasis placed on stylistic sophistication versus functional priorities.

This hierarchical distribution pattern—from foundational data accuracy, through communicative effectiveness, to aesthetic refinement—ensures that our benchmark evaluates models across the complete spectrum of visualization quality assessment.

### A.2.6  LOW-QUALITY SAMPLE DISTRIBUTION AND DESIGN FLAW ANALYSIS

We observe a distinct correlation between visualization granularity and quality stability. The proportion of low-quality samples (score < 2) decreases markedly as complexity increases: from 12.5% in single views (130/1041), to 9.2% in multi-views (94/1024), and down to just 1.3% in dashboards (13/1025). A likely driver is: dashboards often represent mature, engineered products, whereas the single-view dataset contains more exploratory, or structurally complex charts (e.g., network diagrams, treemaps, heatmaps) prone to design failures.

**Single visualizations.** Quality issues in this subset are unevenly distributed, clustering heavily in complex or less common categories. Five specific categories, Treemaps, Scatter Plots, Network Graphs, Heatmaps, and Pie Charts, collectively account for 57.7% of all low-quality instances. Across dimensions, *design style* emerges as the primary bottleneck (mean 2.623, 34% low-score rate), followed by semantic readability (2.831, 31%) and insight discovery (2.779, 32%). Our analysis reveals that the mechanisms of failure diverge significantly based on chart complexity. For complex chart types, quality problems are typically multi-dimensional rather than isolated to a single criterion. In the Candlestick Chart, 55% of such samples fail on at least three dimensions simultaneously, as semantic illegibility triggers a chain reaction that degrades *data fidelity* and *insight discovery*. Conversely, standard formats like Bar and Line charts generally maintain robust fidelity and readability (mean approx 3.000) while suffering primarily from isolated lacks in aesthetic refinement rather than fundamental communicative breakdowns.

**Multiple visualizations.** While Comparison Views have a low-score rate (7.2%), their predominance in the dataset means they contribute over half (51.1%) of all low-quality multi-views. A critical finding here is the divergence between visual and semantic performance. *Visual composition* and *color harmony* achieve high stability (mean > 3.150, low-score rates < 12%), yet *semantic readability* and *insight discovery* lag significantly (mean < 2.700, low-score rates > 33%). This indicates that the core challenge in multi-view design is not layout or coloring, but semantic integration—users struggle to bridge information across views. Furthermore, these semantic failures are strongly coupled with Data Fidelity: in Comparison Views, approximately 20% of cases fail on *data*

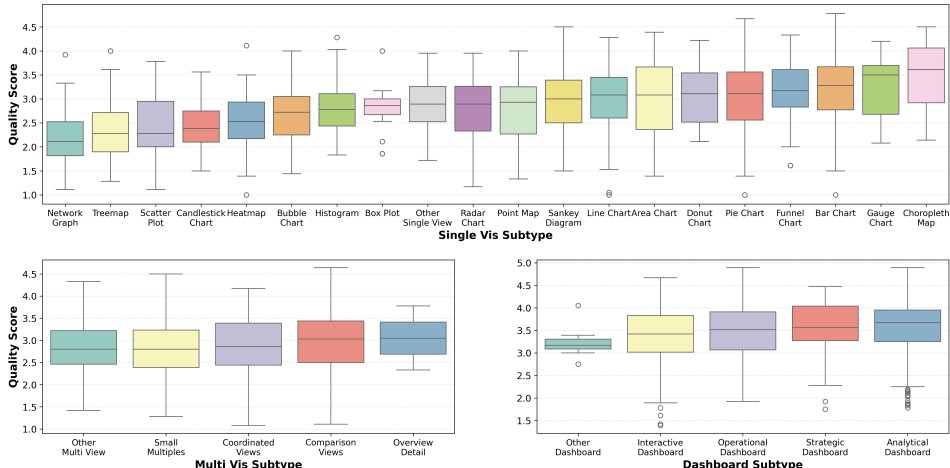

Figure 11: Quality score distributions across visualization subtypes.

*fidelity*, *semantic readability*, and *insight discovery* simultaneously, pointing to a systemic inability to convey trustworthy information.

**Dashboards.** Representing the most stable layer, dashboards exhibit high baseline scores, particularly in visual dimensions (*visual composition* and *color harmony* mean > 3.650, low-score rates < 3.5%). The few remaining low-quality cases are concentrated in Analytical and Interactive subtypes (38.5% of total failures). We observe significant error coupling in these high-stakes scenarios: for instance, 12.1% of Interactive Dashboards suffer from joint failures in *data fidelity* and *semantic readability* and 6.1% of Analytical Dashboards are simultaneously low on *semantic readability* and *insight discovery*. This suggests that dashboard failures are rarely aesthetic but functional, manifesting in high-stakes decision-making scenarios as coupled deficits in semantic explanation, metric summarization, and insight extraction.

To illustrate these design flaw patterns, we provide detailed case studies in Appendix B: Appendix B.1, B.2, and B.3 present representative cases across high-score (>4), medium-score (2–4), and low-score (<2) ranges for different visualization types; Appendix B.4 provides typical low-score examples and problem analyses for the three core dimensions—data fidelity, expressiveness, and aesthetics. These cases clearly demonstrate common quality issues and failure modes for each visualization type.

### A.2.7 TEST SET DISTRIBUTION ANALYSIS

To ensure reliable and comprehensive evaluation of model performance, we partitioned the dataset into training (70%, 2,163 samples), validation (10%, 279 samples), and test (20%, 648 samples) sets using stratified sampling based on visualization types. Table 6 presents the detailed distribution of the test set across visualization types and subtypes, demonstrating that the stratified sampling successfully maintains proportional representation consistent with the overall dataset.

The test set comprises 231 single visualizations (35.6%), 209 multiple visualizations (32.3%), and 208 dashboards (32.1%), closely mirroring the overall dataset distribution. Within single visualizations, the test set covers 20 distinct chart types, ranging from common charts like bar charts (37 samples) and pie charts (27 samples) to specialized visualizations such as Sankey diagrams (14 samples) and network graphs (5 samples). Multiple visualizations include 135 comparison views, 41 small multiples, and 20 coordinated views, while dashboards predominantly feature analytical dashboards (150 samples) alongside operational and interactive dashboards. The test set also preserves domain diversity, covering all major application domains to ensure comprehensive evaluation across different real-world contexts.

The test set maintains quality score distribution characteristics similar to the full dataset, with a mean of 3.13 and standard deviation of 0.72, ranging from 1.11 to 4.89. Score distribution across quality ranges shows 7.4% low-quality samples (1.0–2.0), 31.5% below-average samples (2.0–3.0), 49.5%

Table 6: Test set statistical detailed information (N=648, 20% of VISJUDGE-BENCH).

| Vis Type | Count | Proportion | #-Subtype | Subtype Details | | | |
|----------|-------|-----------|-----------|-----------------|---|---|---|
| Single Vis | 231 | 35.6% | 20 | Bar Chart | 37 | Bubble Chart | 7 |
| | | | | Pie Chart | 27 | Other Single View | 6 |
| | | | | Line Chart | 21 | Choropleth Map | 6 |
| | | | | Area Chart | 15 | Radar Chart | 6 |
| | | | | Treemap | 14 | Candlestick Chart | 5 |
| | | | | Sankey Diagram | 14 | Gauge Chart | 5 |
| | | | | Heatmap | 12 | Network Graph | 5 |
| | | | | Histogram | 11 | Point Map | 4 |
| | | | | Scatter Plot | 11 | Box Plot | 4 |
| | | | | Donut Chart | 11 | | |
| | | | | Funnel Chart | 10 | | |
| Multi Vis | 209 | 32.3% | 4 | Comparison Views | 135 | Other Multi View | 13 |
| | | | | Small Multiples | 41 | | |
| | | | | Coordinated Views | 20 | | |
| Dashboard | 208 | 32.1% | 5 | Analytical Dashboard | 150 | Other Dashboard | 1 |
| | | | | Operational Dashboard | 25 | | |
| | | | | Interactive Dashboard | 19 | | |
| | | | | Strategic Dashboard | 13 | | |
| **Total** | | | | | | | **648 samples** |

above-average samples (3.0–4.0), and 11.6% high-quality samples (4.0–5.0). This balanced distribution ensures comprehensive evaluation across the full quality range, enabling robust assessment of model performance on both challenging low-quality and high-quality visualizations.

## A.3 ANNOTATION PROCESS

### A.3.1 ANNOTATOR RECRUITMENT STANDARDS

To ensure high-quality responses and reduce the risk of careless or malicious submissions, we implemented strict screening criteria during the annotator recruitment process.

To maintain annotation quality, we applied the following recruitment criteria:

- **Education:** Participants were required to have completed at least a Bachelor's degree. Preference was given to those with a Master's, professional, or doctoral degree to ensure familiarity with analytical and design tasks.

- **Approval Rating:** Only individuals with a historical approval rate between 97% and 100% were permitted to participate, reflecting a track record of reliable and consistent task completion on the platform.

- **Approved Projects Count:** Annotators were selected from those who had completed between 100 and 10,000 approved projects, ensuring adequate experience with crowdsourcing workflows.

- **English Language:** All participants were required to be native English speakers to guarantee accurate comprehension of visualization-related terminology and rubric-based questions.

- **Occupation Field:** We targeted professionals working in relevant domains such as arts, business, education, finance, STEM, public administration, and product design, to match the content and context of visual analysis tasks.

- **Job Classification:** Participants were drawn from white-collar, creative, and IT-related professions, including developers, designers, analysts, and content creators, all of whom typically interact with visual content in their daily work.

- **Last Project Completed:** Annotators were required to have completed a project within the past 180 days, ensuring recent and active engagement with the platform.

- **Age:** To maintain a cognitively active and professionally engaged participant pool, we limited participation to those aged between 20 and 50 years.

- **Technical Skills:** We prioritized individuals proficient in data science, product design, front-end development, computer science, and other related technical fields that support informed and thoughtful visual reasoning.

### A.3.2 ANNOTATION TASK DESIGN

Our VISJUDGE-BENCH contains 3,090 visualizations, each requiring evaluation across 6 dimensions. To ensure annotation quality and allow annotators to familiarize themselves with the evaluation criteria, we organized the annotations into batches of 15 images per task, with each batch carefully balanced to include 5 single visualizations, 5 multiple visualizations, and 5 dashboards. Before starting each task, annotators were presented with detailed explanations of the evaluation framework, including the meaning and significance of each dimension (Fidelity, Expressiveness, and Aesthetics), enabling them to quickly understand the task requirements and evaluation standards. With 6 questions per image, each annotation task comprised 90 questions and typically took 30–60 minutes, depending on annotator familiarity with the criteria and visualization complexity. Each task was independently annotated by three qualified participants to ensure reliability through majority voting and enable inter-annotator agreement analysis. Annotators were compensated at an estimated hourly rate of $10 USD, which is competitive relative to the platform average and helped attract and retain qualified workers able to devote adequate attention to the task. Together with our strict screening criteria and embedded validation checks, this compensation structure supported the collection of high-quality annotations.

Each visualization is evaluated across six dimensions derived from our "Fidelity, Expressiveness, and Aesthetics" framework: (1) **Data Fidelity**, which assesses whether the visual representation accurately reflects the underlying data; (2) **Semantic Readability**, which evaluates whether information is clearly conveyed; (3) **Insight Discovery**, which measures whether meaningful patterns are discoverable; (4) **Design Style**, which assesses aesthetic innovation and uniqueness; (5) **Visual Composition**, which evaluates spatial layout and balance; and (6) **Color Harmony**, which measures color coordination and effectiveness. For each dimension, annotators provide ratings on a 1–5 scale, where each rating level is accompanied by clear descriptive criteria to ensure consistent interpretation across annotators (see Appendix C for detailed evaluation questions and scoring criteria for each dimension).

### A.3.3 ANNOTATION INTERFACE AND WORKFLOW

We designed a dedicated crowdsourcing interface to ensure annotators clearly understood the task, followed a structured workflow, and submitted high-quality responses. Before beginning the evaluation, participants were presented with both a brief and an extended task introduction. The short version stated:

> *Evaluate 15 data visualizations with 90 simple multiple-choice questions (6 per chart) covering Fidelity (data accuracy), Expressiveness (information clarity), and Aesthetics (visual aesthetics). Each question has clear 1–5 rating descriptions to make evaluation straightforward. Please only participate if you can provide thoughtful responses—we've designed this to be as simple as possible for you!*

The extended version was shown in full on the task interface:

> *Welcome! Thank you for joining our study on data visualization quality. You will evaluate 15 data visualizations with 90 simple multiple-choice questions based on three classical design principles:*
>
> > *- Fidelity: Data Fidelity – whether the visual representation accurately reflects the underlying data.*

> *- Expressiveness: Semantic Readability, Insight Discovery – whether information is clearly conveyed and meaningful patterns are discoverable.*
>
> *- Aesthetics: Design Style, Visual Composition, Color Harmony – whether the visualization has aesthetic appeal and professional design quality.*
>
> *For each chart:*
>
> *- View the image and its description*
>
> *- Answer 6 straightforward questions (1 for Fidelity, 2 for Expressiveness, 3 for Aesthetics)*
>
> *- Simply select your rating from 1 (Poor) to 5 (Excellent) – each option has clear descriptions to guide your choice*
>
> *We've designed this to minimize your effort while ensuring quality feedback. Please take your time to carefully consider each visualization before making your selections. Please consider the time commitment carefully and only proceed if you can provide thoughtful, quality responses. If you're not ready to participate seriously, feel free to skip this task. We will check for quality and may reject careless or random responses. Your thoughtful and careful feedback is important—thank you!*

As illustrated in Figure 12, the annotation interface presented one chart at a time, along with a textual description. Below the visualization, the six evaluation questions were displayed, customized based on the chart content. To proceed, participants were required to complete all six questions. At the end of each chart, annotators also rated their overall confidence in their answers and were optionally allowed to flag any uncertain responses via a free-text input box. This structured flow encouraged serious participation while enabling us to monitor annotation quality and filter out unreliable data.

### A.3.4 CROWDSOURCING QUALITY CONTROL AND CANDIDATE STRATEGY GENERATION

Ensuring the reliability of collected annotations requires a two-stage quality control design.

**Stage 1: Crowdsourcing Quality Control.** During the crowdsourcing phase, we embedded *validation checks* into the annotation interface to identify inattentive or careless responses. Specifically, a small number of chart-pair questions were designed where the superior or inferior chart was visually and functionally obvious. For instance, one pair compared a clean and readable pie chart against an overly cluttered line chart. Annotators failing such checks were highly likely to be engaging in random or inattentive behavior. These responses were flagged and either discarded or subjected to further scrutiny, thereby improving the reliability of the collected scores. Figure 13 illustrates examples of these validation questions.

**Stage 2: Candidate Strategy Generation for Expert Review.** In the expert adjudication stage, we further designed a systematic conflict identification and resolution mechanism based on representative studies in crowdsourcing quality control, statistical outlier detection, and multi-dimensional evaluation theory (Gadiraju et al., 2015; Rousseeuw & Leroy, 2003; Brennan, 1992). This mechanism provides algorithmic candidate strategies to serve as reference signals for expert review, without replacing expert judgment.

**High-Disagreement Sample Identification.** The system first calculates the standard deviation of initial scores for each sample across all three annotators. Samples with standard deviation $> 1.0$ are automatically identified as "high-disagreement samples" requiring further algorithmic analysis and expert attention.

**Algorithmic Candidate Strategy Generation.** For high-disagreement samples, the system generates three types of candidate resolution strategies:

- **Outlier Removal Strategy:** When two annotators' scores are close (absolute difference $\leq 2.0$) but the third annotator's score differs significantly from both (absolute difference $> 1.5$ from each), the system suggests removing the anomalous score and averaging the remaining two scores. This strategy addresses cases where one annotator may have misunderstood the task or made systematic errors.

- **Malicious Scoring Filter Strategy:** The system identifies and flags abnormal rating behaviors where annotators assign identical scores across all six evaluation dimensions. Such patterns are statistically unlikely for genuine evaluation and may indicate inattentive or gaming behavior. Flagged annotations undergo additional scrutiny or removal.

- **Sub-dimension Bias Correction Strategy:** To address potential systematic biases in specific evaluation dimensions, the system independently applies a dual-threshold mechanism (threshold = 2.0) to each of the six evaluation dimensions. When an annotator's score in any dimension deviates by more than 2.0 points from the other two annotators' average, the system flags this as a potential dimensional bias and suggests score normalization or expert review.

**Strategy Integration and Ranking.** All candidate strategies are processed through a score integration and ranking module that evaluates their statistical validity and consistency with the overall dataset distribution. The ranked strategies are then presented to the expert team as structured recommendations, along with confidence scores and rationale explanations.

The four complementary strategies mentioned include: (1) the *standard evaluation process*, which provides baseline scores; (2) *sub-dimension major deviation detection*, which highlights dimensional inconsistencies; (3) *malicious or abnormal score filtering*, which identifies problematic responses; and (4) *major deviation detection*, which flags overall inconsistencies.

Together, these two complementary mechanisms—validation checks during crowdsourcing and candidate strategies during expert adjudication—form a multi-layered quality control pipeline, ensuring that the final dataset reflects trustworthy and rigorously validated quality scores.

### A.3.5 Expert Interface and Annotation

To streamline post-crowdsourcing adjudication, we developed an expert review interface that aggregates all **3,090** tasks and presents them in a prioritized queue. Samples are ranked by their *score divergence* ($\sigma$, the standard deviation across candidate scores), from high to low, enabling experts to resolve highly contentious cases first.

For each task, the interface displays a chart preview, metadata (type, subtype, and modification status), and the task description, which provide essential context for assigning a fair and informed score. The interface also presents per-dimension evaluation scores together with the outputs of the four candidate strategies—namely the *standard evaluation process*, *sub-dimension major deviation detection*, *malicious or abnormal score filtering*, and *major deviation detection*. These auxiliary signals do not override expert judgment; rather, they support experts in detecting anomalies, validating consistency, and ultimately determining the most reasonable final score. A screenshot of the expert review interface is shown in Figure 14.

### A.4 Annotation Quality Analysis

To validate the reliability and consistency of our annotation process, we conducted comprehensive quality analysis across multiple dimensions of the collected annotations.

### A.4.1 Overall Annotation Quality Assessment

**Crowd-Crowd MAE Analysis:** We used the average rating from all crowd annotators as the baseline and calculated the MAE between each annotator and this baseline to measure consistency and noise levels among annotators. Results show that across the three visualization types, the internal MAE was lowest for dashboards (0.6441), followed by single views (0.6842), and highest for multi-views (0.6963). This indicates that under our task and experimental settings, annotators provided more concentrated ratings with higher consistency for dashboards, while showing relatively larger disagreements for multi-views.

**Crowd-Expert MAE Analysis:** We used expert ratings as the gold standard and calculated the MAE between each crowd annotator and the expert to assess the deviation between crowd annotations and expert standards. Results showed a consistent pattern: the crowd-expert MAE was lowest for dashboards (0.5451), slightly higher for single views (0.5534), and highest for multi-views (0.5854),

meaning crowd annotations were closest to expert ratings for dashboards and showed relatively larger deviations for multi-views.

Notably, the crowd-expert MAE was lower than the corresponding crowd-crowd MAE for all three visualization types. This indicates that the deviation between each annotator and the expert standard was actually smaller than the deviations among annotators themselves, suggesting that expert ratings have good representativeness and can effectively balance different annotator perspectives. For the 1–5 rating scale used in this study, our observed MAE values (crowd-crowd: 0.64–0.70, crowd-expert: 0.54–0.59) are comparable to acceptable levels reported in prior crowdsourced visualization studies (Lan & Liu, 2024) and subjective evaluation task studies (Li et al., 2025c), indicating that annotation quality falls within a reasonable range.

### A.4.2 ANNOTATION QUALITY ACROSS ANNOTATOR BACKGROUNDS

**Crowdsourcing Platform and Geographic Distribution:** Our crowdsourcing experiment was conducted through CloudResearch, a leading online participant recruitment platform. Since the CloudResearch platform primarily targets users in the United States and our experiment used English as the interface language, the majority of participants were from the U.S. In total, 603 participants were recruited for the annotation tasks. Specifically, the U.S. contributed the most participants with 535 individuals (88.7%), followed by Canada (30 participants, 5.0%), New Zealand (24 participants, 4.0%), the UK (11 participants, 1.8%), Ireland (2 participants, 0.3%), and Australia (1 participant, 0.2%). We acknowledge the limited geographic diversity and discuss this as a limitation in the paper. Building on the overall MAE analysis, we further grouped annotators by background information to examine quality and consistency across different backgrounds.

**Professional Background Analysis:** We divided annotators into professionals and non-professionals based on whether they work in data visualization or data analysis-related fields, with professionals accounting for 41% and non-professionals 59%. Results showed that the professional group had an average within-group MAE of 0.4245, while the non-professional group had 0.4881, a difference of approximately 0.0637. This indicates that under our task and experimental settings, the professional group provided more concentrated ratings with slightly higher within-group consistency than the non-professional group.

**Educational Background Analysis:** We divided annotators into undergraduate, master's, and doctoral groups, accounting for 70%, 23%, and 7% respectively. The undergraduate group had an average within-group MAE of 0.4825, the master's group 0.4762, and the doctoral group 0.3944. Overall, the graduate groups (especially doctoral) showed better within-group consistency than the undergraduate group, with more concentrated ratings. However, due to the relatively small sample size of the doctoral group, we take a cautious and conservative approach to interpreting this result in the paper.

Combining both professional background and education level dimensions, we can draw three main conclusions. First, MAE levels do differ somewhat across different background groups, with annotators having relevant professional backgrounds and higher education showing certain advantages in within-group consistency and alignment with expert standards. Second, the magnitude of these differences is relatively limited overall, and after appropriate quality control measures and result aggregation, the overall annotation results from non-professional or lower-education annotators still maintain good reliability. Finally, and most critically, we observed consistent error patterns across all background subgroups that matched the overall analysis (e.g., relative performance across different visualization types remained consistent). This indicates that our main conclusions about crowd-sourced annotation quality and effects under different visualization conditions are relatively robust across different annotator backgrounds.

### A.4.3 EXPERT QUALITY CONTROL IMPACT ANALYSIS

Experts reviewed all 3,090 samples individually. During the review process, approximately 11% of samples underwent adjustments at the overall evaluation level, with 7% using the Malicious Scoring Filter Strategy and 4% using the Outlier Removal Strategy—these strategies remove abnormal ratings before recalculating the base scores. Additionally, 16% of samples used the Sub-dimension Bias Correction Strategy, removing outlier ratings for certain sub-dimensions with large disagreements, particularly those where the standard deviation exceeded 1. Nearly 73% of samples retained

all original ratings and underwent further expert review based on these ratings. These results demonstrate that our quality control process is rigorous and reliable, while also indicating that the overall quality of crowd annotations is high, with the vast majority of samples not requiring quality control through removal of abnormal ratings.

### A.4.4 DIMENSION-SPECIFIC QUALITY ANALYSIS

We further analyzed expert adjustment patterns across different dimensions and found that adjustment frequency is closely related to the objective/subjective characteristics of dimensions.

**Overall Adjustment Patterns:** Among the 511 samples requiring sub-dimension adjustments, the most frequently adjusted dimensions were semantic readability (19.20%), data fidelity (18.04%), and insight discovery (16.79%), which have strong objective characteristics. Semantic readability concerns whether text and legends are clear and easy to read, data fidelity focuses on whether there is misleading information or data distortion, and insight discovery examines whether visualizations effectively convey key information. Because these aspects have relatively clear evaluation standards, experts tend to apply stricter control over these dimensions to ensure annotation accuracy, resulting in higher adjustment rates. In contrast, dimensions with lower adjustment frequencies include design style (17.05%), visual composition (14.73%), and color harmony (14.20%), which involve more aesthetic judgments with strong subjective characteristics. Different annotators may provide different ratings based on personal aesthetic preferences, and these differences often reflect the diverse perspectives of real user populations. Experts made fewer adjustments to these dimensions, indicating that our quality control process ensures annotation accuracy while respecting and preserving a certain degree of subjectivity and diversity, avoiding over-standardization of aesthetic criteria.

**Adjustment Characteristics Across Visualization Types:** Further analysis by visualization type revealed different adjustment patterns:

*Single Visualizations* (159 samples, 31.1%): Data fidelity was adjusted most frequently (20.06%), followed by semantic readability (18.58%), while aesthetics-related dimensions such as color harmony (14.75%), design style (15.34%), and insight discovery (15.04%) had relatively lower adjustment rates. This indicates that single visualizations are most sensitive to data truthfulness and semantic interpretation.

*Multiple Visualizations* (153 samples, 29.9%): Semantic readability had the highest adjustment rate (20.46%), followed by design style (19.02%) and data fidelity (18.73%), while color harmony had the lowest adjustment rate (11.24%). This suggests that multiple visualizations show the largest disagreements in semantic communication and overall design, but relatively fewer color coordination issues.

*Dashboard* (199 samples, 38.9%): Adjustment rates across dimensions were relatively balanced, with semantic readability (18.66%), insight discovery (18.20%), and data fidelity (17.05%) slightly higher, while aesthetic dimensions such as design style (16.82%), color harmony (16.13%), and visual composition (14.29%) had relatively lower adjustment rates. This indicates that dashboards show certain disagreements across all dimensions but are overall more balanced.

This finding validates the rationality of our quality control strategy: applying strict control over objectively verifiable dimensions such as data fidelity, semantic readability, and insight discovery, while maintaining appropriate tolerance for subjective aesthetic dimensions such as color harmony and visual composition, thereby achieving a balance between annotation consistency and perspective diversity. This balance is crucial for building a visualization quality assessment dataset that reflects the diverse preferences of real users.

In summary, our annotation process produced reliable and consistent evaluation data suitable for training and evaluating visualization quality assessment models. To facilitate reproducible research and further academic exploration, we will publicly release all annotation data, quality control records, and related statistical analysis results to enable the research community to verify and extend our work.

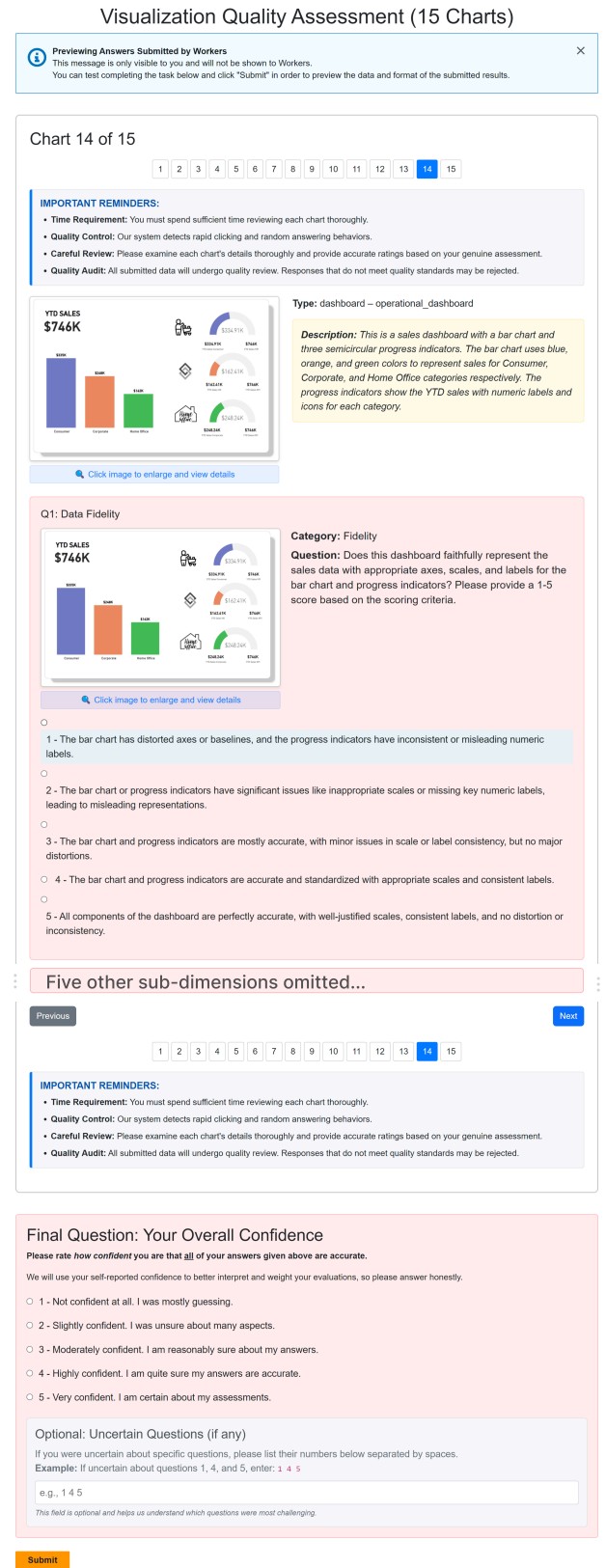

Figure 12: Crowdsourcing interface for expert annotation process.

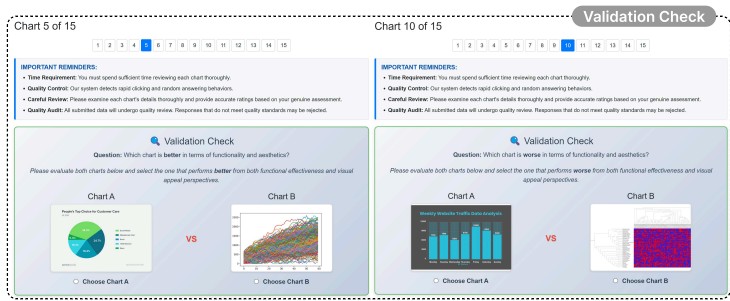

Figure 13: Examples of validation checks embedded in the crowdsourcing interface.

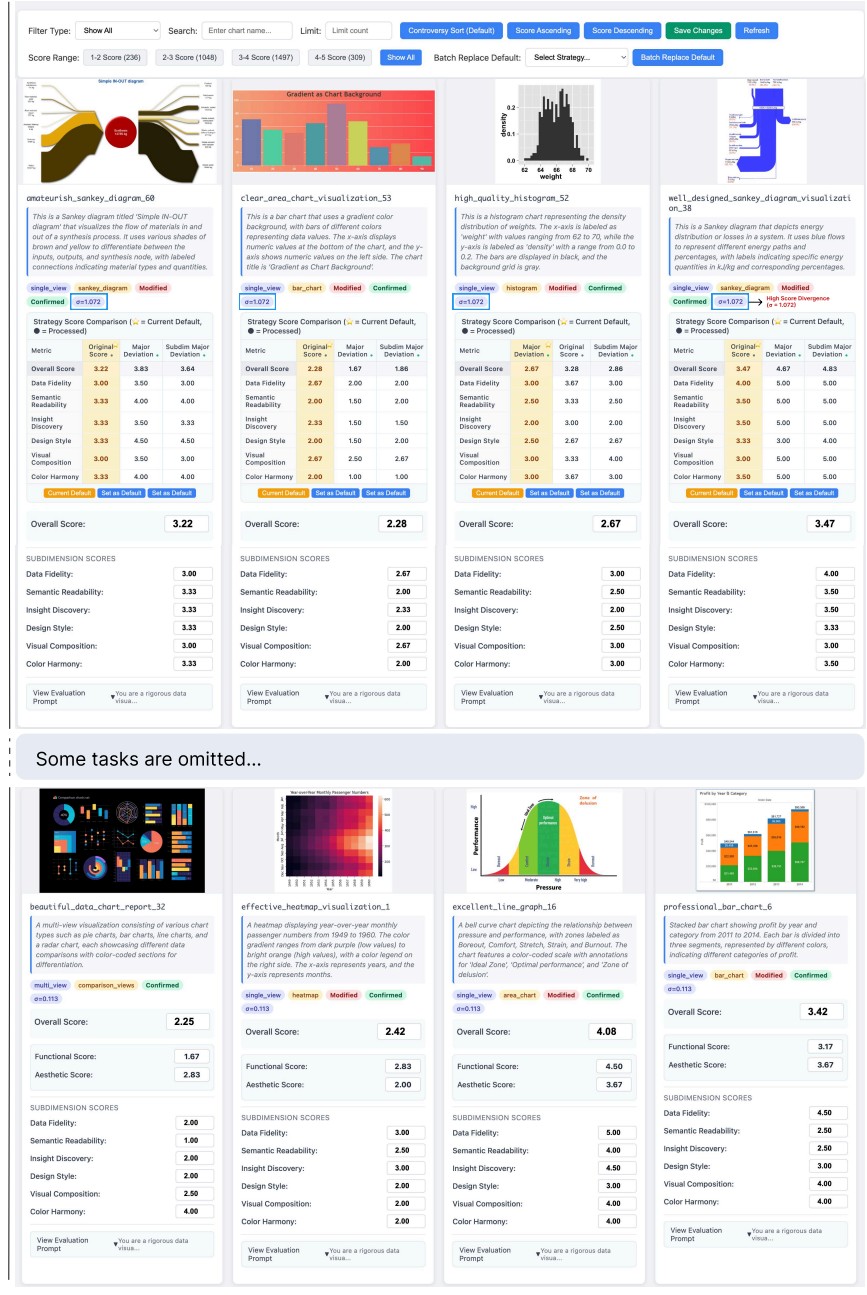

Figure 14: Expert review interface.

# B CASE STUDIES

## B.1 HIGH-SCORE CASE STUDIES: HUMAN-MODEL ALIGNMENT

We define **High-Score visualizations** as those that achieve strong performance across the three dimensions of fidelity, expressiveness, and aesthetics. Such visualizations demonstrate professional design principles and convey information in a manner that is both accurate and visually engaging.

As illustrated in Figure 15, we present a representative high-score case along with the corresponding human ratings and VISJUDGE output. This example demonstrates that, when the visualization adheres to established best practices, the model's evaluation is largely consistent with human judgment.

## B.2 MEDIUM-SCORE CASE STUDIES: HUMAN-MODEL ALIGNMENT

We define **Medium-Score visualizations** as those that perform adequately across fidelity, expressiveness, and aesthetics, but fall short of excellence in at least one of these dimensions. Such visualizations generally succeed in conveying information correctly and remain interpretable, yet they may exhibit shortcomings in specific aspects, most often in visual aesthetics or design refinement.

As shown in Figure 16, we present a representative medium-score case, where the visualization fulfills its communicative purpose but does not achieve high-quality standards in every dimension.

## B.3 LOW-SCORE CASE STUDIES: HUMAN-MODEL ALIGNMENT

We define **Low-Score visualizations** as those that exhibit clear deficiencies across one or more of the three dimensions of fidelity, expressiveness, and aesthetics. Such visualizations often distort or obscure the underlying data, employ ineffective or misleading encodings, or suffer from poor design choices that hinder interpretability.

As illustrated in Figure 17, we present a representative low-score case, where both human ratings and VISJUDGE outputs highlight significant problems that severely compromise the effectiveness of the visualization.

## B.4 DIMENSION-SPECIFIC CASE STUDIES: VALIDATING EVALUATION CRITERIA

To facilitate a clearer understanding of the three evaluation dimensions—fidelity, expressiveness, and aesthetics—we provide representative low-score case studies for each dimension. Specifically, Figure 18 illustrates a case with low fidelity, Figure 19 presents a case with low expressiveness, and Figure 20 shows a case with low aesthetics. These targeted examples highlight how deficiencies in individual dimensions manifest in practice and demonstrate that VISJUDGE's evaluations align with human ratings along the intended criteria.

## B.5 MODEL ERROR ANALYSIS CASES

While evaluating visualizations along the three dimensions of fidelity, expressiveness, and aesthetics, we observe that certain base models fail to correctly identify the deficiencies of low-quality charts and consequently assign them undeservedly high scores. To illustrate these issues, we present two complementary types of error analysis:

- **Overview Analysis:** Figure 21 provides a high-level overview of typical failure patterns across different models, summarizing where and how the evaluations deviate from expert judgment.
- **Detailed Case Studies:** Figures 22, Figure 23, Figure 24 and Figure 25 present representative detailed cases, each showing the full output of a model and highlighting the specific reasons for misalignment with human evaluations.

Together, these analyses reveal both the systematic error modes of existing models and the necessity of explicitly evaluating visualizations along the dimensions of fidelity, expressiveness, and aesthetics.

### B.6 Error Analysis of Multiple Visualizations and Dashboards

Our evaluation identifies a distinct negative correlation between visualization complexity and model performance. As visualization complexity increases, from single charts to multi-view compositions and dashboards, the efficacy of current multimodal models notably declines. To conduct an in-depth analysis of model error patterns in complex visualizations, we filtered from the test set those samples where at least one model's predicted score deviated substantially from the human annotation (absolute deviation > 1.0), yielding a subset of 139 multi-visualization sets and 94 dashboard cases. In this subset, quality scores on *visual composition* and *color harmony* drop sharply. In dashboards, this degradation extends beyond aesthetic metrics, significantly impacting *insight discovery* and *semantic readability*. Based on our analysis, we attribute this degradation to the lack of cross-view visual understanding. The models tend to process each view as an isolated visualization and rely heavily on local visual cues (e.g., grid alignment or color consistency) while failing to construct the cross-view semantic connections and analytical pathways characteristic of human expert interpretation.

**Visual Composition and Color Harmony Evaluation.** Regarding aesthetic dimensions, the models exhibit a problematic dual bias. They tend to overestimate layout regularity, assigning high scores as long as views are arranged in a standard grid with consistent styling, while disregarding redundancy or the absence of meaningful cross-view relationships (see Figure 6, left). For instance, in the multi-visualization subset, this positive bias is widespread: InternVL3, Claude-3.5-Sonnet, and Qwen2.5-VL rated *visual composition* higher than human experts in roughly 90% of the cases (89.2%–92.1%), inflating scores by over 1.1 points on average. This tendency persists in dashboards, where Claude-3.5-Sonnet continued to overestimate Color Harmony in 75.5% of cases, inflating the metric by 1.09 points. Simultaneously, they are overly strict about practical design trade-offs. The models tend to enforce strict encoding rules, penalizing visualizations that make necessary compromises, such as minor inconsistencies in color encoding or redundant legends (see Figure 7, right). Gemini-2.5-Pro, for example, strictly penalized *color harmony*, underestimating scores in 44 multi-visualization cases and 29 dashboard cases, with negative deviations nearing 1 point. This indicates that models perform surface-level pattern matching rather than comprehending the functional intent behind design decisions.

**Data Fidelity Evaluation.** In terms of *data fidelity*, we find that the models often confuse structural completeness with content validity. Charts containing only axes and frames, but no actual data marks, often receive high fidelity scores merely because they adhere to visualization design standards (Figure 25). This hallucination is pervasive across both open-source and proprietary models. In the multi-visualization subset, Qwen2.5-VL and InternVL3 overestimated *data fidelity* in 95.0% and 96.4% of cases, respectively. Proprietary models like GPT-4o and Claude-3.5-Sonnet followed a similar pattern, inflating fidelity scores in 87.8% of cases with an average increase exceeding 1.50 points. In contrast, simplified schematic views, such as those omitting detailed axis labels to emphasize trend comparison, are frequently deemed unreliable by GPT-5 and Gemini-2.5-Pro due to their deviation from standard forms. This suggests current models prioritize structural conformity to standard charts over the accurate conveyance of underlying data.

**Dashboard Insight Discovery and Semantic Readability Evaluation.** In dashboards, we observe that models tend to equate interface completeness with analytical depth. Dashboards featuring diverse chart types within a unified visual style often receive high *insight discovery* and *semantic readability* scores, even when lacking logical coherence or narrative structure (e.g., Figure 19, top). Qwen2.5-VL exhibited the most extreme positive bias in *semantic readability*, inflating scores in 90.4% of the dashboard cases with an average increase of 1.28. Conversely, complex and information-rich dashboards are frequently unfairly penalized by GPT-5 and Gemini-2.5-Pro, which misinterpret their high density as "visual clutter" or "data distortion". GPT-5 underestimated *semantic readability* in 33 cases (average decrease of 0.78 points), while Gemini-2.5-Pro showed an even stronger negative bias, lowering scores in 67 cases by an average of 1.15 points. This underscores a critical limitation: current models lack a robust understanding of narrative logic within visualization interfaces and struggle to distinguish between visually polished artifacts and functionally effective analytical tools.

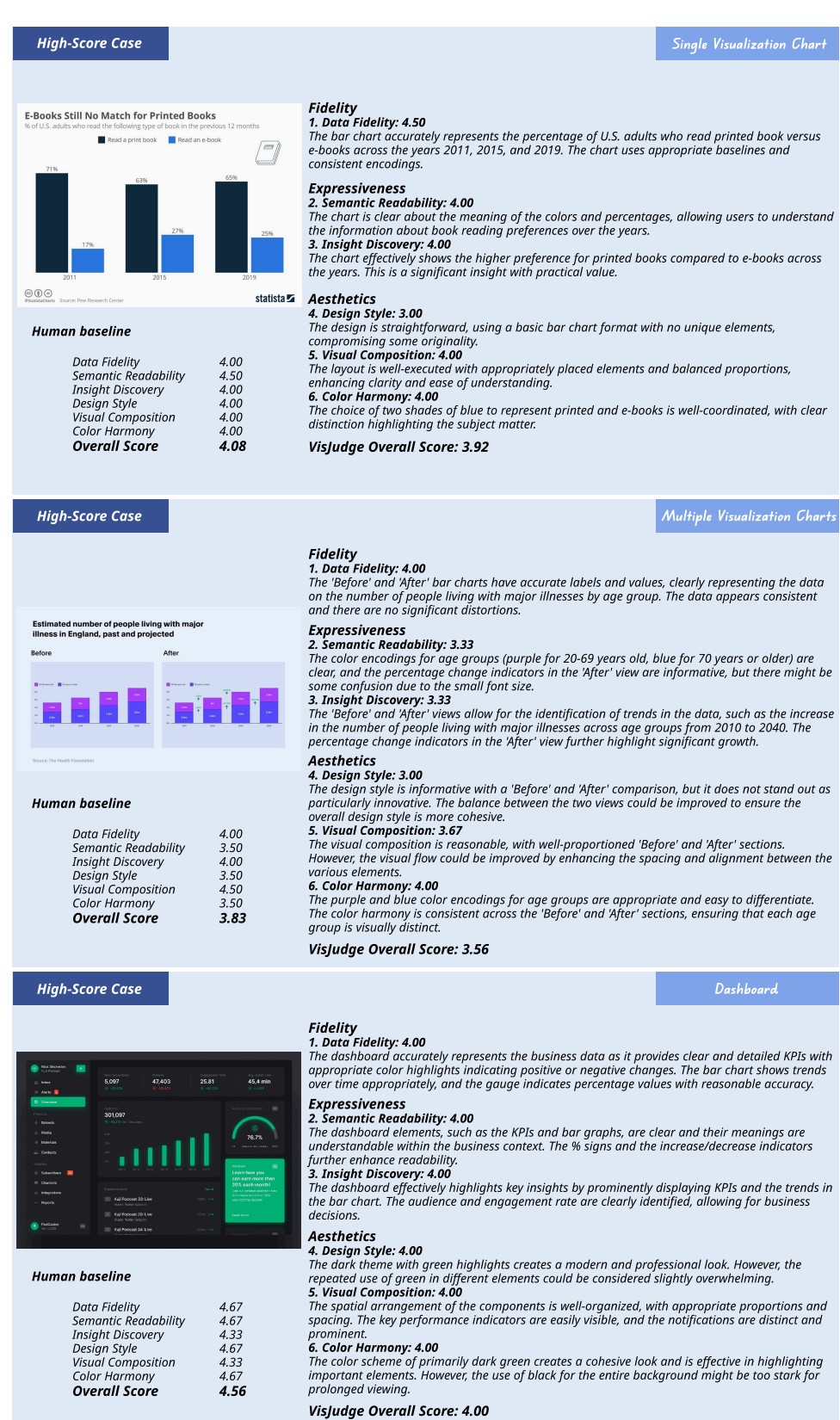

Figure 15: High score cases.

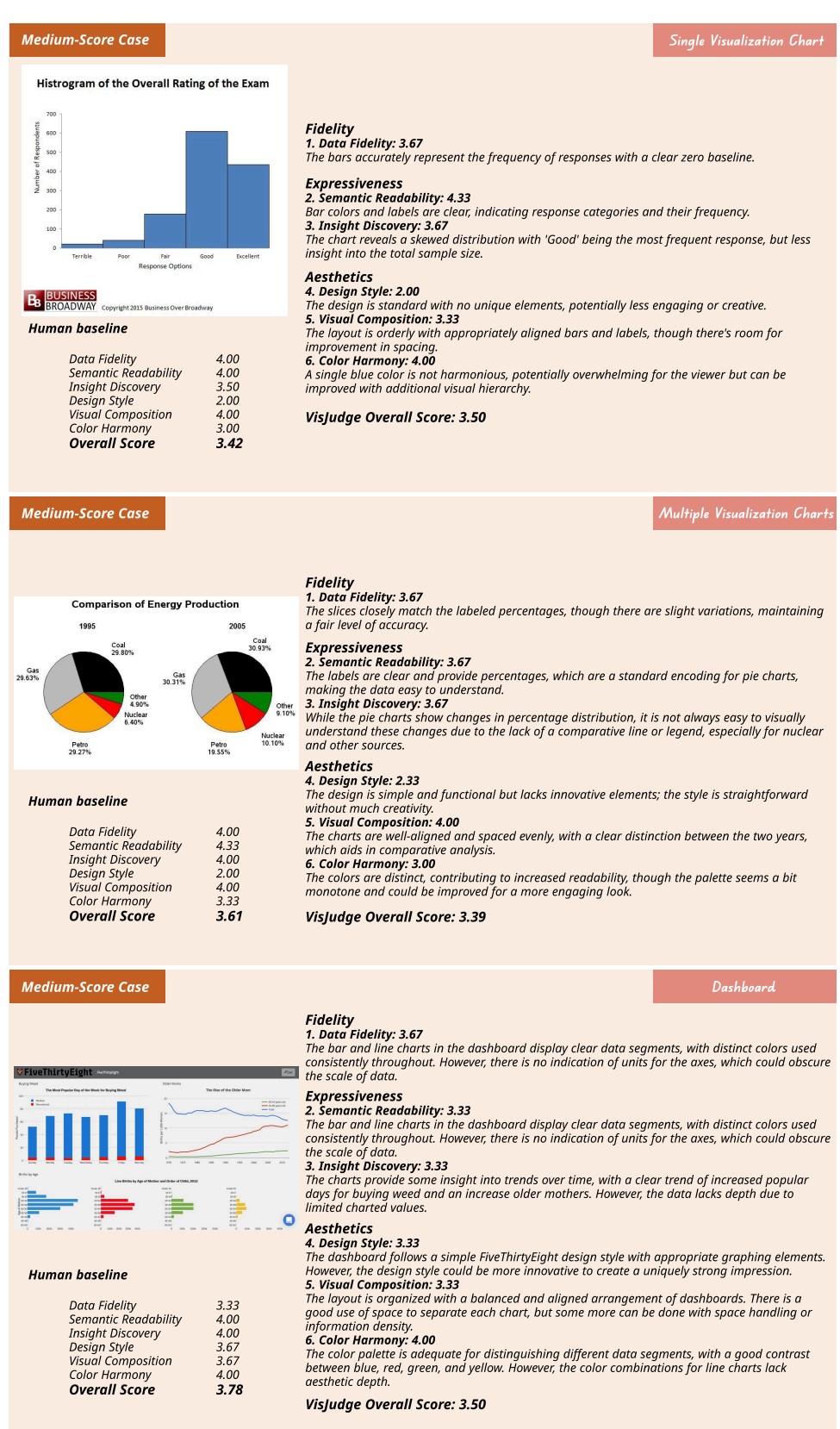

Figure 16: Medium score cases.

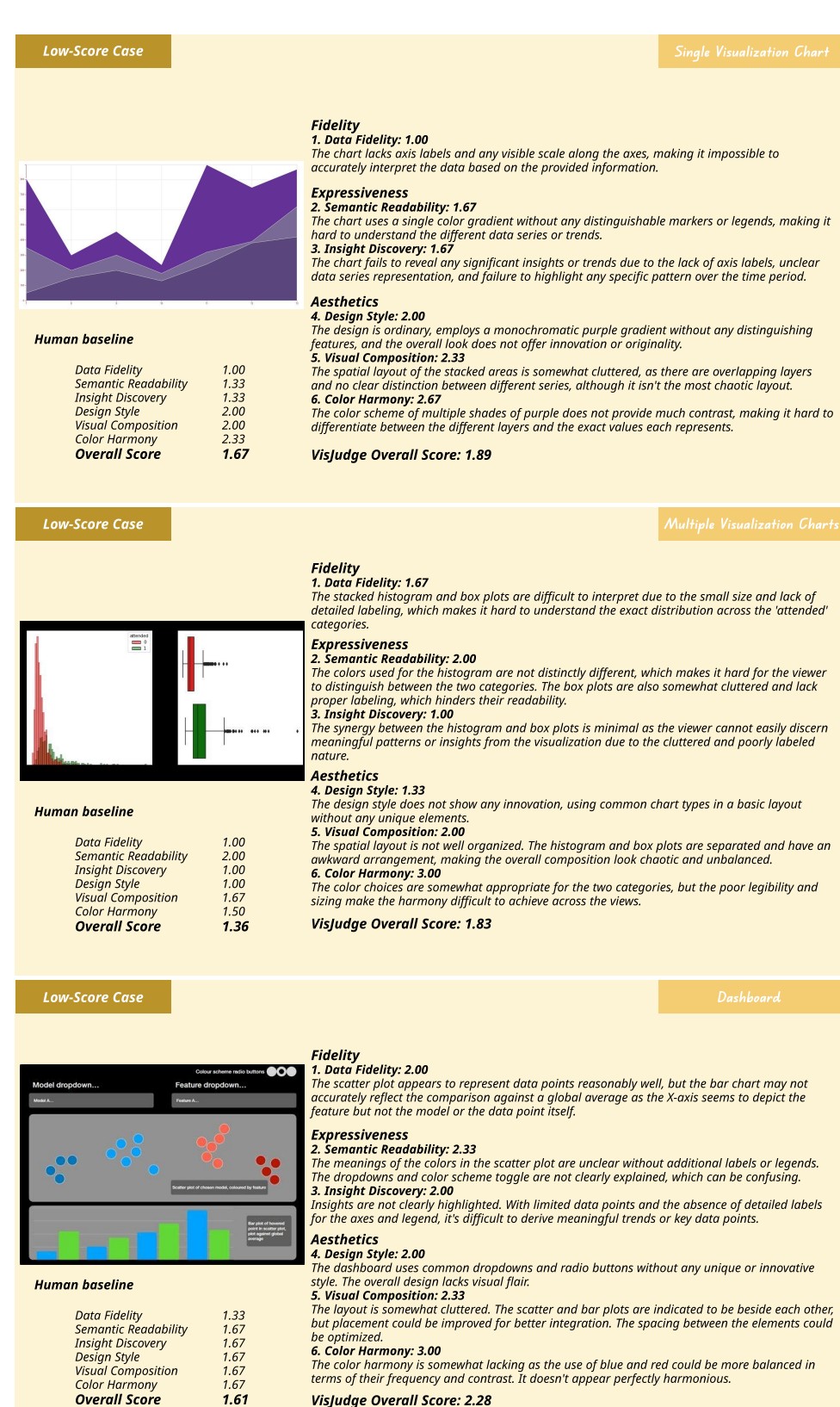

Figure 17: Low score cases.

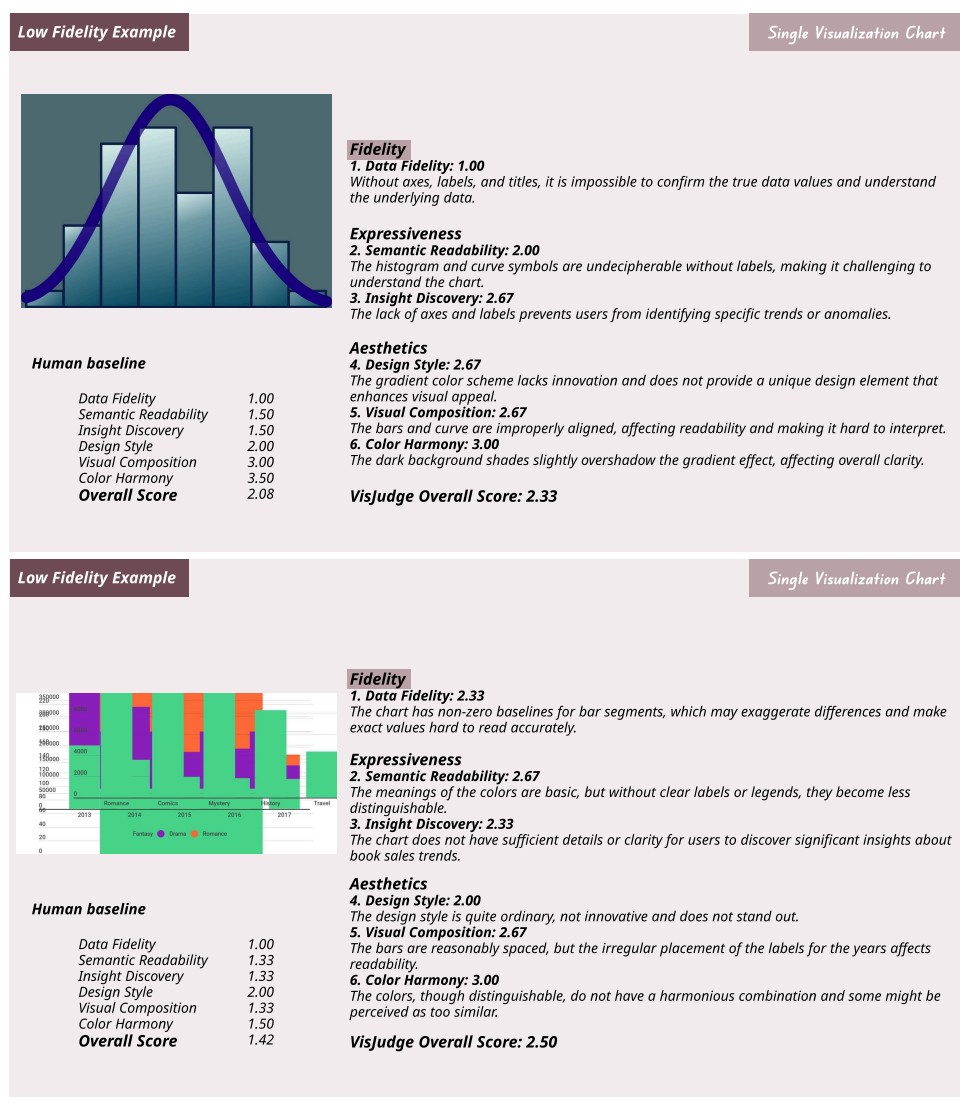

Figure 18: Low fidelity cases.

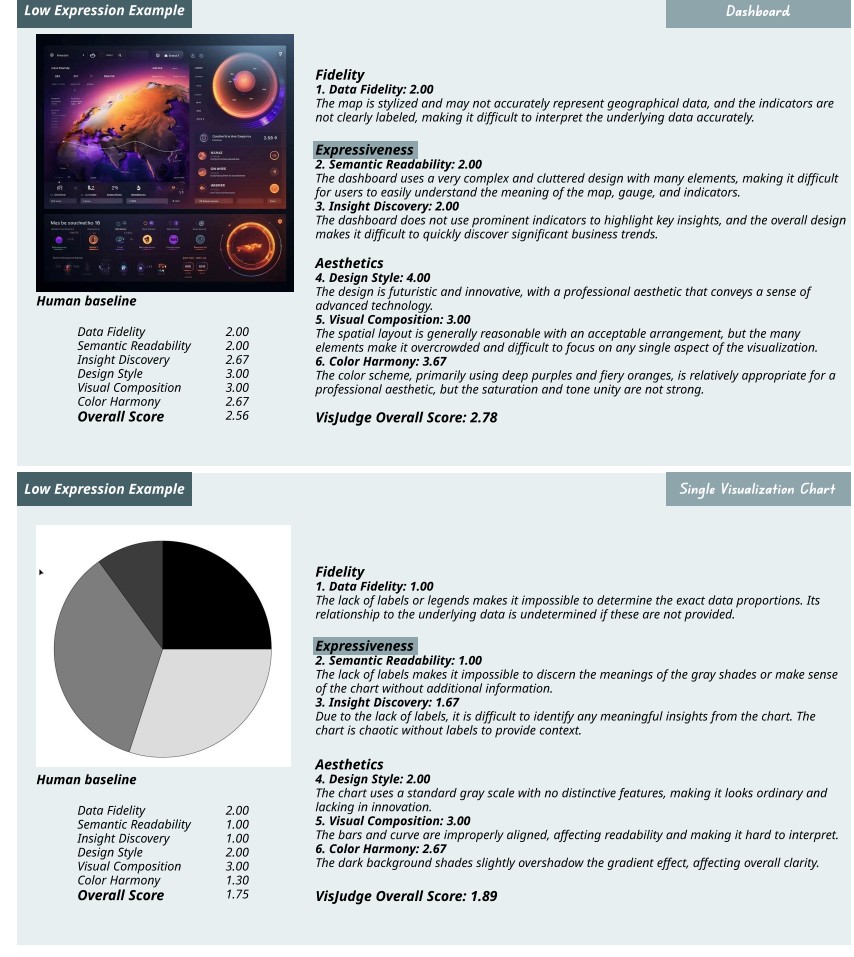

Figure 19: Low expressiveness cases.

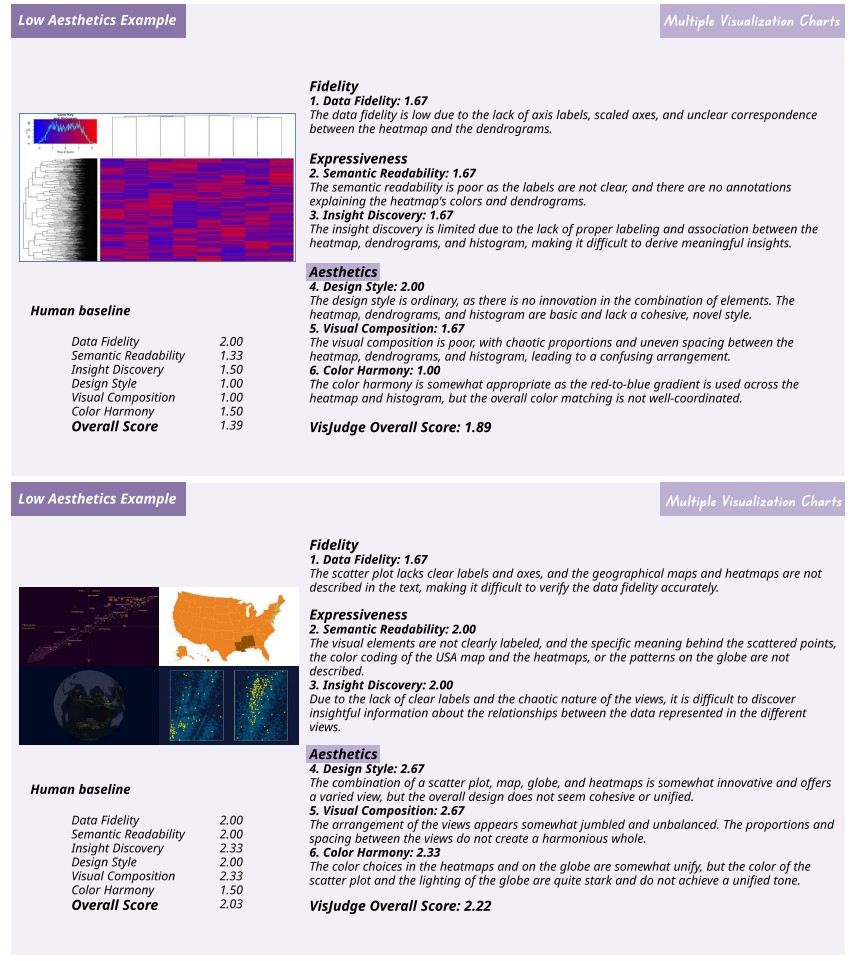

Figure 20: Low aesthetics cases.

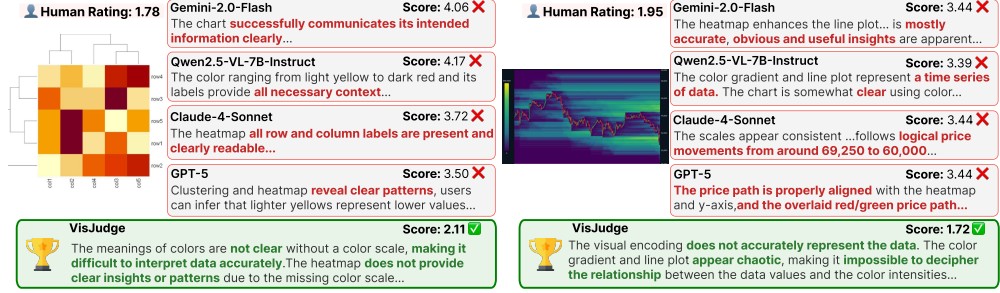

Figure 21: Overview of error cases.

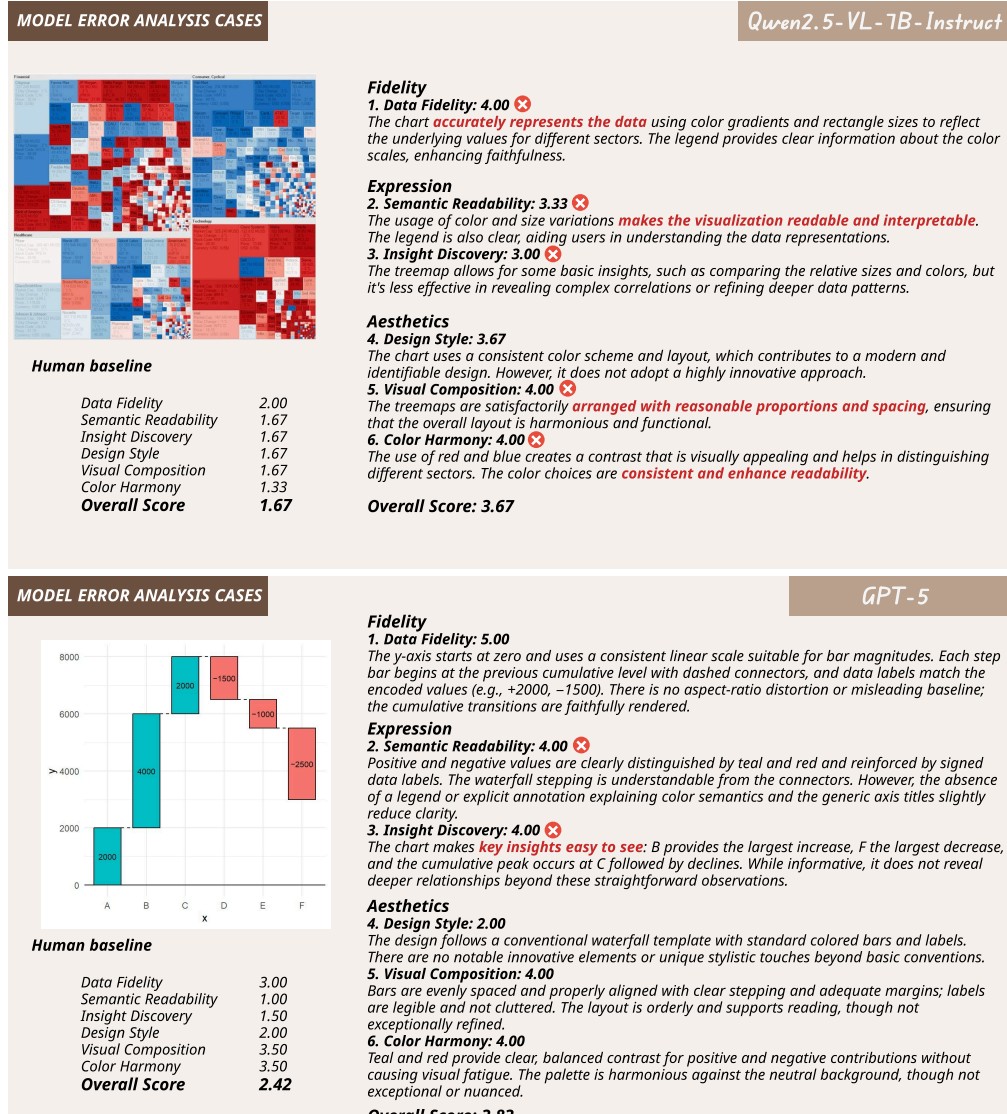

Figure 22: Qwen2.5-VL-7B-Instruct and GPT-5 error cases.

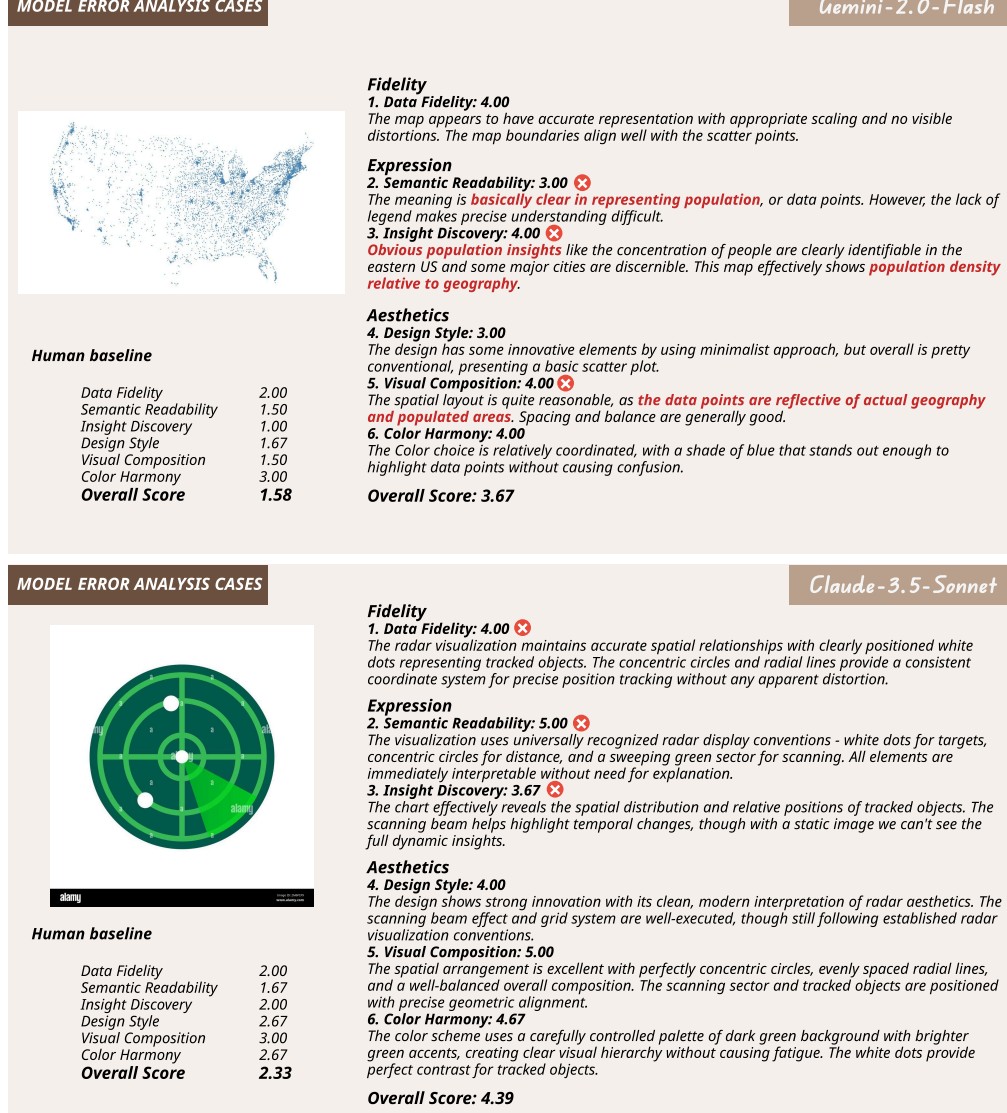

Figure 23: Gemini-2.0-Flash and Claude-3.5-Sonnet error cases.

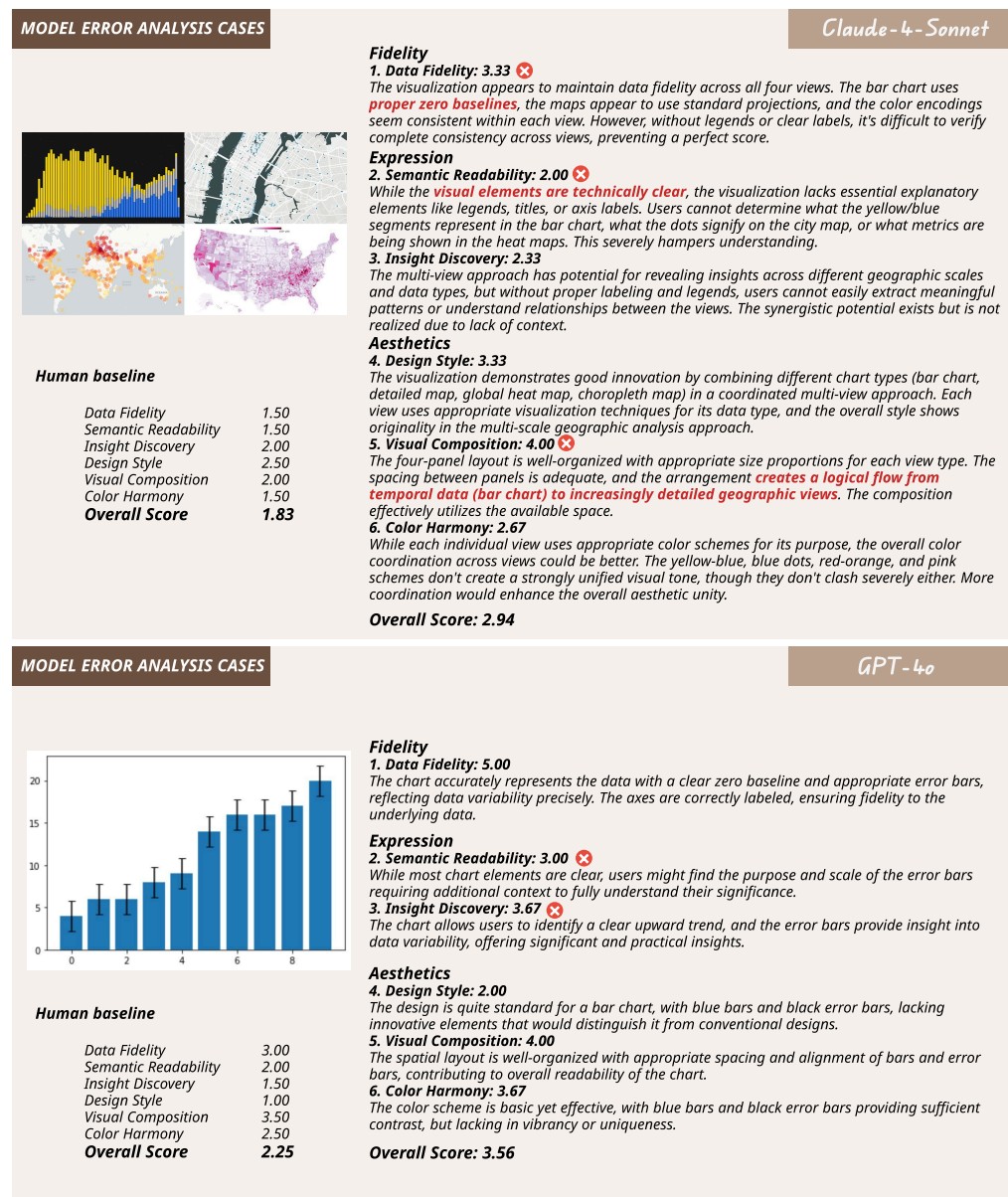

Figure 24: Claude-4-Sonnet and GPT-4o error cases.

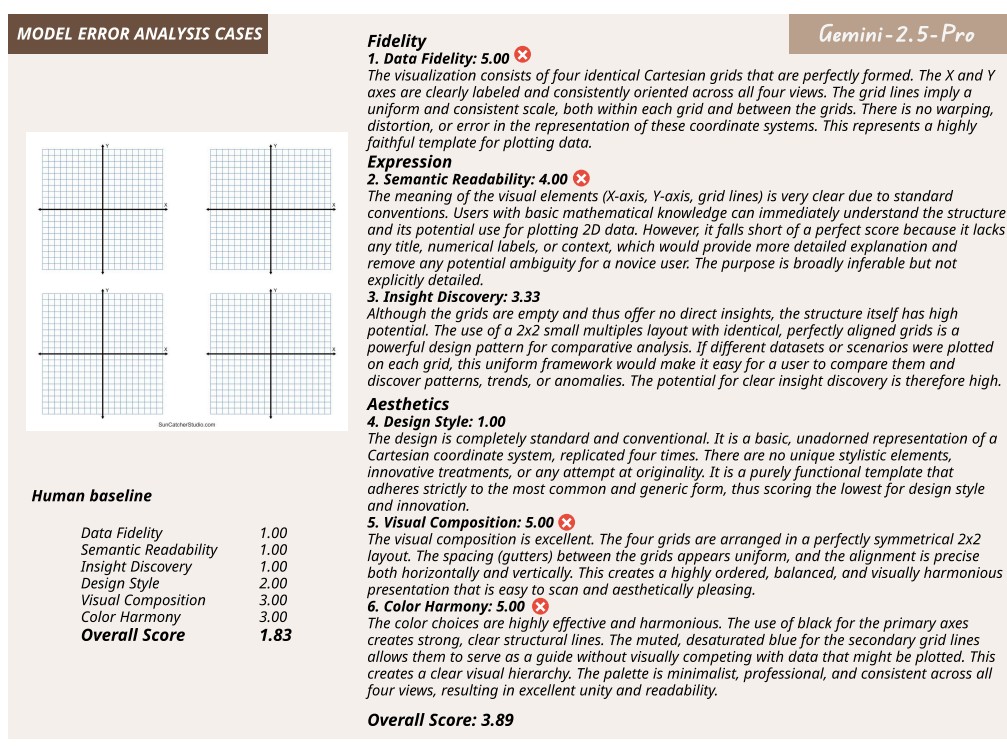

**MODEL ERROR ANALYSIS CASES**

*Gemini-2.5-Pro*

**Human baseline**

| | |
|---|---|
| Data Fidelity | 1.00 |
| Semantic Readability | 1.00 |
| Insight Discovery | 1.00 |
| Design Style | 2.00 |
| Visual Composition | 3.00 |
| Color Harmony | 3.00 |
| **Overall Score** | **1.83** |

**Fidelity**

**1. Data Fidelity: 5.00** ❌
The visualization consists of four identical Cartesian grids that are perfectly formed. The X and Y axes are clearly labeled and consistently oriented across all four views. The grid lines imply a uniform and consistent scale, both within each grid and between the grids. There is no warping, distortion, or error in the representation of these coordinate systems. This represents a highly faithful template for plotting data.

**Expression**

**2. Semantic Readability: 4.00** ❌
The meaning of the visual elements (X-axis, Y-axis, grid lines) is very clear due to standard conventions. Users with basic mathematical knowledge can immediately understand the structure and its potential use for plotting 2D data. However, it falls short of a perfect score because it lacks any title, numerical labels, or context, which would provide more detailed explanation and remove any potential ambiguity for a novice user. The purpose is broadly inferable but not explicitly detailed.

**3. Insight Discovery: 3.33**
Although the grids are empty and thus offer no direct insights, the structure itself has high potential. The use of a 2x2 small multiples layout with identical, perfectly aligned grids is a powerful design pattern for comparative analysis. If different datasets or scenarios were plotted on each grid, this uniform framework would make it easy for a user to compare them and discover patterns, trends, or anomalies. The potential for clear insight discovery is therefore high.

**Aesthetics**

**4. Design Style: 1.00**
The design is completely standard and conventional. It is a basic, unadorned representation of a Cartesian coordinate system, replicated four times. There are no unique stylistic elements, innovative treatments, or any attempt at originality. It is a purely functional template that adheres strictly to the most common and generic form, thus scoring the lowest for design style and innovation.

**5. Visual Composition: 5.00** ❌
The visual composition is excellent. The four grids are arranged in a perfectly symmetrical 2x2 layout. The spacing (gutters) between the grids appears uniform, and the alignment is precise both horizontally and vertically. This creates a highly ordered, balanced, and visually harmonious presentation that is easy to scan and aesthetically pleasing.

**6. Color Harmony: 5.00** ❌
The color choices are highly effective and harmonious. The use of black for the primary axes creates strong, clear structural lines. The muted, desaturated blue for the secondary grid lines allows them to serve as a guide without visually competing with data that might be plotted. This creates a clear visual hierarchy. The palette is minimalist, professional, and consistent across all four views, resulting in excellent unity and readability.

**Overall Score: 3.89**

Figure 25: Gemini-2.5-Pro error case.

## C EVALUATION FRAMEWORK AND CRITERIA

### C.1 DETAILED EVALUATION QUESTIONS AND SCORING CRITERIA

This appendix provides comprehensive details on the evaluation framework underlying VISJUDGE-BENCH, including specific evaluation questions and scoring criteria for each visualization type and evaluation dimension.

**Evaluation Framework Overview** Our evaluation framework is inspired by the classical *Fidelity, Expressiveness, and Aesthetics* principles, operationalized into six orthogonal sub-dimensions:

- **Fidelity:**
  - Data Fidelity: Evaluating visual-level data presentation accuracy and detecting misleading design patterns.
- **Expressiveness:**
  - Semantic Readability: Clarity of information communication.
  - Insight Discovery: Effectiveness in revealing meaningful patterns.
- **Aesthetics:**
  - Design Style: Innovation and uniqueness.
  - Visual Composition: Spatial layout and organization.
  - Color Harmony: Color coordination and visual appeal.

To support context-aware evaluation tailored to specific visualizations, we design a **evaluation questions and scoring criteria rewriting prompt template** that transforms generic evaluation criteria into more concrete versions. An example rewriting is shown in Figure 26, which illustrates a set of customized sub-dimension evaluation questions and scoring criteria tailored to a specific single visualization chart.

This rewriting process incorporates: `sub_dimension_name` (the sub-dimension name under three classical principles), `sub_dimension_text` (the standard evaluation question and scoring criteria definitions). The resulting prompt enables the generation of chart metadata and adapted evaluation rubrics that are grounded in the specific context of the visualization being evaluated.

---

**Prompt Template: Evaluation Questions and Scoring Criteria Rewriting**

You are a professional data visualization evaluation expert. You need to generate targeted evaluation questions and specific scoring criteria for each metric based on the provided visualization chart and {sub_dimension_name} evaluation metrics.

{sub_dimension_name} evaluation metrics information:
{sub_dimension_text}

**Task Requirements:**

1. Carefully observe the provided {image_type} type visualization chart
2. Based on the specific content of the chart (such as chart type, data content, visual elements, design features, etc.)
3. Combine the requirements and scoring criteria of each evaluation metric above
4. For each metric, generate:
   - A specific, targeted evaluation question
   - Custom scoring criteria (1–5 scale) that is specifically adapted to this chart's features

**Question Format Requirements:**

- Questions should clearly point out specific features of the chart (e.g., *"This bar chart uses blue and red contrast..."*)
- Questions should relate to the core points of the evaluation metric

---

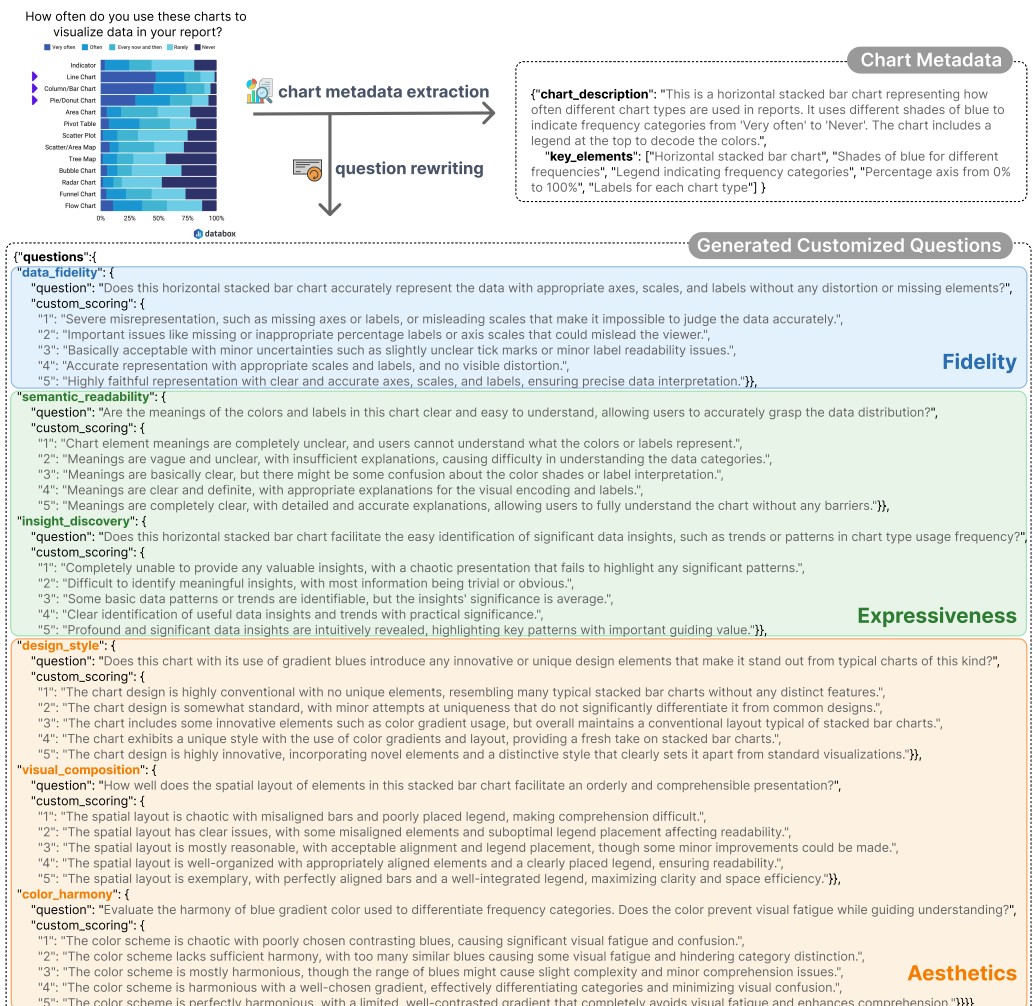

Figure 26: Rewriting result of customized evaluation questions and scoring criteria for a single visualization chart.

- Questions should guide evaluators to think about the specific performance of that metric
- Questions should end with: *"Please provide a 1–5 score based on the scoring criteria."*

**Scoring Criteria Format Requirements:**
- For each question, provide a custom 1–5 rating scale
- Each score description should specifically reference elements in **this chart**
- Score **1** should describe the worst-case scenario for this chart
- Score **5** should describe the ideal implementation for this chart
- Scores **2–4** should provide meaningful intermediate levels

**Return Format:** JSON object with the following structure:

```
{
  "chart_description": "Brief description of the chart (chart type,
    main features, etc.)",
  "key_elements": ["Observed element 1", "Observed element 2", "..."],
```

```
"questions": {
  "sub_dimension_name": {
    "question": "Specific question text...",
    "custom_scoring": {
      "1": "Score 1 description specific to this chart...",
      "2": "Score 2 description...",
      "3": "Score 3 description...",
      "4": "Score 4 description...",
      "5": "Score 5 description..."
    }
  },
  ...
}
}
```

### C.1.1 SINGLE VISUALIZATION EVALUATION CRITERIA

**Data fidelity evaluation questions and scoring criteria:** The following prompt is designed to guide the generation of scoring rubrics for the *Data Fidelity* dimension, using a 1–5 scale. It focuses on ensuring that visual representations truthfully and accurately reflect the underlying data.

**Prompt: Single Visualization - Data Fidelity Evaluation Questions and Scoring Criteria (1–5 Scale)**

**Description:**
Evaluates whether the visual encodings faithfully represent the data without misleading distortions. Focus on: appropriate axes and baselines (e.g., zero baseline for bar charts), reasonable scale ranges and tick intervals, consistency between encodings (position/length/angle/area/color) and numeric labels, and absence of aspect-ratio deformation, cropping/stretching, 3D distortion, or improper broken axes.

**Scoring criteria:**

**S 1.** Severe misrepresentation or impossible to judge due to missing/unreadable key elements (titles, legends, axes/ticks/labels, units, data labels, baselines, etc.), or clear distortions (e.g., non-zero baseline in bars exaggerating differences, 3D effects causing misread, obvious aspect-ratio deformation).

**S 2.** Important issues likely to mislead: inappropriate baseline/scale range, inconsistent encoding vs labels, selective ranges that exaggerate differences, or partial key elements missing causing notable uncertainty.

**S 3.** Basically acceptable with minor uncertainties: some elements may be missing or hard to read but no obvious distortion; scales and encodings are mostly reasonable and roughly consistent with labels.

**S 4.** Accurate and standardized: axes/baselines are appropriate, scale ranges and ticks are reasonable, encodings match labels, and no visible distortion or manipulation is present.

**S 5.** Highly faithful representation: all relevant elements are present and clear; axes/baselines and scale choices are well-justified (e.g., proper zero baseline for bars, appropriate linear/log choice), encodings and labels are fully consistent, with no cropping, deformation, or 3D misuse.

**Semantic readability evaluation questions and scoring criteria:** The following prompt guides the generation of scoring rubrics for the *Semantic Readability* dimension. It evaluates how clearly the chart communicates information through its visual encodings and annotations.

**Prompt: Single Visualization - Semantic Readability Evaluation Questions and Scoring Criteria (1–5 Scale)**

**Description:**
Based on complete and clear chart elements, evaluates whether users can understand the meaning of these elements and the information the chart conveys, including clarity of visual encoding (whether meanings of colors, shapes, sizes are clear) and clarity of information communication (whether users can accurately understand chart content).

**Scoring criteria:**

- **S 1.** Chart element meanings are completely unclear, cannot understand what visual encoding represents, users cannot obtain meaningful information.
- **S 2.** Chart element meanings are vague and unclear, insufficient visual encoding explanations, users have difficulty understanding and need extensive guessing to understand chart content.
- **S 3.** Chart element meanings are basically clear, users can understand main information, but still have understanding barriers in some encoding or labeling aspects.
- **S 4.** Chart element meanings are clear and definite, visual encoding has appropriate explanations, users can smoothly understand the information conveyed by the chart.
- **S 5.** Chart element meanings are completely clear, visual encoding explanations are detailed and accurate, users can understand all chart information without any barriers.

**Insight discovery evaluation questions and scoring criteria:** The following prompt guides the generation of scoring rubrics for the *Insight Discovery* dimension. It assesses the chart's ability to reveal meaningful patterns, trends, or non-obvious findings.

**Prompt: Single Visualization - Insight Discovery Evaluation Questions and Scoring Criteria (1–5 Scale)**

**Description:**
Evaluates whether the chart can easily identify significant and meaningful data insights, trends, patterns, or anomalies that must have actual value and guiding significance for users, rather than trivial or obvious general information.

**Scoring criteria:**

- **S 1.** Completely unable to provide any valuable insights, chaotic information presentation, no significant or meaningful discoveries, insight delivery completely failed.
- **S 2.** Difficult to identify meaningful insights, most presented information is trivial or obvious, lacking practical value, insight discovery difficult.
- **S 3.** Can identify some basic data patterns or trends, but the significance and practicality of insights are average, with limited guiding value for users.
- **S 4.** Can clearly identify obvious and useful data insights and trends, discovered patterns or anomalies have certain practical significance and reference value.
- **S 5.** Very intuitively reveals profound and highly significant data insights, can discover significant trend changes, important anomalies, or key patterns that have important guiding value for decision-making or understanding.

**Design style evaluation questions and scoring criteria:** The following prompt guides the generation of scoring rubrics for the *Design Style* dimension. It reflects the level of visual creativity, uniqueness, and design innovation present in the chart.

**Prompt: Single Visualization - Design Style Evaluation Questions and Scoring Criteria (1–5 Scale)**

**Description:**
Evaluates whether the chart design has innovation and uniqueness, whether it can stand out from many visualization works through novel design elements and unique style characteristics.

**Scoring criteria:**

- **S 1.** Design completely lacks innovation, style is outdated or shows obvious imitation, no uniqueness or originality.
- **S 2.** Design lacks innovation, style is quite ordinary, basically uses common design techniques, insufficient uniqueness.
- **S 3.** Design has some innovative elements, but overall is relatively conventional, uniqueness not prominent enough, innovation level is average.
- **S 4.** Design has strong innovation, style is quite unique, has novel design elements, with certain originality.
- **S 5.** Design is extremely innovative, style is unique and novel, uses innovative design elements or expression techniques, with strong originality and recognizability.

**Visual composition evaluation questions and scoring criteria:** The following prompt guides the generation of scoring rubrics for the *Visual Composition* dimension. It focuses on the spatial arrangement and organization of visual elements for effective communication.

**Prompt: Single Visualization - Visual Composition Evaluation Questions and Scoring Criteria (1–5 Scale)**

**Description:**
Evaluates whether the spatial layout of chart elements is reasonable, including whether element positions, size proportions, spacing distribution and other spatial relationships are appropriately and orderly arranged.

**Scoring criteria:**

- **S 1.** Spatial layout is seriously unreasonable, element positions are chaotic, proportions severely unbalanced, spacing distribution is disorderly.
- **S 2.** Spatial layout has obvious unreasonable aspects, improper element positions, unbalanced proportions, uneven spacing distribution.
- **S 3.** Spatial layout is basically reasonable, element positions are acceptable, proportional relationships are basically coordinated, but may have small layout issues.
- **S 4.** Spatial layout is quite reasonable, element positions are appropriate, good proportional relationships, reasonable spacing distribution, proper space utilization.
- **S 5.** Spatial layout is very reasonable, element positions are perfect, proportions are coordinated, spacing distribution is even, space utilization efficiency is extremely high.

**Color harmony evaluation questions and scoring criteria:** The following prompt guides the generation of scoring rubrics for the *Color Harmony* dimension. It evaluates how effectively the color scheme supports readability, aesthetic appeal, and visual coherence.

**Prompt: Single Visualization - Color Harmony Evaluation Questions and Scoring Criteria (1–5 Scale)**

**Description:**
Evaluates whether chart color choices are coordinated, contrast is appropriate, color quantity is reasonable, and whether it avoids too many colors causing visual fatigue.

**Scoring criteria:**

**S 1.** Color choices are chaotic, extensive use of uncoordinated colors, serious visual fatigue, completely unable to effectively guide user understanding, color use severely affects information delivery.

**S 2.** Color choices are not coordinated enough, using too many colors (5 or more), inappropriate contrast, causing obvious visual fatigue and understanding difficulties.

**S 3.** Color choices are basically coordinated, but color quantity is excessive (4–5 colors), may cause slight visual complexity, somewhat affecting understanding.

**S 4.** Color choices are relatively coordinated, appropriate contrast, reasonable color quantity control (3–4 colors), basically avoiding visual confusion, well guiding user understanding.

**S 5.** Color choices are very coordinated, appropriate contrast, limited color scheme (usually 1–3 main colors), colors are simple and unified, completely avoiding visual fatigue, effectively maintaining user attention focus.

### C.1.2 MULTIPLE VISUALIZATION EVALUATION CRITERIA

**Data fidelity evaluation questions and scoring criteria:** The following prompt is designed to guide the generation of scoring rubrics for the *Data Fidelity* dimension, using a 1–5 scale. It focuses on ensuring that visual representations truthfully and accurately reflect the underlying data.

**Prompt: Multiple Visualization - Data Fidelity Evaluation Questions and Scoring Criteria (1–5 Scale)**

**Description:**
Evaluates whether multiple coordinated views faithfully represent data without distortion individually and in combination. Check appropriate axes/baselines, reasonable scale ranges, consistency between encodings and labels, and cross-view consistency (e.g., comparable scales where appropriate), avoiding aspect-ratio deformation, cropping, 3D misuse, or improper broken axes.

**Scoring criteria:**

**S 1.** Severe misrepresentation or impossible to judge due to missing/unreadable key elements in one or more views, or clear distortions (non-zero bar baseline where required, 3D effects, obvious deformation), leading to unreliable reading.

**S 2.** Important issues exist in one or more views (inappropriate baseline/scale, inconsistent encoding vs labels, selective ranges), or partial key elements missing causing notable cross-view uncertainty.

**S 3.** Basically acceptable overall: minor uncertainties or small issues in some views, but no obvious distortion; scales and encodings are mostly reasonable across views.

**S 4.** Accurate and standardized across views: appropriate axes/baselines and scales; encodings match labels; no visible distortion; reasonable cross-view scale alignment where needed.

**S 5.** Highly faithful multi-vis representation: all views present complete, clear elements; well-justified axes/baselines and scale choices; full consistency between encodings and labels; no cropping/deformation/3D misuse; coherent cross-view comparability.

**Semantic readability evaluation questions and scoring criteria:** The following prompt guides the generation of scoring rubrics for the *Semantic Readability* dimension. It evaluates how clearly the chart communicates information through its visual encodings and annotations.

**Prompt: Multiple Visualization - Semantic Readability Evaluation Questions and Scoring Criteria (1–5 Scale)**

**Description:**
Based on complete and clear chart elements, evaluates whether users can understand the meaning of these elements and the information the chart conveys, including clarity of visual encoding (whether meanings of colors, shapes, sizes are clear) and clarity of information communication (whether users can accurately understand chart content).

**Scoring criteria:**

**S 1.** Chart element meanings are completely unclear, cannot understand what visual encoding represents, users cannot obtain meaningful information.

**S 2.** Chart element meanings are vague and unclear, insufficient visual encoding explanations, users have difficulty understanding and need extensive guessing to understand chart content.

**S 3.** Chart element meanings are basically clear, users can understand main information, but still have understanding barriers in some encoding or labeling aspects.

**S 4.** Chart element meanings are clear and definite, visual encoding has appropriate explanations, users can smoothly understand the information conveyed by the chart.

**S 5.** Chart element meanings are completely clear, visual encoding explanations are detailed and accurate, users can understand all chart information without any barriers.

**Insight discovery evaluation questions and scoring criteria:** The following prompt guides the generation of scoring rubrics for the *Insight Discovery* dimension. It assesses the chart's ability to reveal meaningful patterns, trends, or non-obvious findings.

**Prompt: Multiple Visualization - Insight Discovery Evaluation Questions and Scoring Criteria (1–5 Scale)**

**Description:**
Evaluates whether the entire multi-vis visualization can easily identify significant and meaningful data insights, and whether multiple views synergize to reveal deeper and more valuable comprehensive insights than single views.

**Scoring criteria:**

**S 1.** Completely unable to provide any valuable comprehensive insights, lack of coordination between views, no significant or meaningful discoveries, insight delivery completely failed.

**S 2.** Difficult to identify meaningful comprehensive insights, most information presented by multiple views is trivial or independent, lacking synergistic value, insight discovery difficult.

**S 3.** Can identify some basic multi-vis insights, but the significance and practicality of insights are average, limited cooperation between views, guiding value not prominent enough.

**S 4.** Can clearly identify obvious and useful multi-dimensional insights, good cooperation between views, discovered comprehensive patterns have certain practical significance and reference value.

**S 5.** Very intuitively reveals profound and highly significant comprehensive insights, perfect synergy between multiple views, can discover significant cross-dimensional patterns, important correlations, or key anomalies with important guiding value.

**Design style evaluation questions and scoring criteria:** The following prompt guides the generation of scoring rubrics for the *Design Style* dimension. It reflects the level of visual creativity, uniqueness, and design innovation present in the chart.

---

**Prompt: Multiple Visualization - Design Style Evaluation Questions and Scoring Criteria (1–5 Scale)**

**Description:**
Evaluates whether the overall design of multi-vis visualization has innovation and uniqueness, whether it can stand out from many visualization works through novel design elements, unique style characteristics, and coordinated style language.

**Scoring criteria:**

**S 1.** Multi-vis design completely lacks innovation, style is outdated or views have chaotic styles, no uniqueness or originality.

**S 2.** Multi-vis design lacks innovation, style is quite ordinary, basically uses common design techniques, insufficient uniqueness.

**S 3.** Multi-vis design has some innovative elements, but overall is relatively conventional, style characteristics not prominent enough, innovation level is average.

**S 4.** Multi-vis design has strong innovation, style is quite unique, coordinated style among views with certain originality.

**S 5.** Multi-vis design is extremely innovative, style is unique and novel, views maintain unified style while showing strong originality and recognizability.

---

**Visual composition evaluation questions and scoring criteria:** The following prompt guides the generation of scoring rubrics for the *Visual Composition* dimension. It focuses on the spatial arrangement and organization of visual elements for effective communication.

---

**Prompt: Multiple Visualization - Visual Composition Evaluation Questions and Scoring Criteria (1–5 Scale)**

**Description:**
Evaluates whether the spatial layout of individual sub-views and the overall arrangement of multiple views are reasonable, including element composition within individual views, size proportions between multiple views, arrangement methods, spacing distribution, etc., forming a harmonious and orderly visual whole.

**Scoring criteria:**

**S 1.** Sub-view layouts are chaotic or overall multi-vis arrangement is seriously unreasonable, size proportions severely unbalanced, spacing distribution is disorderly.

**S 2.** Sub-view layouts or multi-vis arrangement have obvious unreasonable aspects, inappropriate size proportions, uneven spacing distribution.

**S 3.** Sub-view layouts are basically reasonable, overall multi-vis arrangement is acceptable, but may have small issues in proportional relationships or spacing handling.

**S 4.** Sub-view layouts are good, overall multi-vis arrangement is reasonable, appropriate size proportions, proper spacing distribution, good space utilization.

**S 5.** Sub-view layouts are perfect, overall multi-vis arrangement is extremely reasonable, coordinated size proportions, even spacing distribution, extremely high space utilization efficiency.

---

**Color harmony evaluation questions and scoring criteria:** The following prompt guides the generation of scoring rubrics for the *Color Harmony* dimension. It evaluates how effectively the color scheme supports readability, aesthetic appeal, and visual coherence.

---

**Prompt: Multiple Visualization - Color Harmony Evaluation Questions and Scoring Criteria (1–5 Scale)**

**Description:**
Evaluates whether color choices in multi-vis are appropriate, including color combinations within each sub-view, coordination of color schemes between multiple views, unity of overall tone, and appropriate control of color quantity and saturation.

**Scoring criteria:**

**S 1.** Sub-view color choices are very inappropriate, overall color matching among multiple views is chaotic, tone conflicts or color use is seriously inappropriate.

**S 2.** Sub-view color choices are not appropriate enough, overall color matching among multiple views is not coordinated enough, tone not unified or color use inappropriate.

**S 3.** Sub-view color choices are basically appropriate, overall color matching among multiple views is acceptable, but may have small issues in tone unity or saturation.

**S 4.** Sub-view color choices are relatively appropriate, overall color schemes among multiple views are quite coordinated, tone basically unified, appropriate color use.

**S 5.** Sub-view color choices are very appropriate, overall color schemes among multiple views are highly coordinated and unified, perfect tone matching, appropriate control of color quantity and saturation.

---

### C.1.3 DASHBOARD EVALUATION CRITERIA

**Data fidelity evaluation questions and scoring criteria:** The following prompt is designed to guide the generation of scoring rubrics for the *Data Fidelity* dimension, using a 1–5 scale. It focuses on ensuring that visual representations truthfully and accurately reflect the underlying data.

---

**Prompt: Dashboard - Data Fidelity Evaluation Questions and Scoring Criteria (1–5 Scale)**

**Description:**
Evaluates whether the dashboard faithfully represents business data in each component and in the overall composition. Focus on appropriate axes/baselines, reasonable scale ranges, consistency between encodings and numeric labels, avoidance of aspect-ratio deformation, cropping/stretching, 3D distortion, or improper broken axes; also consider consistency of comparable metrics across components.

**Scoring criteria:**

**S 1.** Severe misrepresentation or impossible to judge due to missing/unreadable key elements in important components, or clear distortions (e.g., improper bar baselines, distorted 3D, obvious deformation), undermining data fidelity.

**S 2.** Important issues likely to mislead in one or more components: inappropriate baselines/scales, inconsistent encoding vs labels, selective ranges, or missing key elements causing notable uncertainty at the dashboard level.

**S 3.** Basically acceptable overall: minor issues or uncertainties in parts, but no obvious distortion; most components use reasonable scales/encodings consistent with labels.

**S 4.** Accurate and standardized: components and overall dashboard use appropriate axes/baselines and reasonable scales; encodings match labels; no visible distortion.

**S 5.** Highly faithful: all components present complete and clear elements; axes/baselines and scales are well-justified; encodings and labels fully consistent; no cropping/deformation/3D misuse; consistent comparability across related components.

---

**Semantic readability evaluation questions and scoring criteria:** The following prompt guides the generation of scoring rubrics for the *Semantic Readability* dimension. It evaluates how clearly the chart communicates information through its visual encodings and annotations.

---

**Prompt: Dashboard - Semantic Readability Evaluation Questions and Scoring Criteria (1–5 Scale)**

**Description:**
Based on complete and clear dashboard elements, evaluates whether users can understand the meaning of these elements and the information the dashboard conveys, including clarity of visual encoding (whether meanings of colors, shapes, sizes, gauge pointers, status indicators are clear) and clarity of information communication (whether users can accurately understand various parts and overall business meaning).

**Scoring criteria:**

- **S 1.** Dashboard element meanings are completely unclear, cannot understand what charts, indicators, color encoding represent in business terms, users cannot obtain meaningful information.
- **S 2.** Dashboard element meanings are vague and unclear, insufficient business indicator explanations, unclear color encoding and status indication, users have difficulty understanding and need extensive guessing to understand content.
- **S 3.** Dashboard element meanings are basically clear, users can understand main business information and indicator meanings, but still have understanding barriers in some encoding or labeling aspects.
- **S 4.** Dashboard element meanings are clear and definite, business indicators have appropriate explanations, clear visual encoding, users can smoothly understand the business information conveyed by the dashboard.
- **S 5.** Dashboard element meanings are completely clear, business indicator explanations are detailed and accurate, all visual encoding has clear interpretation, users can understand all business information in the dashboard without any barriers.

---

**Insight discovery evaluation questions and scoring criteria:** The following prompt guides the generation of scoring rubrics for the *Insight Discovery* dimension. It assesses the chart's ability to reveal meaningful patterns, trends, or non-obvious findings.

---

**Prompt: Dashboard - Insight Discovery Evaluation Questions and Scoring Criteria (1–5 Scale)**

**Description:**
Evaluates whether the entire dashboard can easily identify significant and business-valuable key insights, whether key business indicators (KPIs) prominently display important information, and whether it can quickly discover business issues or opportunities that require attention.

**Scoring criteria:**

- **S 1.** Completely unable to provide any valuable business insights, key indicators not prominent, no significant or meaningful business discoveries, insight delivery completely failed.
- **S 2.** Difficult to identify meaningful business insights, insufficient highlighting of key indicators, most presented business information is trivial or obvious, lacking decision value.
- **S 3.** Can identify some basic business insights, key indicators have some prominence, but the significance and business value of insights are average, with limited guiding effects.
- **S 4.** Can clearly identify obvious and useful business insights, key indicators prominently displayed, discovered business patterns or anomalies have certain practical value and decision reference significance.
- **S 5.** Very intuitively reveals profound and highly business-valuable insights, key indicators extremely prominent, can quickly discover significant business trends, important anomalies, or key opportunities with important guiding significance for business decisions.

---

**Design style evaluation questions and scoring criteria:** The following prompt guides the generation of scoring rubrics for the *Design Style* dimension. It reflects the level of visual creativity, uniqueness, and design innovation present in the chart.

---

**Prompt: Dashboard - Design Style Evaluation Questions and Scoring Criteria (1–5 Scale)**

**Description:**
Evaluates whether the overall design of the dashboard has innovation and uniqueness, whether it can show attractive visual effects and business professionalism through novel design elements, unique style characteristics, and professional design language.

**Scoring criteria:**

**S 1.** Dashboard design completely lacks innovation, style is outdated or chaotic, no uniqueness, seriously insufficient professionalism.

**S 2.** Dashboard design lacks innovation, style is quite ordinary, basically uses common design techniques, insufficient uniqueness and professionalism.

**S 3.** Dashboard design has some innovative elements, but overall is relatively conventional, style characteristics not prominent enough, average professionalism.

**S 4.** Dashboard design has strong innovation, style is quite unique and professional, with certain originality and good business sense.

**S 5.** Dashboard design is extremely innovative, style is unique, novel, and professional, overall presents strong originality and high business aesthetics, leaving deep impressions.

---

**Visual composition evaluation questions and scoring criteria:** The following prompt guides the generation of scoring rubrics for the *Visual Composition* dimension. It focuses on the spatial arrangement and organization of visual elements for effective communication.

---

**Prompt: Dashboard - Visual Composition Evaluation Questions and Scoring Criteria (1–5 Scale)**

**Description:**
Evaluates whether the spatial layout of dashboard components and overall arrangement are reasonable and beautiful, including size proportions of various blocks, alignment relationships between components, spacing distribution, information density control, etc., forming a harmonious, orderly, and beautiful visual whole.

**Scoring criteria:**

**S 1.** Component layouts are chaotic or overall arrangement is seriously unreasonable, size proportions severely unbalanced, spacing distribution is disorderly, poor information density handling.

**S 2.** Component layouts or overall arrangement have obvious unreasonable aspects, inappropriate size proportions, uneven spacing distribution, or information too crowded.

**S 3.** Component layouts are basically reasonable, overall arrangement is acceptable, but may have small issues in proportional relationships, spacing handling, or information density.

**S 4.** Component layouts are good, overall arrangement is reasonable and beautiful, appropriate size proportions, proper spacing distribution, good information density control.

**S 5.** Component layouts are perfect, overall arrangement is extremely reasonable and beautiful, coordinated size proportions, aesthetic spacing distribution, appropriate information density control, extremely high space utilization efficiency.

---

**Color harmony evaluation questions and scoring criteria:** The following prompt guides the generation of scoring rubrics for the *Color Harmony* dimension. It evaluates how effectively the color scheme supports readability, aesthetic appeal, and visual coherence.

---

**Prompt: Dashboard - Color Harmony Evaluation Questions and Scoring Criteria (1–5 Scale)**

**Description:**
Evaluates whether color choices in the dashboard are appropriate and beautiful, including color combinations within various components, coordination of overall color schemes, unity of tone, and appropriate control of color quantity and saturation, ensuring both beauty and compliance with business professional requirements.

**Scoring criteria:**

**S 1.** Component color choices are very inappropriate, overall color matching is chaotic, tone conflicts or color use is seriously inappropriate, very poor aesthetic effect.

**S 2.** Component color choices are not appropriate enough, overall color matching is not coordinated enough, tone not unified or color use inappropriate, affecting aesthetic effect.

**S 3.** Component color choices are basically appropriate, overall color matching is acceptable, but may have small issues in tone unity, saturation, or business sense.

**S 4.** Component color choices are relatively appropriate, overall color schemes are quite coordinated, tone basically unified, appropriate color use, good business sense.

**S 5.** Component color choices are very appropriate, overall color schemes are highly coordinated and unified, perfect tone matching, appropriate control of color quantity and saturation, presenting excellent business aesthetics.

---

## C.2 EVALUATION PROMPT TEMPLATES

To guide consistent evaluation of visualizations, we present a structured prompt template built upon the classical *Fidelity, Expressiveness, and Aesthetics* principles, operationalized into six orthogonal sub-dimensions. The prompt is generated based on the rewritten evaluation questions and scoring criteria from Appendix C.1.

The following prompt template outlines how evaluators are instructed to conduct evaluation using these customized inputs. The prompt includes the following components:

- The `{total_count}` field specifies the total number of evaluation criteria distributed across the three main dimensions.

- The `{custom_count}` field indicates how many of these criteria adopt customized scoring guidelines tailored to the chart.

- The `{chart_description}` field provides metadata about the visualization, such as chart type and design structure.

- The `{fidelity_section}` field includes rewritten evaluation questions and scoring criteria aligned with the data fidelity sub-dimension.

- The `{expressiveness_section}` field covers the semantic readability and insight discovery sub-dimensions.

- The `{aesthetics_section}` field captures design-related sub-dimensions, including style, spatial composition, and color harmony.

---

**Prompt Template: Visualization Evaluation**

You are a rigorous data visualization evaluation expert. You must strictly judge each visualization based on the "Fidelity, Expressiveness, and Aesthetics" framework and the 1–5 scoring criteria for each metric.

Chart description: `{chart_description}`

---

Note: This chart has {total_count} evaluation metrics across three dimensions, of which {custom_count} use custom scoring criteria.

The evaluation follows the "Fidelity, Expressiveness, and Aesthetics" principle:

- Fidelity: Data accuracy and truthfulness
- Expressiveness: Information clarity and understandability
- Aesthetics: Visual aesthetics and refinement

For each evaluation question, provide a score from 1 to 5 and a reasoning based on the scoring criteria.

{fidelity_section}

{expressiveness_section}

{aesthetics_section}

**Return Format:** JSON object with the following structure:

```
{
  "data_fidelity": {"score": 1-5, "reasoning": "Your explanation here.
    "},
  "semantic_readability": {"score": 1-5, "reasoning": "Your
    explanation here."},
  "insight_discovery": {"score": 1-5, "reasoning": "Your explanation
    here."},
  "design_style": {"score": 1-5, "reasoning": "Your explanation here."
    },
  "visual_composition": {"score": 1-5, "reasoning": "Your explanation
    here."},
  "color_harmony": {"score": 1-5, "reasoning": "Your explanation here.
    "},
  "average_score": "the average of the above six scores, rounded to 2
    decimals"
}
```

Where for each metric, score should be an integer from 1 to 5 based on the above metric descriptions and the 1–5 scoring criteria, and reasoning should explain your choice. average_score is the average of all six scores rounded to 2 decimal places. Do not include any additional text, only the JSON object.

# D    MODEL IMPLEMENTATION AND TRAINING DETAILS

## D.1    SOFTWARE ENVIRONMENT

The training framework is based on the open-source library SWIFT (Scalable lightWeight Infrastructure for Fine-Tuning)[1], utilizing PyTorch and DeepSpeed (ZeRO Stage 2) for distributed training and memory optimization.

## D.2    REWARD FUNCTION

As described in the main text, our composite reward function, $R_{composite}$, is a weighted combination of an **accuracy reward ($R_{acc}$)** and a **format reward ($R_{format}$)**, with weights of 0.9 and 0.1, respectively.

**Accuracy Reward ($R_{acc}$)**    This component measures the proximity between the model's predicted scores (across the six dimensions and the overall average score) and the human-annotated ground-truth values. We employ a smooth exponential decay function to calculate the reward for each individual score:

$$R_{acc\_single} = \exp\left(-\frac{|score_{predicted} - score_{ground\text{-}truth}|}{0.5}\right) \tag{1}$$

The final accuracy reward is the average of the rewards calculated for all dimensional scores and the overall average score.

**Format Reward ($R_{format}$)**    This component ensures the model produces a complete and parsable JSON structure. The reward is 1.0 if the model's output contains all required fields (i.e., the `score` and `reasoning` for each of the six dimensions, plus the `average_score`); otherwise, the reward is 0.

## D.3    HYPERPARAMETER SETTINGS

We fine-tuned four representative open-source multimodal models: Qwen2.5-VL-7B-Instruct, Qwen2.5-VL-3B-Instruct, InternVL3-8B, and Llava-v1.6-mistral-7B. All models employed Low-Rank Adaptation (LoRA) for parameter-efficient fine-tuning with consistent hyperparameter configurations across architectures. Specifically, we set the LoRA rank and alpha to 128 and applied it to all linear layers. For reinforcement learning, we used the Group Relative Policy Optimization (GRPO) algorithm with a beta parameter of 0.01. The models were trained for 5 epochs with a learning rate of 1e-5, using a Cosine Annealing scheduler with a warmup ratio of 0.1. We used the AdamW optimizer with a weight decay of 0.01. The global batch size was 16 (per-device batch size of 1 with 4 gradient accumulation steps, across 4 GPUs). For computational efficiency, we utilized `bfloat16` mixed-precision training.

---

[1]`https://github.com/modelscope/swift`

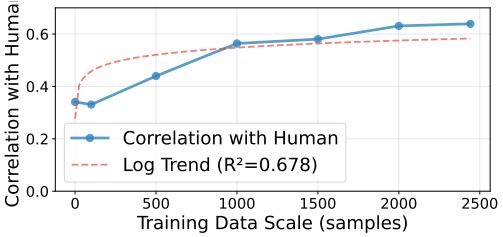 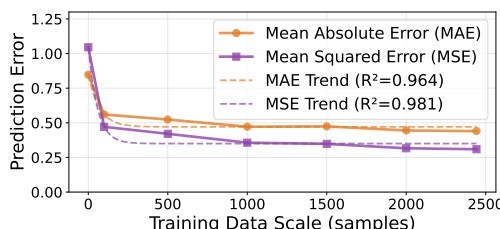

(a) Training data scale vs. human-model correlation   (b) Training data scale vs. prediction error

Figure 27: Impact of training data scale on VISJUDGE model performance.

# E  EXTENDED EXPERIMENTS

## E.1  HOW DOES TRAINING DATA SCALE AFFECT MODEL PERFORMANCE?

To evaluate data scaling effects and guide deployment strategies, we analyze VISJUDGE performance across different training data scales with a single training epoch. To ensure fairness, data samples are proportionally extracted based on visualization types and score distributions. Figure 27 reveals clear mathematical patterns as data scale increases.

**Predictable Scaling Laws.** Model performance follows well-defined trends: human–model correlation shows logarithmic growth (R²=0.678) from 0.341 to 0.639 as the training data scale increases to 2,442 samples, while prediction errors exhibit exponential decay with MAE decreasing from 0.847 to 0.440 (R²=0.964) and MSE from 1.045 to 0.309 (R²=0.981). The 500–1,000 sample range provides the most efficient improvement, contributing 41.7% of total correlation gains, while beyond 1,000 samples, marginal returns diminish but remain valuable for continued enhancement.

## E.2  GENERALIZATION TO REAL-WORLD APPLICATIONS

To validate the practical generalization capability of VISJUDGE, we integrate it into two real-world visualization systems: MatPlotAgent (visualization generation) and HAIChart (visualization recommendation). These scenarios differ significantly from our training set, providing rigorous OOD evaluation.

### E.2.1  EXPERIMENTAL SETUP

**MatPlotAgent (Visualization Generation).** We integrate VISJUDGE into MatPlotAgent (Yang et al., 2024) as a feedback mechanism for iterative quality improvement. We test seven generation models (GPT-5, GPT-4o, Claude-4-Sonnet, Claude-3.5-Sonnet, Gemini-2.5-Pro, Qwen2.5-VL-72B-Instruct, Qwen2.5-VL-7B-Instruct), comparing baseline (no feedback) against feedback from 7B base model, 72B base model, and VISJUDGE. Final visualizations are automatically evaluated using GPT-4o on a 0–100 scale, following the original MatPlotAgent evaluation setup, where higher scores indicate better overall visualization quality.

**HAIChart (Visualization Recommendation).** We integrate VISJUDGE into HAIChart (Xie et al., 2024) as a reward model to re-rank candidate visualizations. We evaluate on the VizML dataset containing real user-created Plotly visualizations, which represents a significant distribution shift from our training data. We compare the original system against versions using 7B, 72B, and VISJUDGE as reward models, measuring Hit@1 and Hit@3 accuracy on data queries, design choices, and overall recommendation tasks, where higher values indicate better top-$k$ recommendation accuracy.

### E.2.2  RESULTS AND ANALYSIS

**MatPlotAgent Results.** Table 7 shows that VISJUDGE achieves consistent quality improvements across all seven generation models, with an average gain of +6.07 points (range: +5.53 to +8.91). In stark contrast, base models without domain-specific fine-tuning often degrade performance: the 7B base model averages -2.39 points while the 72B base model averages -0.61 points. This degradation is attributable to their systematic evaluation biases documented in the main paper: score inflation (7B model $\mu = 3.89$ vs. human $\mu = 3.13$) and sharp score concentration around 4.0 limit dis-

Table 7: MatPlotAgent quality improvement with different feedback models.

| Generation Model | Direct Decoding (Baseline) | Qwen2.5-VL-7B as Feedback | Qwen2.5-VL-72B as Feedback | VISJUDGE as Feedback |
|---|---|---|---|---|
| GPT-5 | 67.15 | 66.49 | 69.80 | **72.07** |
| GPT-4o | 62.71 | 58.49 | 63.53 | **71.62** |
| Gemini-2.5-Pro | 68.10 | 65.04 | 66.12 | **72.20** |
| Claude-4-Sonnet | 64.50 | 67.25 | 65.20 | **72.23** |
| Claude-3.5-Sonnet | 63.89 | 59.23 | 63.14 | **69.04** |
| Qwen2.5-VL-72B-Instruct | 61.18 | 58.70 | 60.30 | **67.32** |
| Qwen2.5-VL-7B-Instruct | 49.76 | 45.38 | 44.94 | **55.29** |

Table 8: Visualization recommendation performance in HAIChart integration on VizML dataset. VISJUDGE as reward model improves recommendation accuracy across different tasks, outperforming larger base models.

| Task | Metric | HAIChart (Baseline) | Qwen2.5-VL-7B as a Reward | Qwen2.5-VL-72B as a Reward | VISJUDGE as a Reward |
|---|---|---|---|---|---|
| Data Queries | Hit@1 | 79.30% | 80.30% | 82.80% | **83.70%** |
| | Hit@3 | 91.90% | 90.70% | 92.10% | **93.20%** |
| Design Choices | Hit@1 | 48.70% | 45.90% | 46.50% | **48.80%** |
| | Hit@3 | 81.50% | 80.70% | 79.80% | **82.70%** |
| Overall | Hit@1 | 36.90% | 36.30% | 39.10% | **39.40%** |
| | Hit@3 | 67.40% | 69.10% | 70.20% | **72.70%** |

criminative capability, leading them to praise flawed visualizations while failing to identify genuine quality issues. Such biased feedback actively misleads generation models, directing iterative refinement toward suboptimal outputs. This demonstrates that domain-specific fine-tuning is not merely beneficial but essential—simply scaling model size without domain expertise can harm rather than help. Even state-of-the-art commercial models benefit significantly from VISJUDGE's feedback: GPT-5 +4.92, Claude-4-Sonnet +7.73, Gemini-2.5-Pro +4.10. This suggests that current generation models can effectively improve their outputs when provided with accurate quality assessments, and VISJUDGE provides this critical evaluation signal.

**HAIChart Results.** Table 8 shows VISJUDGE significantly improves recommendation accuracy across all metrics: Data Queries Hit@1 from 79.30% to 83.70% (+4.40%), Overall Hit@1 from 36.90% to 39.40% (+2.50%), and Overall Hit@3 from 67.40% to 72.70% (+5.30%). The 7B and 72B base models show minimal to moderate improvements, while VISJUDGE achieves larger gains. Notably, the VizML dataset represents a significant distribution shift from our training data (real Plotly community visualizations), yet VISJUDGE generalizes effectively, validating that it learns generalizable quality assessment principles rather than memorizing training patterns.

### E.2.3 CROSS-ARCHITECTURE GENERALIZATION

Beyond application scenarios, we validate that our fine-tuning methodology generalizes across diverse model architectures. We fine-tuned four different base models using the same training procedure: Qwen2.5-VL-7B-Instruct (correlation improves from 0.341 to 0.687, +101.5%), Qwen2.5-VL-3B-Instruct (from 0.272 to 0.648, +138.2%), InternVL3-8B (from 0.409 to 0.660, +61.4%), and LLaVA-1.6-mistral-7B (from 0.180 to 0.605, +236.1%). All models show substantial improvements after fine-tuning, with an average correlation increase of +0.334 (corresponding to +134.3% relative improvement), validating the robustness of our training methodology across diverse architectures.

The improvements are observed across models with different architectural designs (Qwen, InternVL, LLaVA), parameter scales (3B to 8B), and base capabilities, demonstrating that VISJUDGE-BENCH provides effective training signals that benefit diverse model families. Interestingly, models with weaker initial performance (e.g., LLaVA-1.6-mistral-7B with 0.180 baseline correlation) achieve larger relative improvements (+236.1%) compared to stronger base models, suggesting that VISJUDGE-BENCH is particularly valuable for enhancing models with limited initial visualization assessment capabilities.

### E.2.4 DISCUSSION

Our generalization experiments reveal key insights. VISJUDGE substantially outperforms both 7B and 72B base models across applications, demonstrating that domain-specific fine-tuning is more effective than model scaling. The model shows strong OOD performance on MatPlotAgent's generated visualizations and HAIChart's Plotly community data, validating that it learns generalizable quality assessment principles rather than memorizing training patterns.

The consistent improvements across diverse architectures and two distinct applications (generation and recommendation) validate VISJUDGE's practical utility and readiness for deployment in production visualization systems. These findings collectively demonstrate the value of VISJUDGE-BENCH for developing robust, deployable visualization assessment capabilities.

