# OpenReview forum: "VisJudge-Bench: Aesthetics and Quality Assessment of Visualizations"
_ICLR.cc/2026/Conference — ICLR 2026 Poster_

### Official Review · Reviewer_jQs1 · 2025-10-30

**Soundness:** 2
**Presentation:** 3
**Contribution:** 2
**Rating:** 2
**Confidence:** 3

**Summary:**

This paper presents VISJUDGE-BENCH, a large-scale, expert-annotated benchmark for evaluating the aesthetics and quality of data visualizations using multimodal large language models (MLLMs). The dataset covers 3,090 samples across a diverse set of visualization types and is accompanied by a multi-dimensional evaluation framework (“Fidelity-Expressiveness-Aesthetics”). The authors systematically benchmark several MLLMs and introduce VISJUDGE, a fine-tuned model that shows improved alignment with human expert ratings.

**Strengths:**

Addresses an Important Gap: The paper tackles the underexplored problem of automated visualization quality assessment, providing a resource that could catalyze further research in this area.
Comprehensive Dataset: VISJUDGE-BENCH is notable for its scale, diversity (single/multi/dashboard visualizations), and rigorous annotation pipeline.
Systematic Evaluation: The authors benchmark multiple MLLMs and provide detailed analysis of their strengths and weaknesses across different evaluation dimensions.
Open Resources: The dataset and benchmark are made available to the community, supporting reproducibility and future work.

**Weaknesses:**

Data Fidelity Unclear: The paper emphasizes “fidelity” as a core evaluation dimension, but it is not clear how this is assessed given that the source data for the visualizations is not available. Without access to the underlying data, it is difficult to objectively judge whether a visualization is faithful or misleading.
Limited Dataset Benchmarking: As a dataset paper, there is insufficient analysis of the final dataset’s strengths and limitations. There is little discussion of what types of visualizations, design flaws, or domains are well-represented or underrepresented, and how annotation quality varies across the dataset.
Scope of Contributions: Beyond dataset curation and MLLM fine-tuning, the paper’s contributions are somewhat incremental. The model fine-tuning, while useful, is marginal given the rapid progress in adapting MLLMs to new domains.
Generalization and Utility: The paper does not sufficiently characterize what the dataset enables (and what it does not), nor does it discuss how VISJUDGE or the benchmark would perform in real-world or out-of-distribution scenarios.

**Questions:**

How is data fidelity assessed without access to the source data? Are there proxies or heuristics used, and what are their limitations?
Can you provide a more systematic benchmarking of the dataset’s coverage, strengths, and weaknesses? For example, are certain chart types, styles, or domains underrepresented?
Did you analyze annotation quality and agreement across different annotator backgrounds or regions?
What are the practical limitations of using VISJUDGE-BENCH for training or evaluation in real-world visualization systems?

---

> ### Author Response · Authors · 2025-11-27
> **Response to Reviewer jQs1 (Part 1/6)**
>
> > **Q1. How is data fidelity assessed without access to the source data? Are there proxies or ​**​**heuristics**​**​ used, and what are their limitations? ​**
> >
> >**W1. Data Fidelity Unclear: The paper emphasizes “fidelity” as a core evaluation dimension, but it is not clear how this is assessed given that the source data for the visualizations is not available. ​**
>
> Thank you for raising this important concern. Since the charts in VisJudge-Bench are primarily collected from public web pages through search engines, the underlying source data is often unavailable. Moreover, in real-world scenarios, users typically evaluate charts based solely on the final visualization without access to the source data. Therefore, we adopted a **visual-level data fidelity assessment approach** that focuses on identifying visually detectable distortions in the chart presentation itself, rather than verifying against the original data.
>
> * **Theoretical foundation and related work.** This approach is grounded in established visualization research principles. We followed Tufte's foundational work on avoiding "graphical lies"[1], the visual diagnostic principles proposed by Lan & Liu[2], and the data fidelity evaluation approach used in ChartGalaxy[3] under similar conditions without source data. These studies demonstrate that evaluating data fidelity at the visual level is a widely accepted practice in the community when source data is unavailable.
> * **Specific evaluation criteria.** Our evaluation examines **visually detectable distortions** including:
>   * **Visual encoding ​consistency: ​**whether displayed numerical labels match visual encodings (e.g., bar heights proportional to values, line segment lengths, pie slice angles);
>   * **Axis and scale integrity:** whether axis scales are uniform, baselines start at appropriate values (typically zero for bar charts), and scale intervals are consistent;
>   * **Common misleading patterns:** truncated baselines, disproportional encodings, non-uniform scales, and inappropriate aspect ratios that exaggerate differences.
>   * As shown in the left panel of **Figure 1** ("Accurate Representation vs. Misleading Scale"), these visual distortion signals can be reliably identified through careful visual inspection. Detailed fidelity assessment criteria and examples are provided in **Appendix C.1.**
> * **Limitations and scope.** We acknowledge the limitations of this visual-level assessment approach: (1) it cannot verify the correctness of the source data itself or assess whether data selection and aggregation methods are appropriate; (2) it may not identify complex statistical manipulations or context-dependent misleading presentations that require domain knowledge.
>
> However, this approach effectively identifies the most common and visually apparent misleading signals (e.g., truncated baselines, disproportional encodings, scale distortions) that are most likely to mislead readers and are the quality issues most easily noticed by general users. We have explicitly stated the scope and boundaries of this assessment approach in the revised paper (**​Section 3.​1.2**​) and the limitations section (​**Section 7**​), and identified the acquisition and verification of complete source data as an important direction for future work.
>
> [1] Tufte, Edward R., and Peter R. Graves-Morris. ​*The visual display of quantitative information*​. Vol. 2. No. 9. Cheshire, CT: Graphics press, 1983.
>
> [2] Lan, Xingyu, and Yu Liu. "I Came Across a Junk: Understanding Design Flaws of Data Visualization from the Public's Perspective." IEEE TVCG (2024).
>
> [3] Li, Zhen, et al. "Chartgalaxy: A dataset for infographic chart understanding and generation." arXiv preprint arXiv:2505.18668 (2025)

---

> ### Author Response · Authors · 2025-11-27
> **Response to Reviewer jQs1 (Part 2/6)**
>
> > **Q2 Can you provide a more systematic benchmarking of the dataset’s coverage, strengths, and weaknesses? For example, are certain chart types, styles, or domains underrepresented? ​**
> >
> > **W2.1 Limited Dataset Benchmarking: As a dataset paper, there is insufficient analysis of the final dataset’s strengths and limitations. There is little discussion of what types of visualizations, design flaws, or domains are well-represented or underrepresented**
>
> Thank you for this valuable feedback. We have conducted extensive statistical analyses to characterize VisJudge-Bench's coverage, strengths, and limitations. Detailed results are in  **Appendix A.2** , with key findings below:
>
> 1. **Visualization Type Coverage Analysis**
>
> We systematically analyze the distribution of **32 chart subtypes** across three levels, comparing against real-world usage patterns. Detailed statistics are provided in  **Appendix A.2.2** ; key findings include:
>
> * **Single visualizations (1,041 samples):** Basic chart types account for 46%, mid-frequency analytical charts 38%, and specialized types 16%, exhibiting a clear long-tail pattern consistent with empirical findings by Borkin et al. [1] on real-world chart usage frequencies.
> * **Multiple visualizations (1,024 samples):** Comparison views dominate (65%), followed by small multiples (19%) and coordinated views (10%), highly consistent with Chen et al.'s [2] analysis of multi-view configuration patterns in real systems.
> * **Dashboards (1,025 samples):** Analytical dashboards account for 73%, operational 12%, and interactive/strategic 15%, aligning with research findings by Sarikaya et al. [3] and Bach et al. [4] on real-world dashboard ecosystems.
>
> 2. **Design Style Distribution Analysis**
>
> Design style scores show a mean of 2.97 with uniform distribution ( **Appendix Figure 11** ), indicating balanced coverage. Manual inspection identified three categories [1,5]: Professional/Standard Style (53.3%), Creative/Infographic Style (28.4%), and Minimalist Style (18.3%), demonstrating diverse design approaches in practice. **Detailed analysis is provided in Appendix A.2.3.**
>
> 3. **Domain Coverage Analysis**
>
> We annotated 93.8% of samples using IPTC classification standards [6]. Since the dataset is constructed from real-world web sources, its domain distribution naturally reflects actual visualization demand. Economy, business and finance (26.3%), science and technology (20.3%), and labor (15.7%) collectively account for 62.3%, aligning with high practical demand in these fields. The dataset also covers other important domains including society, health, politics, and environment. Domain co-occurrence shows economy pairs with labor and technology, while technology combines with health, environment, and education. **Detailed analysis is provided in Appendix A.2.4.**
>
> 4. **Low-Quality Sample Distribution and Design Flaw Analysis**
>
> We analyze quality score distributions and design flaw patterns in  **Appendices A.2.5-A.2.6 (Figure 9)** . Low-scoring samples (overall score < 2) decrease with complexity: 12.5% for single visualizations, 9.2% for multi-view, and 1.3% for dashboards. This pattern suggests dashboards, often from mature business scenarios with professional screening, maintain higher quality. **Single visualizations** show weakest performance in design style, with complex types (treemaps, scatter plots, network graphs) contributing over half of low-score cases. **Multiple visualizations** struggle with semantic readability and insight discovery in expressing view relationships, while **dashboards** face challenges in complex interactive scenarios. Representative cases across quality levels are in  **Appendices B.1-B.4** .
>
> 5. **Summary of Dataset Strengths and Limitations**
>
> ​**Strengths: ​**​(1) Comprehensive coverage of commonly used visualization types consistent with real-world usage patterns; (2) Balanced quality distribution across the full score range; (3) Multi-dimensional evaluation framework across six dimensions spanning fidelity, expressiveness, and aesthetics.
>
> **Limitations:** The current version focuses on static visualizations. Dynamic and interactive visualizations represent an important direction for future expansion. We discuss this in ​**Section 7**​.
>
> [1] Borkin et al., "What makes a visualization memorable?" IEEE TVCG (2013).
>
> [2] Chen et al., "Composition and configuration patterns in multiple-view visualizations." IEEE TVCG (2020).
>
> [3] Sarikaya et al., "What do we talk about when we talk about dashboards?" IEEE TVCG (2018).
>
> [4] Bach et al., "Dashboard design patterns." IEEE TVCG (2022).
>
> [5] Bateman et al., "Useful junk? The effects of visual embellishment on comprehension and memorability of charts." CHI (2010).
>
> [6] Li et al., "ChartGalaxy: A dataset for infographic chart understanding and generation." arXiv (2025).

---

> ### Author Response · Authors · 2025-11-27
> **Response to Reviewer jQs1 (Part 3/6)**
>
> > **Q3. Did you analyze annotation quality and agreement across different annotator backgrounds or regions? ​**
> >
> > **W2.2 How annotation quality varies across the dataset. ​**
>
> Thank you for raising this important question. We acknowledge that our initial submission lacked comprehensive annotation quality analysis. In response, we conducted systematic analysis examining annotation consistency and quality variations across different annotator backgrounds and visualization types. Detailed results have been added to**​ Appendix A.4.** Below we summarize the main findings:
>
> 1. **Overall Annotation Quality Assessment**
>
> We analyzed two types of MAE metrics: Crowd-crowd MAE (each annotator vs. crowd average) ranged from 0.64-0.70 across visualization types (dashboard: 0.6441, single view: 0.6842, multi-view: 0.6963). Crowd-expert MAE (each annotator vs. expert) ranged from 0.54-0.59 (dashboard: 0.5451, single view: 0.5534, multi-view: 0.5854). Notably, crowd-expert MAE was consistently lower than crowd-crowd MAE, indicating expert ratings effectively balance different annotator perspectives. These values are comparable to acceptable levels in prior crowdsourced visualization studies [1, 2]. Detailed analysis is provided in**​ Appendix A.4.1.**
>
> 2. **Annotator Background Analysis**
>
> We recruited **603 participants** through CloudResearch, an online crowdsourcing platform with a predominantly U.S.-based user base. Since our experiment used English as the interface language, the geographic distribution reflected the platform demographics: U.S. (535, 88.7%), Canada (30, 5.0%), New Zealand (24, 4.0%), UK (11, 1.8%), Ireland (2, 0.3%), and Australia (1, 0.2%).
>
> Analysis by professional background: professionals (41%) had within-group MAE of 0.4245 vs. non-professionals (59%) at 0.4881. Analysis by education: doctoral (7%, MAE 0.3944), master's (23%, MAE 0.4762), and undergraduate (70%, MAE 0.4825) groups showed better consistency with higher education, though differences were relatively modest. Critically, consistent error patterns were observed across all background subgroups, indicating our conclusions are robust across different annotator backgrounds. Detailed analysis is provided in**​ Appendix A.4.2.**
>
> 3. **Quality Control Impact**
>
> Experts reviewed all 3,090 samples individually. Only 11% required adjustments at the overall evaluation level (7% for malicious scoring patterns, 4% for statistical outliers), and 16% had sub-dimension adjustments, particularly for dimensions where standard deviation exceeded 1. This targeted approach preserves genuine diverse perspectives while correcting clear errors (e.g., misunderstanding criteria or providing consistently extreme ratings), demonstrating high overall crowd annotation quality. Detailed analysis is provided in **Appendix A.4.3.**
>
> 4. **Dimension-Specific Patterns**
>
> Among **511** samples requiring sub-dimension adjustments, objective dimensions had higher adjustment rates (semantic readability 19.20%, data fidelity 18.04%, insight discovery 16.79%) than aesthetic dimensions (design style 17.05%, visual composition 14.73%, color harmony 14.20%). Analysis by visualization type revealed: Single views (31.1% of adjusted samples) showed highest adjustments in data fidelity (20.06%); Multi-views (29.9%) in semantic readability (20.46%); Dashboards (38.9%) showed balanced adjustment rates. This validates our strategy of strict control for objective criteria while respecting subjective diversity. Detailed analysis is provided in**​ Appendix A.4.4.**
>
> In summary, our annotation process produced reliable and consistent evaluation data. Detailed statistics and analysis have been added to **Appendix A.4.** To facilitate reproducible research, **we will publicly release all annotation data, quality control records, and statistical analysis results.**
>
>
> [1] Heer, Jeffrey, and Michael Bostock. "Crowdsourcing graphical perception: using mechanical turk to assess visualization design." Proceedings of the SIGCHI conference on human factors in computing systems. 2010.
>
> [2] Kazai, Gabriella, Jaap Kamps, and Natasa Milic-Frayling. "An analysis of human factors and label accuracy in crowdsourcing relevance judgments." Information retrieval 16.2 (2013): 138-178.

---

> ### Author Response · Authors · 2025-11-27
> **Response to Reviewer jQs1 (Part 4/6)**
>
> > **W3. Scope of Contributions: Beyond dataset curation and MLLM fine-tuning, the paper’s contributions are somewhat incremental. The model fine-tuning, while useful, is marginal given the rapid progress in adapting MLLMs to new domains. ​**
>
> Thank you for your feedback. We would like to clarify the core positioning and main contributions of this work.
>
> **Research Motivation and Core Positioning:**
>
> * **Motivation:** The ability of MLLMs to evaluate visualization quality and aesthetics is crucial, as it directly affects their effectiveness as reward models in visualization generation and recommendation systems. However, there is a lack of systematic benchmarks to measure MLLMs' true capabilities in this task, preventing us from understanding model capability boundaries and providing clear directions for future improvements.
> * **Positioning:** Our primary goal is to establish the standardized evaluation benchmark VisJudge-Bench. As you pointed out, MLLMs' ability to adapt to new domains is rapidly improving, which precisely highlights the importance of establishing evaluation standards. Our work provides the necessary evaluation benchmark and high-quality training resources for this rapidly developing field.
> * **Approach:** We first discovered significant limitations of current MLLMs through benchmark evaluation, then further explored whether high-quality domain data could mitigate these issues through fine-tuning.
>
> **Benchmark Construction and Multi-dimensional Framework.** We constructed VisJudge-Bench containing 3,090 high-quality expert-annotated samples, covering 32 chart types and three complexity levels. Our "Fidelity-Expressiveness-Aesthetics" evaluation framework decomposes visualization quality into six fine-grained dimensions, filling the gap where existing work focuses only on single aspects (see **Section 3** and ​**Appendix A**​).
>
> **Systematic Capability Gap Analysis.** Through comprehensive evaluation of 12 representative models, we found that even the most advanced GPT-5 achieves only a correlation of 0.428 with experts. More importantly, we discovered that models exhibit a clear hierarchical capability structure: relatively better performance on fidelity dimensions but generally poor performance on aesthetic dimensions, indicating current models struggle with subjective aesthetic assessment. This systematic gap analysis fills a void in existing literature (see **Section 5.2.1** and ​**Table 3**​).
>
> **In-depth Failure Mode Analysis.** We identified typical problems with MLLMs: difficulty understanding multi-view relationships, insufficient information density processing capabilities, and lack of domain knowledge (see ​**Appendix B.6**​). For example, GPT-5 and Claude-3.5-Sonnet show negative correlations (-0.013 and -0.031) in dashboard data fidelity assessment, indicating systematic errors. These findings provide clear directions for future model improvements (see ​**Sections 5.2.2-5.2.4**​).
>
> **Effectiveness of Domain-Specific Fine-tuning.** Based on our understanding of model capability boundaries, we explored whether high-quality domain data could mitigate these issues through fine-tuning. We validated this approach through two complementary experiments:
>
> * **Cross-architecture ​**​​**validation**​: All fine-tuned models across different architectures and scales (Qwen2.5-VL-3B/7B, InternVL3-8B, LLaVA-1.6-7B) achieved significant improvements, with average MAE reduction of 31.5%-40.2% and correlation improvements of 0.3-0.4 (see ​**Section 5.2.1 and Table 3**​).
> * ​**Out-of-distribution generalization**​: In real-world applications, VisJudge achieved +6.07 points improvement in visualization generation (MatPlotAgent) and +4.40% Hit@1 in recommendation tasks (HAIChart). Notably, base models without domain training degraded performance (-2.39 for 7B, -0.61 for 72B), demonstrating that domain-specific fine-tuning is essential for effective feedback (see ​**Section 5.2.5 and Appendix E.2**​).
>
> In summary, we established a systematic research framework for visualization quality and aesthetic assessment, including evaluation benchmarks, problem diagnosis, and optimization approaches, rather than merely incremental technical improvements.

---

> > ### Author Response · Authors · 2025-11-27
> > **Response to Reviewer jQs1 (Part 5/6)**
> >
> > > **W4. Generalization and Utility: The paper does not sufficiently characterize what the dataset enables (and what it does not), nor does it discuss how VISJUDGE or the benchmark would perform in real-world or out-of-distribution scenarios.**
> > >
> > > **Q4. What are the practical limitations of using VISJUDGE-BENCH for training or evaluation in real-world visualization systems?**
> >
> > Thank you for your feedback. We agree on the need to clearly define the benchmark's capability boundaries and validate its generalization performance. To address this, we conducted additional out-of-distribution (OOD) experiments to verify VisJudge's generalization ability and practical value (see **Section 5.2.5** and ​**Appendix E.2**​), and added detailed discussions on the benchmark's limitations and future work in ​**Section 7**​. Below we highlight the key findings.
> >
> > 1. **Out-of-Distribution Generalization Experiments**
> >
> > To validate VisJudge's generalization ability and practical value, we tested it in two real-world visualization application scenarios that differ from our training set in terms of data distribution and  task settings.
> >
> > **Scenario 1: Integration into MatPlotAgent Visualization Generation System.** We integrated VisJudge into MatPlotAgent [1], an LLM-based visualization generation system, as a feedback mechanism to guide iterative quality improvement. We evaluated this integration on MatPlotBench, a benchmark for scientific data visualization tasks.
> >
> > **Table 1: Visualization Generation Quality Comparison**
> >
> > |Model|Direct Decoding|Qwen2.5-VL-7B as Feedback|Qwen2.5-VL-72B as Feedback|VisJudge as Feedback|
> > |-------------------------|-----------------|---------------------------|----------------------------|----------------------|
> > |GPT-5|67.15|66.49|69.8|**72.07(+4.92)**|
> > |GPT-4o|62.71|58.49|63.53|**71.62(+8.91)**|
> > |Gemini-2.5-Pro|68.1|65.04|66.12|**72.20(+4.10)**|
> > |Claude-4-Sonnet|64.5|67.25|65.2|**72.23(+7.73)**|
> > |Claude-3.5-Sonnet|63.89|59.23|63.14|**69.04(+5.15)**|
> > |Qwen2.5-VL-72B-Instruct|61.18|58.7|60.3|**67.32(+6.14)**|
> > |Qwen2.5-VL-7B-Instruct|49.76|45.38|44.94|**55.29(+5.53)**|
> >
> > *Note: Values in parentheses indicate improvement over Direct Decoding baseline.*
> >
> > Experimental results across seven generation models show that VisJudge achieves consistent quality improvements with an average gain of +6.07 points. Notably, base models without domain fine-tuning often degrade performance when used as feedback providers (7B: -2.39, 72B: -0.61 average), as they show significant gaps with human experts and lack domain-specific knowledge in visualization assessment (​**Section 5.2**​), leading to misleading feedback. In contrast, VisJudge consistently improves all generators, with even state-of-the-art models like GPT-5 and Claude-4-Sonnet achieving +4.92 and +7.73 improvements respectively, demonstrating that domain-specific fine-tuning is essential for providing effective feedback.
> >
> > **Scenario 2: Integration into HAIChart Visualization Recommendation System.** We tested VisJudge as a reward model in HAIChart [2] on the VizML dataset, which contains real user-created visualizations with different distributions from our training set.
> >
> > **Table 2: Visualization Recommendation Performance Comparison on VizML Dataset**
> >
> > |Task|Metric|HAIChart (Baseline)|Qwen2.5-VL-7B as Reward Model|Qwen2.5-VL-72B as Reward Model|VisJudge as Reward Model|
> > |----------------|--------|---------------------|-------------------------------|--------------------------------|--------------------------|
> > |DataQueries|Hit@1|79.30%|80.30%|82.80%|**83.70%(+4.40%)**|
> > ||Hit@3|91.90%|90.70%|92.10%|**93.20%(+1.30%)**|
> > |DesignChoices|Hit@1|48.70%|45.90%|46.50%|**48.80%(+0.10%)**|
> > ||Hit@3|81.50%|80.70%|79.80%|**82.70%(+1.20%)**|
> > |Overall|Hit@1|36.90%|36.30%|39.10%|**39.40%(+2.50%)**|
> > ||Hit@3|67.40%|69.10%|70.20%|**72.70%(+5.30%)**|
> >
> > *Note: Values in parentheses indicate improvement over HAIChart baseline.*
> >
> > VisJudge significantly improves recommendation accuracy: +4.40% on Data Queries Hit@1, +2.50% on Overall Hit@1, and +5.30% on Overall Hit@3. Base models show minimal to moderate improvements, while VisJudge consistently achieves larger gains, demonstrating effective quality evaluation in OOD scenarios.

---

> ### Author Response · Authors · 2025-11-27
> **Response to Reviewer jQs1 (Part 6/6)**
>
> > **W4. Generalization and Utility [continued] ​**
> >
> > **Q4. Practical limitations of VISJUDGE-BENCH [continued]**
>
> 2. **Cross-Architecture ​**​**Validation**
>
> Beyond validation in different application scenarios, we also tested the effectiveness of our fine-tuning strategy across different model architectures and scales (Qwen2.5-VL-3B/7B, InternVL3-8B, LLaVA-1.6-7B). Detailed results are presented in ​**Section 5.2.1 and Table 3**​.
>
> All fine-tuned models achieved significant improvements with Spearman correlation increasing by an average of over 0.3 (e.g., Qwen2.5-VL-7B from 0.341 to 0.687), demonstrating the universal value of our benchmark for training across different model architectures and scales.
>
> **Table ​3: Performance Comparison of Open-source MLLMs and VisJudge (MAE and Correlation metrics shown; for complete results see Table 3 in the paper)**
>
> |Metric|Type|Model|Overall|Fidelity|Readability|Insight|DesignStyle|Composition|Color|
> |---|---|---|---|---|---|---|---|---|---|
> |**MAE(↓)**|**Open-source**|InternVL3-8B|0.793|1.234|1.193|0.870|0.679|0.759|0.753|
> |||Llava-v1.6-mistral-7B|0.724|0.929|1.124|0.939|0.801|0.814|0.818|
> |||Qwen2.5-VL-3B-Instruct|0.821|1.085|1.258|1.088|0.723|0.727|0.808|
> |||Qwen2.5-VL-7B-Instruct|0.847|1.171|1.296|0.858|0.756|0.812|0.772|
> ||**VisJudge(Ours)**|InternVL3-8B|0.541|0.797|0.769|0.691|0.608|0.645|0.616|
> |||Llava-v1.6-mistral-7B|0.496|0.767|0.695|0.775|0.643|0.574|**0.598**|
> |||Qwen2.5-VL-3B-Instruct|0.491|0.705|0.721|0.696|0.625|0.571|0.616|
> |||Qwen2.5-VL-7B-Instruct|**0.421**|**0.661**|**0.648**|**0.677**|**0.580**|**0.545**|0.604|
> |**Corr.(↑)**|**Open-source**|InternVL3-8B|0.409|0.323|0.407|0.344|0.419|0.216|0.170|
> |||Llava-v1.6-mistral-7B|0.180|0.201|0.137|0.160|0.188|0.064|0.076|
> |||Qwen2.5-VL-3B-Instruct|0.272|0.199|0.222|0.275|0.338|0.130|0.155|
> |||Qwen2.5-VL-7B-Instruct|0.341|0.341|0.352|0.281|0.357|0.149|0.155|
> ||**VisJudge(Ours)**|InternVL3-8B|0.660|0.533|0.594|0.545|0.499|0.391|**0.420**|
> |||Llava-v1.6-mistral-7B|0.605|0.226|0.536|0.443|0.432|0.406|0.403|
> |||Qwen2.5-VL-3B-Instruct|0.648|0.533|0.581|**0.579**|0.504|0.490|0.402|
> |||Qwen2.5-VL-7B-Instruct|**0.687**|**0.574**|**0.628**|0.576|**0.568**|**0.513**|0.385|
>
> 3. **Dataset Capabilities and Limitations**
>
> **Benchmark Capabilities:** VisJudge-Bench provides multi-dimensional quality assessment across six dimensions, containing 3,090 samples spanning 32 chart types and multiple domains. As demonstrated by OOD experiments, models trained on this benchmark can: (1) assess static visualization quality with overall and sub-dimension scores; (2) provide feedback for visualization generation systems; (3) serve as reward models for recommendation tasks, showing robust generalization across different distributions and scenarios.
>
> **Limitations and Future Work:** The current benchmark focuses on static visualizations, with limited capability for dynamic and interactive features. Future work includes expanding dynamic visualization samples and introducing temporal evaluation dimensions, as discussed in **Section 7.**
>
> To facilitate reproducible research, we will publicly release all annotation data, quality control records, and trained VisJudge model weights, and welcome community contributions.
>
> [1] Yang Z, Zhou Z, Wang S, et al. MatPlotAgent: Method and Evaluation for LLM-Based Agentic Scientific Data Visualization. ACL (Findings)​, 2024.
>
> [2] Xie Y, Luo Y, Li G, et al. HAIChart: Human and AI Paired Visualization System. *Proceedings of the ​*​*VLDB*​​*​ Endowment*​, 2024, 17(11): 3178-3191.

---

> ### Comment · Reviewer_jQs1 · 2025-11-27
>
> Thanks a lot for the details response! Most of my concerns are addressed. I am happy to increase my rating.

---

> > ### Author Response · Authors · 2025-12-03
> > **Thank you for your detailed review and increased rating**
> >
> > Dear Reviewer jQs1,
> >
> > Thank you very much for your detailed review and for increasing your rating after reviewing our responses. We greatly appreciate the time and effort you invested in providing comprehensive feedback.
> >
> > Your concerns about data fidelity assessment, dataset benchmarking, annotation quality analysis, and generalization capabilities were crucial in helping us improve the paper. The additional experiments and analyses we conducted in response have substantially strengthened our work and made our contributions clearer.
> >
> > We are particularly grateful for your recognition that most of your concerns have been addressed. Your feedback has been invaluable in improving this work.
> >
> > Best regards,
> >
> > Authors

---

### Official Review · Reviewer_dX6V · 2025-10-31

**Soundness:** 2
**Presentation:** 3
**Contribution:** 3
**Rating:** 6
**Confidence:** 5

**Summary:**

This paper proposes a MLLM benchmarking framework called VISJUDGE-BENCH for visualization quality assessment, aiming to address the notable shortcomings of existing models in evaluating data fidelity, information expressiveness, and aesthetic quality. The research team constructed a benchmark dataset containing 3,090 expert-annotated samples, covering three major categories (single visualizations, multiple visualizations, and dashboards) across 32 chart types. Through systematic testing, it was found that the current state-of-the-art MLLMs exhibit significant biases when assessing visualization quality. To bridge this gap, the research team developed the VISJUDGE model, which reduces the Mean Absolute Error (MAE) to 0.442 (a 19.8% reduction) and increases the correlation with human expert ratings to 0.681 (a 58.7% improvement). This study provides a standardized framework and optimization pathway for the development of automated visualization evaluation technology.

**Strengths:**

Originality: It proposes VISJUDGE-BENCH, the first MLLM visualization evaluation benchmark covering the three dimensions of "Data Fidelity-Expressiveness-Aesthetics", filling the gap where existing benchmarks only focus on a single dimension. Through GRPO reinforcement learning and LoRA fine-tuning, it achieves significant alignment between MLLMs and human experts in visualization evaluation for the first time.
Quality: The benchmark construction undergoes rigorous multi-stage screening and multi-level quality control; the experimental design adopts a unified environment, uses multiple metrics, and covers scenarios of different complexities, ensuring data reliability.
Clarity: The paper uses figures and diagrams to support its arguments, with clear logic and presentatio.
Significance: It addresses the issue of the lack of standardized benchmarks for MLLM visualization evaluation, providing a benchmark for subsequent model optimization.

**Weaknesses:**

The paper only uses 3 annotators, which may pose a risk of concentrated subjective bias, and the elimination of abnormal scores may lead to insufficient valid data volume.
The 7 MLLMs tested in the experiment do not cover gradient comparison of models with different parameter scales, and the only open-source model included is Qwen2.5-VL-7B, making it difficult to verify the adaptability of VISJUDGE's fine-tuning strategy and the universality of the benchmark in the open-source ecosystem.

**Questions:**

The specific scoring criteria are only presented in the appendix, and some representative scoring criteria could be selected and incorporated into the main text.
All 3,090 samples of VISJUDGE-BENCH are static visualizations, and it is suggested to include dynamic samples to support the measurement of MLLM evaluation capabilities in dynamic scenarios.

---

> ### Author Response · Authors · 2025-11-27
> **Response to Reviewer dX6V (Part 1/2)**
>
> > **W1. The paper only uses 3 annotators, which may pose a risk of concentrated subjective bias, and the elimination of abnormal scores may lead to insufficient valid data volume. ​**
>
> Thank you for pointing out this concern. We would like to clarify the design of our annotation process and quality control strategy.
>
> * **Regarding the number of annotators:** A total of **603 annotators** participated in our annotation task. Each sample was independently rated by three different annotators randomly selected from this pool, rather than using a fixed team of three people. This design effectively avoids concentrated subjective bias while ensuring annotation diversity.
> * **Regarding the handling of abnormal ratings:** We did not simply remove all ratings with large standard deviations. Expert review aimed to identify abnormal ratings caused by misunderstandings or obvious errors, rather than eliminating all inconsistencies. Among the 3,090 samples, approximately 11% underwent adjustments at the overall evaluation level (7% for malicious scoring patterns, 4% for statistical outliers), 16% required sub-dimension adjustments where standard deviation exceeded 1, while the **majority of samples retained all original ratings. ​**
>
> This targeted approach preserved genuine diverse perspectives while correcting clear errors (e.g., misunderstanding criteria or providing consistently extreme ratings). Our quality control ensures annotation accuracy while respecting subjective diversity and avoiding over-standardization.
>
> We have added detailed annotation quality analysis in ​**Appendix A.4**​, including inter-annotator consistency, expert adjustment patterns, and rating characteristics across different annotator backgrounds. Additionally, to support future research on modeling annotation diversity, we will release the original ratings from all annotators along with their confidence scores in our public dataset.
>
> > **Q1. The specific scoring criteria are only presented in the appendix, and some representative scoring criteria could be selected and incorporated into the main text. ​**
>
> Thank you for this suggestion. We agree that presenting representative scoring criteria in the main text helps readers better understand our evaluation process and annotation details.
>
> * In the revised manuscript, **we have**​**added a concrete example of scoring criteria in the "Adaptive Question Generation Mechanism"** section of **Section 3.1.2.** We use the Data Fidelity dimension as an example to demonstrate how we generate customized evaluation questions and corresponding five-point scoring criteria for specific charts. Specifically, for a horizontal bar chart, the generated question asks whether "the horizontal bar chart accurately represents injury counts with a proper zero baseline and proportional bar lengths." The corresponding scoring criteria are clearly defined: a score of 1 indicates "major distortions, such as truncated axes or misleading scales that exaggerate differences," while a score of 5 confirms a "highly faithful representation where bar lengths are strictly proportional to the data."
> * This example demonstrates how we adapt abstract evaluation dimensions to specific chart types with concrete scoring criteria. Complete scoring criteria for all six dimensions are provided in **Appendix C.**
> * To further illustrate how scores translate to actual visualizations, **we have added representative cases in Appendix B** covering high, medium, and low score ranges across different visualization types, with detailed examples for each core dimension (fidelity, expressiveness, and aesthetics).
>
> > **Q2. All 3,090 samples of VISJUDGE-BENCH are static visualizations, and it is suggested to include dynamic samples to support the measurement of MLLM evaluation capabilities in dynamic scenarios.**
>
> Thank you for this valuable suggestion. We agree that **dynamic visualization evaluation is an important research direction.**
>
> * **Our current version focuses on static visualizations because:** (1) static visualizations dominate real-world applications and represent the most fundamental form; (2) their annotation process is well-defined, allowing us to establish standardized evaluation; (3) this provides a foundation for future extensions to dynamic scenarios.
> * We fully acknowledge the importance of evaluating dynamic and interactive visualizations, which require new annotation processes and evaluation metrics for temporal changes, animation effects, and interaction responsiveness. **We have stated this limitation in Section 7** and identify dynamic visualization expansion as important future work. Specifically, we plan to: (1) collect dynamic visualization samples; (2) design evaluation dimensions for temporal aspects and interactivity; (3) develop suitable annotation processes and quality control strategies. We welcome community contributions to advance dynamic visualization evaluation benchmarks.

---

> ### Author Response · Authors · 2025-11-27
> **Response to Reviewer dX6V (Part 2/2)**
>
> > **W2. The 7 MLLMs tested in the experiment do not cover gradient comparison of models with different parameter scales, and the only open-source model included is Qwen2.5-VL-7B, making it difficult to verify the adaptability of VISJUDGE's fine-tuning strategy and the universality of the benchmark in the open-source ecosystem.**
>
> Thank you for pointing out this important issue. We fully agree on the need to test models of different parameter scales to verify the universality of our benchmark and fine-tuning strategy. In the revised manuscript, ​**we conducted extensive additional experiments covering open-source models with different parameter scales and architectures ​**​(see **Section 5.2.1 and Table 3** for details). Below we highlight the key findings.
>
> * **Additional Inference Evaluation across Model Scales:** We evaluated multiple open-source models spanning a complete gradient from small-scale (3B) to large-scale (72B parameters), including the Qwen2.5-VL series (3B/7B/32B/72B), InternVL3-8B, and LLaVA-1.6-7B. Key findings include: (1) Even the largest open-source model (Qwen2.5-VL-72B) achieves performance comparable to the best closed-source model GPT-5 (correlation 0.440 vs. 0.428, MAE 0.702 vs. 0.553); (2) **Model performance generally improves with scale, but shows diminishing returns beyond 32B parameters** (correlation increases from 0.435 to only 0.440 from 32B to 72B), indicating that model architecture and training methods are equally important as scale in visualization assessment tasks.
>
> **Table 1: Performance Comparison across Different Model Scales (MAE and Correlation metrics shown; for complete results see Table 3 in the paper)**
>
> |Metric|Model|Overall|Fidelity|Readability|Insight|DesignStyle|Composition|Color|
> |---|---|---|---|---|---|---|---|---|
> |**MAE(↓)**|Qwen2.5-VL-3B-Instruct|0.821|1.085|1.258|1.088|0.723|**0.727**|0.808|
> ||Qwen2.5-VL-7B-Instruct|0.847|1.171|1.296|0.858|0.756|0.812|0.772|
> ||Qwen2.5-VL-32B-Instruct|0.703|**0.909**|0.987|**0.801**|0.678|0.761|0.719|
> ||Qwen2.5-VL-72B-Instruct|**0.702**|0.999|**0.926**|0.811|**0.663**|0.735|**0.717**|
> |**Corr.(↑)**|Qwen2.5-VL-3B-Instruct|0.272|0.199|0.222|0.275|0.338|0.130|0.155|
> ||Qwen2.5-VL-7B-Instruct|0.341|0.341|0.352|0.281|0.357|0.149|0.155|
> ||Qwen2.5-VL-32B-Instruct|0.435|**0.348**|0.468|0.408|**0.449**|**0.200**|**0.268**|
> ||Qwen2.5-VL-72B-Instruct|**0.440**|0.331|**0.479**|**0.416**|0.435|0.165|0.251|
>
> * **Fine-tuning Effectiveness across Architectures and Scales:**​​**We verified our fine-tuning strategy across multiple architectures ​**​(Qwen2.5-VL-3B/7B, InternVL3-8B, LLaVA-1.6-7B). **All fine-tuned models achieved substantial improvements with ​**​**Spearman**​​**​ correlation increasing by 0.3-0.4 ​**​(e.g., Qwen2.5-VL-7B from 0.341 to 0.687). Notably, the fine-tuned 3B model (correlation 0.648) surpasses all unfine-tuned open-source models and approaches some closed-source models, demonstrating that **high-quality ​**​**training data**​**​ and targeted fine-tuning can effectively compensate for model scale limitations. ​**Furthermore, fine-tuned ​**smaller models even outperform unfine-tuned larger models in certain dimensions ​**​(e.g., VisJudge(Qwen2.5-VL-3B) achieves MAE 0.705 in data fidelity vs. unfine-tuned 72B's 0.999).
>
> **Table 2: Fine-tuning Performance Comparison across Architectures (MAE and Correlation metrics shown; for complete results see Table 3 in the paper)**
>
> |Metric|Type|Model|Overall|Fidelity|Readability|Insight|DesignStyle|Composition|Color|
> |---|---|---|---|---|---|---|---|---|---|
> |**MAE(↓)**|**Open-source**|InternVL3-8B|0.793|1.234|1.193|0.870|0.679|0.759|0.753|
> |||Llava-v1.6-mistral-7B|0.724|0.929|1.124|0.939|0.801|0.814|0.818|
> |||Qwen2.5-VL-3B-Instruct|0.821|1.085|1.258|1.088|0.723|0.727|0.808|
> |||Qwen2.5-VL-7B-Instruct|0.847|1.171|1.296|0.858|0.756|0.812|0.772|
> ||**VisJudge(Ours)**|InternVL3-8B|0.541|0.797|0.769|0.691|0.608|0.645|0.616|
> |||Llava-v1.6-mistral-7B|0.496|0.767|0.695|0.775|0.643|0.574|**0.598**|
> |||Qwen2.5-VL-3B-Instruct|0.491|0.705|0.721|0.696|0.625|0.571|0.616|
> |||Qwen2.5-VL-7B-Instruct|**0.421**|**0.661**|**0.648**|**0.677**|**0.580**|**0.545**|0.604|
> |**Corr.(↑)**|**Open-source**|InternVL3-8B|0.409|0.323|0.407|0.344|0.419|0.216|0.170|
> |||Llava-v1.6-mistral-7B|0.180|0.201|0.137|0.160|0.188|0.064|0.076|
> |||Qwen2.5-VL-3B-Instruct|0.272|0.199|0.222|0.275|0.338|0.130|0.155|
> |||Qwen2.5-VL-7B-Instruct|0.341|0.341|0.352|0.281|0.357|0.149|0.155|
> ||**VisJudge(Ours)**|InternVL3-8B|0.660|0.533|0.594|0.545|0.499|0.391|**0.420**|
> |||Llava-v1.6-mistral-7B|0.605|0.226|0.536|0.443|0.432|0.406|0.403|
> |||Qwen2.5-VL-3B-Instruct|0.648|0.533|0.581|**0.579**|0.504|0.490|0.402|
> |||Qwen2.5-VL-7B-Instruct|**0.687**|**0.574**|**0.628**|0.576|**0.568**|**0.513**|0.385|
>
> These comprehensive experiments thoroughly verify the universality of VisJudge-Bench in the open-source ecosystem and the adaptability of our fine-tuning strategy across different parameter scales and architectures.

---

### Official Review · Reviewer_w8Z6 · 2025-10-31

**Soundness:** 3
**Presentation:** 3
**Contribution:** 3
**Rating:** 4
**Confidence:** 3

**Summary:**

VisJudge-Bench introduces a large-scale benchmark for evaluating multimodal large language models (MLLMs) on visualization aesthetics and quality assessment. It contains expert-annotated samples across diverse chart types and visualization settings. Experimental results show that existing MLLMs, including GPT-5, still fall short of human performance. To mitigate this, the authors propose VisJudge, a model trained via the GRPO method, which achieves substantial improvements in accuracy and alignment with human judgments.

**Strengths:**

1. The topic is interesting and the task is practical in real-world applications.
2. The authors constructed VISJUDGE-BENCH, a comprehensive benchmark based on the principles of Fidelity, Expressiveness, and Aesthetics to evaluate MLLMs’ visualization assessment capabilities.
3. They systematically evaluated representative MLLMs and found significant gaps compared with human expert standards.

**Weaknesses:**

1. Annotators are mainly presented with specific questions and asked to do scoring tasks. Are they asked to do enough free-form critique? It may be beneficial to diversity and generalizability.

2. While the construction of the benchmark dataset is well illustrated, I am very confused about the use of it. In Section 4, authors claimed: "To validate VisJudge-Bench as an effective training resource...". I am confused with the authors' purpose.

(1) Given that its called 'VisJudge-Bench' and referred as a benchmark in prior sections, as well as its tiny data size (only ~3k images), it is naturally a benchmark. However, authors further split it into train/test set and fintuned a MLLM on it. What is the purpose of the experiment?

(2) Training a MLLM with such a tiny dataset and testing it on the dataset from the same distribution does not make much sense. The model can easily overfit and achieve good results with the certain data distribution, therefore the results are not convincing. Generalizability is a big concern.

In summary, the experiment part is confusing. As a benchmark paper, we may expect a comprehensive evaluation of MLLMs on all of the
collected data, instead of an experiment where the benchmark data is split into two parts for training and testing, which does not convey much useful information.

Overall the topic is interesting and the construction of dataset can contribute to related studies. However, the organization of the paper is confusing. The whole experiment section is a bit off. I may accept it as a benchmark paper but not as a dataset one. The current version is not a solid ICLR submission.

**Questions:**

1. Annotators are mainly presented with specific questions and asked to do scoring tasks. Are they asked to do enough free-form critique? It may be beneficial to diversity and generalizability.

2. What is the purpose of the experiment part?

---

> ### Author Response · Authors · 2025-11-27
> **Response to Reviewer w8Z6 (Part 1/4)**
>
> > **W1. Annotators are mainly presented with specific questions and asked to do scoring tasks. Are they asked to do enough free-form critique? It may be beneficial to diversity and generalizability.**
>
> Thank you for this suggestion. We understand the value of free-form critique in capturing diversity. While our annotation primarily used structured scoring, **we incorporated multiple mechanisms to capture diverse perspectives and ensure generalizability:**
>
> * **Adaptive Scoring Criteria: ​**Our evaluation **questions and scoring criteria were dynamically generated for each visualization ​**based on expert-designed evaluation standards (​**Figure 3, Section 3.1.2, Appendix C.1**​). The system extracts chart metadata and populates expert-defined templates to generate customized questions. As illustrated in ​**Figure 3**​, for a horizontal bar chart, the Data Fidelity question specifically asks whether it "accurately represents injury counts with a proper zero baseline and proportional bar lengths," with criteria ranging from "major distortions" (score 1) to "highly faithful representation" (score 5). This chart-specific customization allows annotators to evaluate based on each visualization's particular characteristics rather than applying rigid generic templates, thereby capturing diverse judgments while maintaining consistency.
> * **Confidence Indicators: ​**To capture annotation uncertainty while maintaining structured scoring advantages, annotators reported confidence levels (1-5) for each rating and could flag uncertain cases. **Statistics show 63.27% had high confidence (scores 4-5, average 3.76), with only 2.47% flagged as uncertain.** This indicates most annotators could make clear judgments within our framework while being able to signal uncertainty on subjective dimensions, allowing us to identify cases requiring additional expert attention.
> * **Post-task Open Feedback: ​**After completing all annotation tasks, annotators provided open-ended feedback on the annotation process and question design. We collected and analyzed this feedback to iteratively improve the annotation guidelines.
> * **Respecting Subjective Diversity: Our quality control strategy maintained appropriate ​**​**tolerance**​**​ for subjective dimensions. ​**Analysis shows that objective dimensions (data fidelity 18.04%, semantic readability 19.20%) had significantly higher adjustment rates than aesthetic dimensions (color harmony 14.20%, visual composition 14.73%). This demonstrates that while ensuring annotation consistency on objective criteria, we respect and preserve annotators' diverse perspectives on subjective aesthetic dimensions (​**Appendix A.4.4**​).
>
> We acknowledge that incorporating more extensive free-form critique per sample could provide additional qualitative insights. However, for our large-scale benchmark with over 9,000 annotations, we balanced this against the need for systematic quantitative evaluation that enables reliable model training and benchmarking. Our approach captures diversity through adaptive criteria, confidence indicators, and tolerance for subjective variation while maintaining the consistency necessary for establishing evaluation standards.

---

> ### Author Response · Authors · 2025-11-27
> **Response to Reviewer w8Z6 (Part 2/4)**
>
> > **W2.1 While the construction of the benchmark dataset is well illustrated, I am very confused about the use of it. In Section 4, authors claimed: "To validate VisJudge-Bench as an effective training resource...". I am confused ​**​**with**​**​ the authors' purpose.**
> >
> > **Q2:  Given that its called 'VisJudge-Bench' and referred as a benchmark in prior sections, as well as its tiny data size (only \~3k images), it is naturally a benchmark. However, authors further split it into train/test set and fintuned a MLLM on it. What is the purpose of the experiment?**
>
> Thank you for pointing out this confusion. We would like to clarify the dataset's positioning and the paper's organizational logic.
>
> **Core Positioning of the Dataset:** VisJudge-Bench is **primarily positioned as a benchmark** for systematically evaluating MLLMs' capabilities in visualization quality and aesthetic assessment. In the revised manuscript, we have clarified that **Section 4** introduces the fine-tuning exploration on open-source models, while **Section 5** serves as the core experimental evaluation section based on the benchmark.
>
> **Paper Organization Logic:** The core objective of this paper is to systematically evaluate MLLMs' capabilities in visualization quality and aesthetic assessment. Building on the high-quality VisJudge-Bench, we further explore whether domain-specific fine-tuning can improve this capability. The paper is organized as follows:
>
> * **Section 4 (Open-source Model Fine-tuning):** Based on VisJudge-Bench's high-quality annotated data, we explore: **Can open-source models effectively learn visualization aesthetics and quality assessment tasks through fine-tuning ​**and match or even surpass the evaluation capabilities of closed-source models? To this end, we fine-tune multiple open-source model architectures across different parameter scales, developing VisJudge models, which are then evaluated alongside closed-source and open-source baselines in**​ Section 5.**
> * **Section 5 (Unified Evaluation and Analysis):** We conduct **comprehensive evaluation and comparative analysis of all model types** (closed-source models, open-source baseline models, and fine-tuned VisJudge) on the same representative test set. The evaluation systematically reveals different models' capability boundaries, systematic biases, and typical failure modes. We find that even the most advanced GPT-5 achieves only 0.428 correlation with human experts, while fine-tuned VisJudge significantly narrows this gap (improving correlation to 0.687), **demonstrating that high-quality domain data can effectively improve MLLMs' visualization evaluation capabilities through fine-tuning.**
>
> **Dataset Split Rationale:** Our dataset systematically covers ​**32 visualization types**​, multiple design styles, and major application domains, with all **3,090 samples** undergoing rigorous quality control by **603 annotators** and an expert team (see ​**Appendices A.2 and A.4**​). Building on this foundation of high-quality annotations and comprehensive coverage, we employ stratified random sampling for the dataset split. The test set contains **648 samples ​**and fully preserves the distributional characteristics of the original dataset across visualization types, complexity levels, and quality ranges, **ensuring sufficient representativeness**. Detailed statistics are provided in **Appendix A.2.7.**
>
> **Main Research Contributions:** Based on this experimental design, our systematic evaluation reveals:
>
> * **Revealing significant capability gaps (Section 5.2.1 and Table 3)**: Even GPT-5 achieves only 0.428 correlation with experts (MAE 0.553), with models performing relatively better on objective dimensions but significantly weaker on subjective aesthetics.
> * **Identifying systematic biases (Section 5.2.2 and Figure 4)**: Most models exhibit score inflation (e.g., Qwen2.5-VL-7B mean 3.89 vs. human 3.13) or overly conservative patterns (e.g., Gemini-2.5-Pro mean 3.02), while VisJudge achieves human-like distributions (mean 3.11).
> * **Discovering performance degradation with complexity (Section 5.2.3 and Appendix B.6)**: All models show degradation from single views to dashboards, with GPT-5 and Claude-3.5-Sonnet showing negative correlations in dashboard assessment.
> * **Analyzing failure modes (Section 5.2.4 and Appendix B)**: Models struggle with multi-view understanding, confusing form with content and evaluating views independently rather than holistically.
> * **Validating fine-tuning effectiveness and generalization (Section 5.2.5 and Appendix E.2)**: VisJudge improves correlation to 0.687 and demonstrates strong generalization in visualization generation (+6.07 in MatPlotAgent) and recommendation tasks (+5.3% Hit@3 in HAIChart).
>
> These findings demonstrate the value of our benchmark for systematic model evaluation and provide important directions for future research.

---

> ### Author Response · Authors · 2025-11-27
> **Response to Reviewer w8Z6 (Part 3/4)**
>
> > **W2.2 Training a MLLM with such a tiny dataset and testing it on the dataset from the same distribution does not make much sense. The model can easily overfit and achieve good results with the certain data distribution, therefore the results are not convincing. Generalizability is a big concern.**
>
> Thank you for raising this important concern about generalization. We fully acknowledge the potential risk of overfitting when training and testing on the same distribution. To verify that VisJudge has learned generalizable evaluation principles rather than simply memorizing training samples, we conducted two types of validation:
>
> **First, we tested VisJudge's generalization ability in real-world application scenarios with different distributions.** These scenarios differ from our training set in terms of data sources, chart styles, and quality distributions, effectively testing the model's out-of-distribution performance (see **Section 5.2.5 and Appendix E.2 ​**for details).
>
> **Scenario 1: Integration into MatPlotAgent Visualization Generation System.** We integrated VisJudge into MatPlotAgent [1], an LLM-based visualization generation system, as a feedback mechanism to guide iterative quality improvement. We evaluated this integration on MatPlotBench, a benchmark for scientific data visualization tasks.
>
> Experimental results across seven generation models show that **VisJudge achieves consistent quality improvements with an average gain of +6.07 points.** Notably, base models without domain fine-tuning often degrade performance when used as feedback providers (7B: -2.39, 72B: -0.61 average), as they show significant gaps with human experts and lack domain-specific knowledge in visualization assessment (​**Section 5.2**​), leading to misleading feedback. In contrast, VisJudge consistently improves all generators, ​**with even state-of-the-art models like GPT-5 and Gemini-2.5-Pro achieving +4.92 and +4.10 improvements respectively**​, demonstrating that domain-specific fine-tuning is essential for providing effective feedback.
>
> **Table 1: Visualization Generation Quality Comparison ​**
>
> |Model|Direct Decoding|Qwen2.5-VL-7B as Feedback|Qwen2.5-VL-72B as Feedback|VisJudge as Feedback|
> |-------------------------|-----------------|---------------------------|----------------------------|----------------------|
> |GPT-5|67.15|66.49|69.8|**72.07(+4.92)**|
> |GPT-4o|62.71|58.49|63.53|**71.62(+8.91)**|
> |Gemini-2.5-Pro|68.1|65.04|66.12|**72.20(+4.10)**|
> |Claude-4-Sonnet|64.5|67.25|65.2|**72.23(+7.73)**|
> |Claude-3.5-Sonnet|63.89|59.23|63.14|**69.04(+5.15)**|
> |Qwen2.5-VL-72B-Instruct|61.18|58.7|60.3|**67.32(+6.14)**|
> |Qwen2.5-VL-7B-Instruct|49.76|45.38|44.94|**55.29(+5.53)**|
>
> *Note: Values in parentheses indicate improvement over Direct Decoding baseline.*
>
>
> **Scenario 2: Integration into HAIChart Visualization Recommendation System.** We tested VisJudge as a reward model in HAIChart [2] on the VizML dataset, which contains real user-created visualizations with significantly different distributions from our training set.
>
> VisJudge significantly **improves recommendation accuracy: +4.40% on Data Queries Hit@1, +2.50% on Overall Hit@1, and +5.30% on Overall Hit@3. ​**Base models show minimal to moderate improvements, while VisJudge consistently achieves larger gains, demonstrating effective quality evaluation in OOD scenarios.
>
> **Table 2: Visualization Recommendation Performance Comparison on VizML Dataset**
>
>
> |Task|Metric|HAIChart (Baseline)|Qwen2.5-VL-7B as Reward Model|Qwen2.5-VL-72B as Reward Model|VisJudge as Reward Model|
> |----------------|--------|---------------------|-------------------------------|--------------------------------|--------------------------|
> |DataQueries|Hit@1|79.30%|80.30%|82.80%|**83.70%(+4.40%)**|
> ||Hit@3|91.90%|90.70%|92.10%|**93.20%(+1.30%)**|
> |DesignChoices|Hit@1|48.70%|45.90%|46.50%|**48.80%(+0.10%)**|
> ||Hit@3|81.50%|80.70%|79.80%|**82.70%(+1.20%)**|
> |Overall|Hit@1|36.90%|36.30%|39.10%|**39.40%(+2.50%)**|
> ||Hit@3|67.40%|69.10%|70.20%|**72.70%(+5.30%)**|
>
> *Note: Values in parentheses indicate improvement over HAIChart baseline.*
>
> [1] Yang Z, Zhou Z, Wang S, et al. MatPlotAgent: Method and Evaluation for LLM-Based Agentic Scientific Data Visualization. ACL (Findings), 2024.
>
> [2] Xie Y, Luo Y, Li G, et al. HAIChart: Human and AI Paired Visualization System. Proceedings of the VLDB Endowment, 2024, 17(11): 3178-3191.

---

> ### Author Response · Authors · 2025-11-27
> **Response to Reviewer w8Z6 (Part 4/4)**
>
> > **W2.2 (continued): Training a MLLM with such a tiny dataset and testing it on the dataset from the same distribution does not make much sense. The model can easily overfit and achieve good results with the certain data distribution, therefore the results are not convincing. Generalizability is a big concern.**
>
> **Second, we verified the effectiveness of the fine-tuning strategy across different model architectures and scales.** We conducted fine-tuning experiments on multiple architectures (Qwen2.5-VL-3B/7B, InternVL3-8B, LLaVA-1.6-7B), as detailed in ​**Section 5.2.1 and Table 3**​.
>
> All fine-tuned models achieved significant improvements with Spearman correlation increasing by an average of over 0.3 (e.g., Qwen2.5-VL-7B from 0.341 to 0.687), demonstrating the universal value of our benchmark for training across different model architectures and scales. These consistent improvements further indicate that the model learns generalizable evaluation principles rather than simply memorizing training set features.
>
> **Table 3: Performance Comparison of Open-source MLLMs and VisJudge (Correlation metrics shown; for complete results including MAE and ​**​**MSE**​**​ see Table 3 in Section 5.2.1)**
>
> |Type|Model|Overall|Fidelity|Readability|Insight|DesignStyle|Composition|Color|
> |---|---|---|---|---|---|---|---|---|
> |**Open-source**|InternVL3-8B|0.409|0.323|0.407|0.344|0.419|0.216|0.170|
> ||Llava-v1.6-mistral-7B|0.180|0.201|0.137|0.160|0.188|0.064|0.076|
> ||Qwen2.5-VL-3B-Instruct|0.272|0.199|0.222|0.275|0.338|0.130|0.155|
> ||Qwen2.5-VL-7B-Instruct|0.341|0.341|0.352|0.281|0.357|0.149|0.155|
> |**VisJudge(Ours)**|InternVL3-8B|0.660|0.533|0.594|0.545|0.499|0.391|**0.420**|
> ||Llava-v1.6-mistral-7B|0.605|0.226|0.536|0.443|0.432|0.406|0.403|
> ||Qwen2.5-VL-3B-Instruct|0.648|0.533|0.581|**0.579**|0.504|0.490|0.402|
> ||Qwen2.5-VL-7B-Instruct|**0.687**|**0.574**|**0.628**|0.576|**0.568**|**0.513**|0.385|
>
> Through these two types of validation (effectiveness in real-world out-of-distribution application scenarios and consistent improvements across model architectures), we have demonstrated that **VisJudge possesses generalizable evaluation capabilities rather than simple ​overfitting.** These additional experimental results and detailed analyses have been added to the revised manuscript.

---

### Official Review · Reviewer_iwcV · 2025-11-01

**Soundness:** 3
**Presentation:** 4
**Contribution:** 3
**Rating:** 8
**Confidence:** 4

**Summary:**

This paper introduces VISJUDGE-BENCH, a comprehensive benchmark for evaluating the capabilities of Multimodal Large Language Models (MLLMs) in assessing the quality and aesthetics of data visualizations. The authors propose a "Fidelity-Expressiveness-Aesthetics" framework, operationalized into six measurable dimensions, to create a dataset of 3,090 expert-annotated samples. The work systematically benchmarks several state-of-the-art MLLMs, revealing significant gaps compared to human expert judgment, particularly in subjective aesthetic evaluations. To address this, the paper also presents VISJUDGE, a fine-tuned model that demonstrates substantially improved alignment with human ratings. The paper is well-written, the experiments are thorough, and the dataset is a valuable contribution to both the machine learning and visualization communities. However, the work rests on the critical assumption that a singular ground truth for visualization quality can be established, a premise that may oversimplify the inherent subjectivity and context-dependency of human perception, which is the main point of concern.

**Strengths:**

1. **Timely problem:**
   The paper addresses a relevant and underexplored challenge—evaluating MLLMs on data visualizations, which differ significantly from general images and remain difficult for current models.

2. **Well-motivated framework:**
   The Fidelity–Expressiveness–Aesthetics framework is grounded in visualization theory and provides a clearer structure than using a single quality score.

3. **Solid experiments:**
   The benchmarking results across multiple MLLMs offer useful insights into current model strengths, weaknesses, and issues such as score inflation.

4. **Reproducibility efforts:**
   The release of data, code, and detailed documentation supports replicability and future work.

5. **Clear presentation:**
   The paper is well-written, logically structured, and supported by clear visuals and appendices.

**Weaknesses:**

1. Fundamental Assumption of a Singular "Ground Truth": The primary and most significant weakness of this work stems from a fundamental assumption that a singular, objective "ground truth" for visualization quality exists and can be established. This is particularly contentious for subjective dimensions like Aesthetics and even Expressiveness, where human preferences are known to be diverse, context-dependent, and often lack a single consensus.
The paper's methodology—collecting scores from three crowdworkers and then using expert review to resolve disagreements—effectively enforces a consensus rather than acknowledging or modeling the inherent diversity in human perception. The authors do not sufficiently discuss the possibility that disagreement among annotators may not be a sign of low-quality annotation but rather a reflection of genuine, valid differences in interpretation and preference within the human population. This core assumption is critical, as its potential invalidity could undermine the significance of the benchmark's "ground truth" and, consequently, the meaning of the entire evaluation framework. Regrettably, because the potential invalidity of this core assumption would render all subsequent experiments meaningless, this is the main reason I cannot give a higher score, despite the authors' thorough and professional work.
2. Domain-Specificity and Audience Dependence: A compelling, concrete example illustrates the issue above. In Figure 2 and Figure 21, the authors present a "layered heatmap with candlesticks" as an example of a poor visualization that MLLMs incorrectly rate highly. The authors claim it "does not accurately present the data," is "chaotic," and "impossible to decipher." However, this visualization is a standard and highly effective tool in the cryptocurrency trading domain, known as a "liquidation heatmap," which displays price action (candlesticks) alongside market liquidation levels (heatmap) (see, for example, https://www.coinglass.com/pro/futures/LiquidationHeatMap). To a domain expert (e.g., a trader), the chart is professional, intuitive, and information-dense. To a generalist audience, it is indeed confusing.
This example powerfully illustrates the central weakness: visualization quality is often domain-specific and audience-dependent. A "ground truth" established by generalist annotators may not be valid for specialized audiences. I do not criticize the authors for their lack of specific investment knowledge; rather, I use this to emphasize that finding a universal consensus on visualization quality is difficult, if not impossible. A deeper discussion of this challenge is needed.
3. Insufficient Analysis of Complex Visualizations: The results clearly show that model performance degrades significantly on more complex visualizations like multi-view charts and dashboards compared to single-view charts. While the paper notes this drop, it lacks a deep, quantitative analysis of why this occurs. A more detailed investigation into the specific failure modes (e.g., assessing inter-chart relationships, handling information density) would have been highly valuable, especially since evaluating such complex artifacts is a major research gap that this work laudably aims to address. This feels like a missed opportunity to generate deeper insights.

**Questions:**

- **Modeling Disagreement vs. Consensus:**
  The benchmark uses a single consolidated ground truth score. Given the known subjectivity in aesthetic and design judgments, could the authors share whether inter-annotator disagreement was analyzed before expert consolidation? It may be helpful to discuss whether such disagreement could carry meaningful signal rather than being treated purely as noise.

- **Potential for Distribution-Based Evaluation:**
  Related to the above, have the authors considered representing human ratings as a distribution instead of a single averaged score? Capturing variance or multimodal preferences might provide a more realistic target for model alignment with diverse human judgments.

- **Role of Domain Expertise:**
  Some visualizations (e.g., financial heatmaps with candlesticks) may be more interpretable to domain experts than to general audiences. Were annotators screened for domain familiarity, and do the authors foresee value in domain-specific subsets of the benchmark? It would be interesting to know how “ground truth” ratings might vary under domain-aware annotation.

- **Deeper Analysis of Multi-View Challenges:**
  The performance drop on multi-view and dashboard visualizations is an important observation. A brief breakdown of typical failure modes—e.g., difficulty assessing inter-chart relationships vs. cumulative single-chart errors—would provide useful insight into where models struggle most and help guide future work.

---

> ### Author Response · Authors · 2025-11-27
> **Response to Reviewer iwcV (Part 1/4)**
>
> > **W1. Fundamental Assumption of a Singular "Ground Truth"**
> >
> > **Q1: Modeling Disagreement vs. Consensus**
>
> Thank you for raising this important and insightful question. We understand your concern about the single "ground truth" assumption, which is indeed a core challenge in subjective evaluation tasks. We would like to clarify our annotation process and quality control strategy, and address concerns about how we handle disagreement.
>
> * **Analysis of inter-annotator disagreement:** We conducted systematic analysis before expert consolidation. Crowd-crowd MAE (each annotator vs. crowd average) ranged from 0.64-0.70, while crowd-expert MAE ranged from 0.54-0.59. Notably, crowd-expert MAE was consistently lower across all visualization types, indicating that ​**expert ratings effectively represent a balanced consensus among annotators**​. Detailed consistency analysis is provided in **Appendix A.4.1.**
> * **Targeted quality control rather than enforced consensus:** Expert review results show that **only 11% of samples underwent overall-level adjustments** (7% for malicious scoring patterns, 4% for statistical outliers), and 16% had sub-dimension adjustments, particularly for dimensions where standard deviation exceeded 1. This targeted approach preserves genuine diverse perspectives while ​**correcting clear errors ​**​(e.g., misunderstanding criteria or providing consistently extreme ratings). See **Appendix A.4.3** for details.
> * **Differentiated treatment of objective vs. subjective dimensions:** Analysis of**​ adjustment patterns reveals** that experts adjusted objective dimensions (semantic readability, data fidelity, insight discovery) more frequently (18-19%) than aesthetic dimensions (color harmony, visual composition, design style) (14-17%). This indicates that **while ensuring accuracy in objective dimensions, we respect and preserve diversity in subjective dimensions, avoiding over-standardization.** Detailed dimension-specific analysis is in Appendix **A.4.4.**
> * **Preserving original ratings for distribution-based research:** In our publicly released dataset, **we preserve the original ratings and confidence information from all annotators** (**603** annotators, each sample rated by 3), not just the final averaged or expert-adjusted scores. **Researchers can use this ​raw data for distribution-based modeling to explore rating ​variance, multimodal preferences, and other characteristics. ​**We acknowledge that disagreement may carry meaningful signals, and future work can explore using rating distributions rather than point estimates as evaluation targets.
>
> In summary, our approach establishes a standardized benchmark while preserving annotation diversity. We address the challenge of subjective evaluation through: (1) preserving original rating distributions for future research, (2) implementing differentiated quality control for objective vs. subjective dimensions, and (3) conducting detailed disagreement analysis. Complete annotation quality analysis has been added to **Appendix A.4.** We look forward to future work exploring distribution-based evaluation methods to more comprehensively capture the diversity of human preferences.

---

> ### Author Response · Authors · 2025-11-27
> **Response to Reviewer iwcV (Part 2/4)**
>
> > **Q2: Potential for Distribution-Based Evaluation**
>
> Thank you for this valuable suggestion. We fully agree that distribution-based evaluation methods have important research value and can more authentically capture the diversity of human judgments.
>
> * **Current support for distribution-based research:** In our dataset release, ​**we preserve the original ratings and confidence information from all three annotators per sample ​**​(**603** annotators total), along with anonymized annotator background information where privacy permits. Researchers can directly access rating distributions for each sample, including variance, standard deviation, and potential multimodal characteristics, to explore distribution-based modeling approaches.
> * **Value**​**​ of distribution-based evaluation:** We agree that using rating distributions rather than point estimates may better reflect diverse human judgments, particularly for subjective dimensions like color harmony and design style where different user groups may have multimodal preferences. Distribution-based evaluation methods could enable models to predict entire rating distributions, thereby better aligning with diverse population preferences.
> * **Current design rationale:** Our current use of average scores is motivated by: (1) maintaining comparability with existing visualization evaluation research and practice [1,2]; (2) practical integration into applications like visualization generation systems [3,4] where single scores are more convenient.
> * **Future directions:**​**We explicitly identify distribution-based evaluation as an important direction in**​​**Section 7 ​**​(Limitations and Future Work). Future research could explore: (1) training models to predict rating distributions rather than point estimates; (2) developing metrics to measure consistency between predicted and actual distributions; and (3) dynamically adjusting evaluation criteria across different user groups or scenarios. We believe such methods will provide richer characterization of visualization quality that better reflects the inherent subjectivity in human perception.
>
> [1] Chen, Nan, et al. "Viseval: A benchmark for data visualization in the era of large language models." *IEEE*​*​ TVCG* (2024).
>
> [2] Dibia, Victor. "LIDA: A Tool for Automatic Generation of Grammar-Agnostic Visualizations and Infographics using Large Language Models." Proceedings of ACL (2023).
>
> [3] Yang, Zhaorui, et al. "Multimodal DeepResearcher: Generating Text-Chart Interleaved Reports From Scratch with Agentic Framework." AAAI (2025).
>
> [4] Islam, Mohammed Saidul, et al. "DataNarrative: Automated Data-Driven Storytelling with Visualizations and Texts." *​Proceedings of the 2024 Conference on EMNLP (​*2024).

---

> ### Author Response · Authors · 2025-11-27
> **Response to Reviewer iwcV (Part 3/4)**
>
> > **W2. Domain-Specificity and Audience Dependence**
> >
> > The examples in Figure 2 and Figure 21 provide an interesting case illustrating the issue above. In certain professional contexts—such as cryptocurrency trading—this type of visualization, often called a "liquidation heatmap," is a well-established and effective analytical tool that integrates price action (candlesticks) with market liquidation levels (heatmap) (e.g., Coinglass). For domain experts, such charts are both familiar and informative, even though they may appear complex to general audiences. A deeper discussion of this challenge is needed.
> >
> > **Q3: Role of Domain Expertise**
> >
> > Some visualizations (e.g., financial heatmaps with candlesticks) may be more interpretable to domain experts than to general audiences. Were annotators screened for domain familiarity, and do the authors foresee value in domain-specific subsets of the benchmark? It would be interesting to know how "ground truth" ratings might vary under domain-aware annotation.
>
> Thank you for raising this insightful question about domain expertise. Your points about domain-specific interpretations and domain-aware annotation are highly valuable and provide important guidance for our future research directions.
>
> * **First, regarding the case in Figures 2 and 21:** We appreciate your point about domain-specific interpretations. While liquidation heatmaps are indeed effective tools for domain experts, this particular sample has clear design flaws. ​**It lacks x-axis labels and a clear color legend**​. Without these critical elements, the chart fails to meet basic readability standards that apply to all visualization types, regardless of viewer domain familiarity. We selected this sample for case analysis precisely because **MLLMs (e.g., GPT-5) made excessive inferences based on prior knowledge while ignoring the absence of critical labeling information.**
> * **Second, regarding our benchmark positioning and recruitment strategy:**​**Our benchmark aims to evaluate general visualization quality across multiple domains** (our dataset spans economy, technology, labor, and other fields as detailed in​**​ Appendix A.2.4**​). The target audience consists of general users who evaluate visualizations across diverse domains (e.g., data analysts, designers) rather than deep experts in a single specialized field. Based on this positioning, our recruitment strategy (detailed in ​**Appendix A.3.1**​) focused on selecting professionals who regularly encounter visualizations in their daily work, but **did not restrict recruitment to experts in specific domains.** This strategy yielded a diverse annotator pool: 41% work directly in data visualization or analysis, while 59% come from other professional backgrounds but possess evaluation capability.
> * **Third, regarding empirical analysis of background effects:** We collected and analyzed annotator background information. By professional background, visualization and data analysis professionals (41%) had MAE of 0.4245, while others (59%) had 0.4881. By education level, doctoral (7%), master's (23%), and bachelor's (70%) groups showed MAEs of 0.3944, 0.4762, and 0.4825 respectively. While background affects evaluation consistency, more importantly, annotators from all backgrounds showed consistent relative patterns across different visualization types, indicating our main conclusions are robust​**​ ​**​(see ​**Appendix A.4.2**​).
> * **Finally, regarding domain-specific subsets:** We fully agree this is an important and promising research direction. Your suggestion holds significant implications for the field. In our public dataset release, we will provide annotator background information (within privacy constraints) to support domain-aware evaluation research. Future work could explore evaluation differences between domain experts and general users, build domain-specific subsets, and study how to adjust evaluation criteria for different audiences.

---

> ### Author Response · Authors · 2025-11-27
> **Response to Reviewer iwcV (Part 4/4)**
>
> > **W3. Insufficient Analysis of Complex Visualizations. ​**The results clearly show that model performance degrades significantly on more complex visualizations like multi-view charts and dashboards compared to single-view charts. While the paper notes this drop, it lacks a deep, quantitative analysis of why this occurs. A more detailed investigation into the specific failure modes (e.g., assessing inter-chart relationships, handling information density) would have been highly valuable, especially since evaluating such complex artifacts is a major research gap that this work laudably aims to address. This feels like a missed opportunity to generate deeper insights.
> >
> > **Q4: Deeper Analysis of Multi-View Challenges. ​**The performance drop on multi-view and dashboard visualizations is an important observation. A brief breakdown of typical failure modes—e.g., difficulty assessing inter-chart relationships vs. cumulative single-chart errors—would provide useful insight into where models struggle most and help guide future work.
>
> Thank you for pointing out this important issue. We agree that a deeper analysis of failure modes in complex visualizations is needed. In response, we have systematically examined typical error cases in multi-view visualizations and dashboards, and added detailed error pattern analysis with quantitative evidence in **Appendix B.6.** Below we summarize the main findings:
>
> * **Performance Degradation and the Absence of Cross-View Understanding. ​**Through systematic analysis of **139 Multi-view** and **94 Dashboard** cases, we find that the core reason for performance degradation is the lack of holistic cross-view understanding. Models tend to evaluate each view independently, relying on local visual cues (e.g., grid alignment, color consistency) while failing to understand cross-view semantic connections.
> * **Aesthetic Dimensions: Dual Bias in Visual Composition and Color Harmony.** Models exhibit contradictory biases:
>   * (1) overestimating layout regularity, assigning high scores to standard grid layouts regardless of redundancy or lack of meaningful relationships (see ​**Figure 6**​, left). Quantitatively, InternVL3, Claude-3.5-Sonnet, and Qwen2.5-VL overestimated Visual Composition in \~90% of Multi-view cases by over 1.1 points on average;
>   * (2) over-penalizing practical design trade-offs, strictly enforcing rigid encoding rules. Gemini-2.5-Pro underestimated Color Harmony in 44 Multi-view and 29 Dashboard cases, with deviations nearing 1 point (see ​**Figure 7**​, right).
> * **Data Fidelity: Confusing Form with Content.** Models often assign high fidelity scores to charts with only axes and frames but no actual data marks (see ​**Figure 25**​). This is pervasive: Qwen2.5-VL and InternVL3 overestimated Data Fidelity in 95.0% and 96.4% of Multi-view cases; proprietary models (GPT-4o, Claude-3.5-Sonnet) inflated scores in 87.8% of cases by over 1.5 points on average. This suggests models prioritize structural conformity over accurate data conveyance.
> * **Dashboard Evaluation: Equating Completeness with Analytical Depth.** Models assign high Insight Discovery and Semantic Readability scores to dashboards with diverse chart types and unified styling, even when lacking logical coherence (see ​**Figure 19**​, top). Qwen2.5-VL inflated Semantic Readability in 90.4% of Dashboard cases by 1.275 points on average. Conversely, GPT-5 and Gemini-2.5-Pro misinterpret information-rich dashboards as "clutter", underestimating Semantic Readability in 33 and 67 cases by 0.78 and 1.15 points, respectively.
> * **Complexity-Specific Performance Patterns.** From single-view to multi-view visualizations, Visual Composition and Color Harmony drop most notably; from multi-view to dashboards, Insight Discovery and Semantic Readability show larger decreases. This indicates models fundamentally lack understanding of overall structure and cross-view relationships.
>
> Detailed error pattern classification, quantitative analysis, and representative cases have been added to ​**Appendix B.6**​. Additionally, **Appendices B.1-B.3** present cases across different score ranges, and **Appendix B.4** provides typical low-score examples.

---

> ### Comment · Reviewer_iwcV · 2025-11-28
>
> Thanks for the thorough and well-reasoned responses. The additional analyses and discussions adequately address my concerns, and the proposed approach to handling subjective, multi-annotator visualization ratings appears solid. The release of diverse raw ratings will also meaningfully benefit future research. I will keep my original score unchanged.

---

> > ### Author Response · Authors · 2025-12-03
> > **Thank you for your valuable feedback and recognition**
> >
> > Dear Reviewer iwcV,
> >
> > Thank you very much for your thoughtful review and constructive feedback throughout the review process. We greatly appreciate your recognition of our additional analyses and the value you see in releasing diverse raw ratings for future research.
> >
> > Your insightful questions about the fundamental assumptions of ground truth, domain-specific interpretations, and complex visualization analysis have significantly strengthened our work. The points you raised about distribution-based evaluation and domain-aware annotation have opened important directions for future research in this field.
> >
> > We are grateful for your time and expertise in reviewing our paper, and we look forward to contributing these resources to the community.
> >
> > Best regards,
> >
> > Authors

---

### Author Response · Authors · 2025-11-27
**Public Comment 1**

We sincerely thank all reviewers for their valuable feedback. We have substantially revised the paper, and all important revised content is marked in ​**blue**​. Below is a concise summary of the key updates:

1. (For Reviewer jQs1) **We**​**clarified the data fidelity assessment approach** in **Section 3.1.2** and ​**Appendix C.1**​, explaining our visual consistency-based evaluation method and its limitations in **Section 7.**
2. (For Reviewer jQs1) We conducted **comprehensive dataset analysis** characterizing VisJudge-Bench's coverage, strengths, and limitations in ​**Appendix A.2**​, including visualization type distribution (A.2.2), design style analysis (A.2.3), domain coverage (A.2.4), quality score distribution (A.2.5), and design flaw patterns (A.2.6), with representative case studies in ​**Appendix B.1-B.4**​.
3. (For Reviewers jQs1 and w8Z6) We added systematic **annotation quality analysis** in ​**Appendix A.4**​, covering: (1) overall annotation quality metrics across visualization types (A.4.1); (2) annotator background analysis by professional experience and education level (A.4.2); (3) quality control impact through expert review (A.4.3); and (4) dimension-specific adjustment patterns (A.4.4).
4. (For Reviewers jQs1 and w8Z6) To address generalization concerns, **we conducted out-of-distribution**​**generalization experiments** in **Section 5.2.5** and ​**Appendix E.2**​, integrating VisJudge into MatPlotAgent (average +6.07 improvement) and HAIChart (up to +5.3% Hit@3 improvement), demonstrating robust generalization capabilities.
5. (For Reviewer w8Z6) **We clarified VisJudge-Bench's positioning as primarily a benchmark.**​**Section 4** explores open-source model fine-tuning, and **Section 5** presents the core evaluation of 12 MLLMs. Dataset split rationale and statistics are added in **Appendix A.2.7.**
6. (For Reviewer dX6V) We **evaluated multiple open-source models with different parameter scales** (Qwen2.5-VL 3B/7B/32B/72B, InternVL3-8B, LLaVA-1.6-7B) and **conducted fine-tuning experiments** across architectures in **Section 5.2.1** and ​**Table 3**​, showing consistent improvements and validating the universality of our approach.
7. (For Reviewer dX6V) We **added representative scoring criteria examples** in **Section 3.1.2 ​**and acknowledged the limitation of static visualizations in ​**Section 7**​, identifying dynamic visualization support as important future work.
8. (For Reviewer iwcV) We ​**preserved original ratings from all annotators in the released dataset to support distribution-based research**​, and added analysis of quality control strategies differentiating objective vs. subjective dimensions in ​**Appendix A.4.4**​.
9. (For Reviewer iwcV) We **added detailed analysis of model failure modes on complex visualizations** in ​**Appendix B.6**​, identifying key problems in multi-view relationship understanding, information density processing, and domain knowledge handling.

We greatly appreciate the reviewers' constructive feedback and their recognition of our contribution.

---

### Author Response · Authors · 2025-12-03
**Rebuttal Summary (Part 7/7)**

**Scenario 2: Integration into Visualization Recommendation System.** We tested VisJudge as a reward model in HAIChart [2] on the VizML dataset, which contains real user-created visualizations with different distributions from the training set.

**Table 2: Visualization Recommendation Performance Comparison on VizML Dataset**

|Task|Metric|HAIChart (Baseline)|Qwen2.5-VL-7B as Reward Model|Qwen2.5-VL-72B as Reward Model|VisJudge as Reward Model|
|----------------|--------|---------------------|-------------------------------|--------------------------------|--------------------------|
|DataQueries|Hit@1|79.30%|80.30%|82.80%|**83.70%(+4.40%)**|
||Hit@3|91.90%|90.70%|92.10%|**93.20%(+1.30%)**|
|DesignChoices|Hit@1|48.70%|45.90%|46.50%|**48.80%(+0.10%)**|
||Hit@3|81.50%|80.70%|79.80%|**82.70%(+1.20%)**|
|Overall|Hit@1|36.90%|36.30%|39.10%|**39.40%(+2.50%)**|
||Hit@3|67.40%|69.10%|70.20%|**72.70%(+5.30%)**|

VisJudge **significantly improved recommendation accuracy:**​**Data Queries Hit@1 +4.40%, Overall Hit@1 +2.50%, Overall Hit@3 +5.30%.** Base models showed minimal to moderate improvements, while VisJudge consistently achieved larger gains, demonstrating effective quality evaluation capability in out-of-distribution scenarios.

These results collectively demonstrate that VisJudge learned transferable evaluation principles rather than simply overfitting the training data.

**(6) In-Depth Analysis of Failure Modes in Complex Visualizations**

We conducted systematic analysis of **139 multiple visualization and 94 dashboard cases** in ​**Appendix B.6**​. The core finding is that models lack holistic cross-view understanding, tending to evaluate each view independently while ignoring cross-view semantic connections, leading to over-focus on surface forms while neglecting actual functionality. Typical failure modes include:

* **Dual Biases in Aesthetic Dimensions.** InternVL3, Claude-3.5-Sonnet, and others over-reward neat grid layouts (overestimating visual composition by over 1.1 points on average) in approximately 90% of multiple visualization cases (see ​**Figure 6**​, left), even when data dimensions are inconsistent or color schemes conflict; Gemini-2.5-Pro is overly strict about color encoding consistency, underestimating color harmony by nearly 1 point in multiple visualizations and dashboards (see ​**Figure 7**​, right).
* **Data Fidelity: Emphasizing Form Over Content.** Models assign high fidelity scores to charts with only axis frames but no actual data (see ​**Figure 25**​). Qwen2.5-VL and InternVL3 overestimated fidelity in over 95% of multiple visualization cases, and GPT-4o and Claude-3.5-Sonnet overestimated by over 1.5 points on average in 87.8% of cases.
* **Dashboard Evaluation: Confusing Completeness with Effectiveness.** Qwen2.5-VL assigns high scores to dashboards with diverse charts and unified styling (overestimating semantic readability by 1.275 points on average), even when lacking logical coherence (see ​**Figure 19**​, top); GPT-5 and Gemini-2.5-Pro misjudge information-rich dashboards as "cluttered" and over-penalize them (underestimating by 0.78 and 1.15 points respectively).
* **Cross-complexity analysis further validates these findings:** from single to multiple visualizations, visual composition and color harmony evaluation quality decline most significantly; from multiple visualizations to dashboards, insight discovery and semantic readability evaluation quality decline more, indicating that models fundamentally lack understanding of overall structure and cross-view relationships.

Detailed analysis is in ​**Appendix B.6**​, with **Appendices B.1-B.4** providing different score ranges and typical low-score cases.

We greatly appreciate your time and attention in reviewing our submission. In response to the reviewers' concerns, we have conducted additional supplementary experiments and revised the manuscript accordingly, with all major additions marked in ​**blue**​.

Thank you for your careful consideration.

[1] Yang Z, Zhou Z, Wang S, et al. MatPlotAgent: Method and Evaluation for LLM-Based Agentic Scientific Data Visualization. ACL (Findings), 2024.

[2] Xie Y, Luo Y, Li G, et al. HAIChart: Human and AI Paired Visualization System. Proceedings of the VLDB Endowment, 2024, 17(11): 3178-3191.

---

### Author Response · Authors · 2025-12-03
**Rebuttal Summary (Part 6/7)**

**(5) Cross-Architecture and Real-World Application Scenario Generalization ​**​**Validation**

We validated generalization capability from two perspectives: cross-architecture and scale model evaluation, and out-of-distribution performance testing in real-world application scenarios.

**Cross-Architecture and Scale Evaluation** (​**Section 5.2.1, Table 3**​): We evaluated multiple open-source models covering a complete gradient from 3B to 72B.

**Table 1: Performance Comparison across Different Model Scales (MAE and Correlation metrics shown; for complete results see Table 3 in the paper)**

|Metric|Model|Overall|Fidelity|Readability|Insight|DesignStyle|Composition|Color|
|---|---|---|---|---|---|---|---|---|
|**MAE(↓)**|Qwen2.5-VL-3B-Instruct|0.821|1.085|1.258|1.088|0.723|**0.727**|0.808|
||Qwen2.5-VL-7B-Instruct|0.847|1.171|1.296|0.858|0.756|0.812|0.772|
||Qwen2.5-VL-32B-Instruct|0.703|**0.909**|0.987|**0.801**|0.678|0.761|0.719|
||Qwen2.5-VL-72B-Instruct|**0.702**|0.999|**0.926**|0.811|**0.663**|0.735|**0.717**|
|**Corr.(↑)**|Qwen2.5-VL-3B-Instruct|0.272|0.199|0.222|0.275|0.338|0.130|0.155|
||Qwen2.5-VL-7B-Instruct|0.341|0.341|0.352|0.281|0.357|0.149|0.155|
||Qwen2.5-VL-32B-Instruct|0.435|**0.348**|0.468|0.408|**0.449**|**0.200**|**0.268**|
||Qwen2.5-VL-72B-Instruct|**0.440**|0.331|**0.479**|**0.416**|0.435|0.165|0.251|



**Table 2: Fine-tuning Performance Comparison across Architectures (MAE and Correlation metrics shown; for complete results see Table 3 in the paper)**

|Metric|Type|Model|Overall|Fidelity|Readability|Insight|DesignStyle|Composition|Color|
|---|---|---|---|---|---|---|---|---|---|
|**MAE(↓)**|**Open-source**|InternVL3-8B|0.793|1.234|1.193|0.870|0.679|0.759|0.753|
|||Llava-v1.6-mistral-7B|0.724|0.929|1.124|0.939|0.801|0.814|0.818|
|||Qwen2.5-VL-3B-Instruct|0.821|1.085|1.258|1.088|0.723|0.727|0.808|
|||Qwen2.5-VL-7B-Instruct|0.847|1.171|1.296|0.858|0.756|0.812|0.772|
||**VisJudge(Ours)**|InternVL3-8B|0.541|0.797|0.769|0.691|0.608|0.645|0.616|
|||Llava-v1.6-mistral-7B|0.496|0.767|0.695|0.775|0.643|0.574|**0.598**|
|||Qwen2.5-VL-3B-Instruct|0.491|0.705|0.721|0.696|0.625|0.571|0.616|
|||Qwen2.5-VL-7B-Instruct|**0.421**|**0.661**|**0.648**|**0.677**|**0.580**|**0.545**|0.604|
|**Corr.(↑)**|**Open-source**|InternVL3-8B|0.409|0.323|0.407|0.344|0.419|0.216|0.170|
|||Llava-v1.6-mistral-7B|0.180|0.201|0.137|0.160|0.188|0.064|0.076|
|||Qwen2.5-VL-3B-Instruct|0.272|0.199|0.222|0.275|0.338|0.130|0.155|
|||Qwen2.5-VL-7B-Instruct|0.341|0.341|0.352|0.281|0.357|0.149|0.155|
||**VisJudge(Ours)**|InternVL3-8B|0.660|0.533|0.594|0.545|0.499|0.391|**0.420**|
|||Llava-v1.6-mistral-7B|0.605|0.226|0.536|0.443|0.432|0.406|0.403|
|||Qwen2.5-VL-3B-Instruct|0.648|0.533|0.581|**0.579**|0.504|0.490|0.402|
|||Qwen2.5-VL-7B-Instruct|**0.687**|**0.574**|**0.628**|0.576|**0.568**|**0.513**|0.385|



Key findings: (1) The largest open-source model (Qwen2.5-VL-72B) performed comparably to GPT-5 (correlation 0.440 vs 0.428); (2) **All fine-tuned models improved significantly, ​**with Spearman correlation increasing by over 0.3 on average; (3) The ​**fine-tuned 3B model (0.648) surpassed all unfine-tuned open-source models**​, demonstrating that targeted fine-tuning can effectively compensate for scale limitations.

**Out-of-Distribution Generalization Experiments** (​**Section 5.2.5, Appendix E.2**​): We tested VisJudge's generalization in two real-world scenarios with different distributions from the training set.

**Scenario 1: Integration into Visualization Generation System.** We integrated VisJudge as a feedback mechanism into MatPlotAgent[1] and evaluated it on MatPlotBench.

**Table 1: Visualization Generation Quality Comparison**

|Model|Direct Decoding|Qwen2.5-VL-7B as Feedback|Qwen2.5-VL-72B as Feedback|VisJudge as Feedback|
|-------------------------|-----------------|---------------------------|----------------------------|----------------------|
|GPT-5|67.15|66.49|69.8|**72.07(+4.92)**|
|GPT-4o|62.71|58.49|63.53|**71.62(+8.91)**|
|Gemini-2.5-Pro|68.1|65.04|66.12|**72.20(+4.10)**|
|Claude-4-Sonnet|64.5|67.25|65.2|**72.23(+7.73)**|
|Claude-3.5-Sonnet|63.89|59.23|63.14|**69.04(+5.15)**|
|Qwen2.5-VL-72B-Instruct|61.18|58.7|60.3|**67.32(+6.14)**|
|Qwen2.5-VL-7B-Instruct|49.76|45.38|44.94|**55.29(+5.53)**|


VisJudge achieved an **average ​**​**quality improvement**​​**​ of +6.07 points across 7 generation models**​. Notably, **unfine-tuned base models as feedback providers often degraded performance (7B: -2.39, 72B: -0.61)** because they exhibit significant gaps with human experts and lack domain-specific knowledge, leading to misleading feedback. In contrast, VisJudge consistently improved all generators, with even state-of-the-art models like **GPT-5 and Gemini-2.5-Pro achieving improvements of +4.92 and +4.10** respectively, demonstrating that domain-specific fine-tuning is crucial for providing effective feedback.

---

### Author Response · Authors · 2025-12-03
**Rebuttal Summary (Part 5/7)**

**(4) Clarification of Benchmark Positioning and Research Contributions**

We clarified the positioning of VisJudge-Bench, dataset split strategy, and the systematic contributions of this research beyond dataset curation and model fine-tuning.

* **Benchmark Positioning:** The ability of MLLMs to evaluate visualization quality and aesthetics is crucial for their effectiveness as reward models in visualization generation and recommendation systems, yet there lacks a systematic benchmark to evaluate their performance on this task. To address this, we constructed ​**VisJudge-Bench as a systematic evaluation benchmark for MLLM visualization quality and aesthetic assessment capabilities**​. Building on this high-quality benchmark, we further explore whether open-source models can effectively learn visualization evaluation tasks through fine-tuning and match or even surpass the evaluation capabilities of closed-source models.
* **Experimental Organization: Section 4** explores the feasibility of fine-tuning open-source models based on VisJudge-Bench, training VisJudge models across different base models and parameter scales; **Section 5** conducts unified evaluation and comparative analysis of **all model types** (closed-source models, open-source baseline models, and fine-tuned VisJudge) on the same representative test set, systematically revealing different models' capability boundaries, systematic biases, and typical failure modes.
* **Dataset Split and ​**​**Validation**​**​ Strategy:** Our dataset systematically covers ​**32 visualization types**​, multiple design styles, and major application domains, with all **3,090 samples** undergoing rigorous quality control by **603 annotators** and an expert team (​**Appendix A.2 and A.4**​). We employed **stratified random sampling** for the dataset split, with the test set containing **648 samples** that fully preserve the distributional characteristics of the original dataset, ensuring sufficient representativeness (detailed in ​**Appendix A.2.7**​). To address concerns about potential overfitting, we validated generalization through cross-architecture evaluation and out-of-distribution experiments in real-world applications (​**detailed in the next section**​).
* **Systematic Research Contributions:** The core contributions extend beyond dataset construction and model fine-tuning: (1) Constructing the ​**first benchmark to systematically evaluate MLLMs' visualization quality and aesthetics assessment capabilities based on the "Fidelity, Expressiveness, and Aesthetics" framework**​, filling the gap where existing research focuses only on isolated single aspects; (2) **Systematically revealing capability gaps across 12 MLLMs** (even ​**GPT-5 achieves only 0.428 correlation with experts**​), hierarchical capability structures, and systematic biases; (3) **In-depth analysis of typical failure modes** (lack of multi-view understanding, confusing form with content), providing clear directions for future improvements; (4) **Validating the effectiveness of domain-specific fine-tuning** (correlation improved to ​**0.687**​). These collectively constitute a complete research framework, rather than merely incremental work.

---

### Author Response · Authors · 2025-12-03
**Rebuttal Summary (Part 4/7)**

**(3) Systematic Analysis of Annotation Quality and ​**​**Consistency**

We added comprehensive annotation quality analysis (​**Appendix A.4**​), systematically describing the annotation process, quality assessment, and the complete mechanism for handling annotation disagreements while preserving rating diversity.

* **Overall Annotation Quality Assessment** (​**Appendix A.4.1**​): We conducted systematic inter-annotator consistency analysis before expert review. Crowd-crowd MAE (each annotator vs. crowd average) ranged from **0.64 to 0.70** across different visualization types, while crowd-expert MAE (each annotator vs. expert) ranged from ​**0.54 to 0.59**​. The key finding is that ​**crowd-expert MAE was consistently lower than crowd-crowd MAE**​, indicating that the deviation between each annotator and expert standards is smaller than the deviation among annotators themselves, demonstrating that expert ratings have good representativeness and can effectively balance different annotators' perspectives. **These values are comparable to acceptable levels in prior crowdsourced visualization studies** [1, 2].
* **Impact of Annotator Background on ​**​**Consistency** (​**Appendix A.4.2**​): We analyzed the background information of **603 annotators** and its impact on evaluation quality. By professional background, visualization and data analysis professionals (41%) had within-group MAE of ​**0.4245**​, while other backgrounds (59%) had ​**0.4881**​. By education level, doctoral (7%), master's (23%), and bachelor's (70%) groups showed MAEs of **0.3944, 0.4762, and 0.4825** respectively. Although background affects evaluation consistency, more importantly, **all background subgroups exhibited consistent relative error patterns** across different visualization types, indicating that our main conclusions are robust across different annotator backgrounds.
* **Quality Control Mechanisms for Disagreement Handling** (​**Appendix A.4.3-A.4.4**​): Experts reviewed all **3,090 samples** individually. **Only 11% of samples required adjustments** at the overall evaluation level (7% for malicious scoring patterns, 4% for statistical outliers), and ​**16% required sub-dimension adjustments**​, particularly for dimensions where standard deviation exceeded 1. This targeted approach **preserved genuine diverse perspectives while correcting clear errors** (e.g., misunderstanding criteria or providing consistently extreme ratings), rather than eliminating all inconsistencies.
* Further analysis shows that **objective dimensions** (such as semantic readability 19.20%, data fidelity 18.04%) **had higher adjustment rates than aesthetic dimensions** (such as visual composition 14.73%, color harmony 14.20%), validating our strategy: strict control for objective standards to ensure accuracy, while respecting and preserving diversity in subjective dimensions to avoid over-standardization.

In the publicly released dataset, we **preserve the original ratings and confidence information from all annotators (603 annotators, 3 ratings per sample)** to support future distribution-based research.

[1] Heer, J., & Bostock, M. "Crowdsourcing graphical perception: using mechanical turk to assess visualization design." CHI 2010.

[2] Kazai, G., Kamps, J., & Milic-Frayling, N. "An analysis of human factors and label accuracy in crowdsourcing relevance judgments." Information Retrieval 16.2 (2013): 138-178.

---

### Author Response · Authors · 2025-12-03
**Rebuttal Summary (Part 3/7)**

**(2) Systematic Dataset Analysis**

We conducted systematic analysis of VisJudge-Bench in ​**Appendix A.2**​, with key findings as follows:

* **Visualization Type Coverage and Real-World ​**​**Consistency**​**​ (Appendix A.2.2):** The distribution of **32 chart subtypes** across three levels is **highly consistent with real-world usage patterns** [1, 2, 3]. Single visualizations exhibit a long-tail pattern (basic charts 46%, mid-frequency analytical charts 38%, specialized types 16%), multiple visualizations are dominated by comparison views (65%), and dashboards focus on analytical type (73%).
* **Balanced Coverage of Design Styles (Appendix A.2.3): ​**Design style scores show a mean of 2.97 with uniform distribution (Appendix Figure 11). Manual inspection identified three style categories: Professional/Standard Style (53.3%), Creative/Infographic Style (28.4%), and Minimalist Style (18.3%), demonstrating diverse design approaches in practice.
* **Domain Distribution Reflecting Actual Demand** (​**Appendix A.2.4**​): **93.8% of samples** were annotated using IPTC standards. **Since the dataset is sourced from real-world web resources, its domain distribution naturally reflects actual visualization demand.** Economy, business and finance (26.3%), science and technology (20.3%), and labor (15.7%) collectively account for 62.3%, consistent with high practical demand in these fields. The dataset also covers other domains including society, health, politics, and environment.
* ​**Quality Distribution and Design Flaw Analysis (Appendix A.2.5-A.2.6, Figure 9)**​: Quality scores span the full score range. Low-scoring samples (overall score < 2) ​**decrease with complexity**​: single visualizations 12.5%, multiple visualizations 9.2%, dashboards 1.3%, indicating that dashboards from mature business scenarios typically maintain higher quality. The analysis also identified typical design flaws for each type: single visualizations perform weakest in design style, the core challenge for multiple visualizations lies in semantic integration, and failures in dashboards concentrate on functional deficiencies. Representative cases are shown in ​**Appendices B.1-B.4**​.
* **Dataset Strengths and Limitations:** **Strengths include** ​comprehensive coverage of commonly used visualization types, balanced quality distribution, and a six-dimensional evaluation framework. **Limitations:** The current version focuses on static visualizations; dynamic and interactive visualizations are future expansion directions (​**Section 7**​).

[1] Borkin et al., "What makes a visualization memorable?" IEEE TVCG (2013).

[2] Chen et al., "Composition and configuration patterns in multiple-view visualizations." IEEE TVCG (2020).

[3] Sarikaya et al., "What do we talk about when we talk about dashboards?" IEEE TVCG (2018).

---

### Author Response · Authors · 2025-12-03
**Rebuttal Summary (Part 2/7)**

2. **Main Supplementary Experiments and Paper Revisions**

The main concerns raised by the four reviewers can be summarized into the following ​**6 aspects**​:

(1) Data fidelity assessment methods need further clarification, particularly the evaluation basis when source data is unavailable (Reviewer jQs1);

(2) The benchmark dataset needs more systematic analysis (Reviewer jQs1);

(3) Annotation quality and consistency need systematic analysis, including inter-annotator differences and disagreement handling mechanisms (Reviewers jQs1, w8Z6, iwcV, dX6V);

(4) Benchmark positioning, experimental design logic, and research contributions require further clarification (Reviewers w8Z6, jQs1);

(5) Need to expand evaluation across different model architectures and scales, and add generalization validation in real-world application scenarios (Reviewers jQs1, dX6V, w8Z6);

(6) Need to deepen the analysis of failure modes in complex visualizations, providing quantitative evidence and representative cases (Reviewer iwcV).

During the discussion phase, we conducted the following supplementary experiments and detailed explanations to address these concerns.

**(1) Clarification of Data Fidelity Assessment Methodology**

We clarified the visual-level data fidelity assessment method in **Section 3.1.2** and ​**Appendix C.1**​. Since the charts in VisJudge-Bench are primarily collected from public web pages, the underlying source data is typically unavailable, and in real-world scenarios, users generally evaluate charts based solely on the final visualization. Therefore, we adopted a visual-level assessment approach that focuses on identifying visually detectable distortions in chart presentations.

This approach is based on ​**Tufte's foundational work on avoiding "graphical lies" ​**​[1] and similar methods in recent visualization evaluation research [2, 3], representing a ​**widely accepted practice in the community when source data is unavailable**​. Specific evaluation criteria include visual encoding consistency, axis and scale integrity, and common misleading patterns (truncated baselines, disproportionate encodings, etc.), with detailed criteria in ​**Appendix C.1**​.

We clarified the scope and limitations of this method in ​**Section 7**​: it can effectively identify visually apparent misleading signals, but cannot verify the correctness of the source data itself or assess complex statistical manipulations requiring domain knowledge.

[1] Tufte, Edward R., and Peter R. Graves-Morris. ​The visual display of quantitative information​. Vol. 2. No. 9. Cheshire, CT: Graphics press, 1983.

[2] Lan, X., & Liu, Y. "I Came Across a Junk: Understanding Design Flaws of Data Visualization from the Public's Perspective." IEEE TVCG 2024.

[3] Li, Z., et al. "Chartgalaxy: A dataset for infographic chart understanding and generation." arXiv:2505.18668, 2025.

---

### Author Response · Authors · 2025-12-03
**Rebuttal Summary (Part 1/7)**

Dear Area Chair,

Thank you for reviewing our submission VisJudge-Bench. During the discussion phase, we provided detailed responses to the concerns raised by all four reviewers, conducted additional experiments and extensive analysis, and revised the paper accordingly. ​**This summary aims to provide you with a comprehensive overview of the paper's current status**​, including:

(1) reviewer feedback and rating changes based on our detailed responses and additional experimental results during the discussion phase;

(2) the main supplementary experiments, new analyses, and paper revisions we completed to address the core concerns raised by the reviewers.

1. **Reviewer Feedback and Rating Changes**

The rating changes and current status of the four reviewers are as follows:


|Reviewer|Original Rating|After Discussion|
|---|---|---|
|iwcV|8|8 (Unchanged)|
|w8Z6|4|4 (No Reply)|
|dX6V|6|6 (No Reply)|
|jQs1|2|6 (Nov 27, 11:35)|



**Reviewer iwcV (Rating 8, Unchanged):** The reviewer ​**confirmed on November 28, 8:46 that our responses adequately addressed their concerns**​. Main concerns included annotation consistency and disagreement handling, distribution-based evaluation methods, domain expertise and audience dependence, and failure mode analysis of complex visualizations. ​**Our responses: ​**​(1) Systematically analyzed inter-annotator consistency (Appendix A.4.1) and quality control mechanisms (Appendix A.4.3-A.4.4), and preserved all annotators' original ratings and confidence information in the dataset to support distribution-based research; (2) Discussed the value of distribution-based evaluation methods and identified it as a future research direction (Section 7); (3) Analyzed the impact of annotator backgrounds (Appendix A.4.2) and discussed domain coverage (Appendix A.2.4); (4) Systematically analyzed failure modes in complex visualizations (Appendix B.6, B.1-B.4).

**Reviewer w8Z6 (Rating 4, ​**​**No Reply**​**​ During Discussion):** Main concerns focused on annotation method diversity, experimental design clarity, and generalization capability. **Our responses:** (1) Explained the annotation method design: customized scoring criteria for each chart, with annotators able to report confidence levels, flag uncertain cases, and provide post-task open-ended feedback (Section 3.1.2, Appendix A.4.3-A.4.4); (2) Clarified the benchmark positioning and experimental organization logic, and supplemented dataset split strategy details (Appendix A.2.7); (3) Demonstrated **strong out-of-distribution generalization** through integration into MatPlotAgent visualization generation system **(average +6.07 improvement)** and HAIChart visualization recommendation system **(+5.3% Hit@3)** (Section 5.2.5, Appendix E.2).

**Reviewer dX6V (Rating 6, ​**​**No Reply**​**​ During Discussion):** Main concerns focused on annotator quantity and quality, cross-architecture generalization validation, scoring criteria presentation, and dynamic visualization support. **Our responses:** (1) Clarified the annotation process design: **603 annotators** participated in total, with each sample independently rated by 3 randomly selected annotators; expert review employed targeted quality control to correct clear errors while preserving rating diversity (Appendix A.4); (2) Evaluated **multiple open-source models across different parameter scales** and conducted cross-architecture fine-tuning experiments (Section 5.2.1, Table 3); (3) Added representative scoring criteria examples in the main text (Section 3.1.2); (4) Acknowledged the limitation of static visualizations and identified dynamic visualization support as important future work (Section 7).

**Reviewer jQs1 (Rating Increased from 2 to 6, Updated Nov 27, 11:35):** Main concerns included unclear data fidelity assessment methods, insufficient dataset benchmarking analysis, missing annotation quality analysis, scope of research contributions, and insufficient generalization validation. Our responses: (1) Clarified the visual-level data fidelity assessment method and its theoretical foundation (Section 3.1.2, Appendix C.1); (2) Supplemented **comprehensive dataset characterization analysis** covering visualization types, design styles, domain coverage, and quality distributions (Appendix A.2); (3) Provided **systematic annotation quality analysis** including inter-annotator consistency (Appendix A.4.1), annotator background effects (Appendix A.4.2), and quality control mechanisms (Appendix A.4.3-A.4.4); (4) Clarified the research positioning and systematic contributions: establishing a systematic evaluation benchmark, ​**revealing capability gaps and failure modes through evaluation of 12 MLLMs**​, and validating the effectiveness of domain-specific fine-tuning; (5) Conducted out-of-distribution generalization experiments (Section 5.2.5, Appendix E.2). ​**The reviewer confirmed that their concerns were adequately addressed and increased their rating**​.

---

### Meta-Review · Area_Chair_yaPJ · 2026-01-02

**Summary:**

The paper addresses a timely and important problem and proposes a better benchmark for evaluating MLLMs on visualization quality. While in the initial submission, there are several concerning points raised. Major points include the strong assumption that a single, objective ground truth for visualization quality, potentially underestimates the inherent subjectivity, domain dependence, and audience-specific nature of judgments, particularly for aesthetics and expressiveness. The experimental design is also a bit confusing and not fully convincing: using a small benchmark dataset for both training and testing raises overfitting and generalization concerns and blurs the paper’s positioning as a benchmark versus a dataset or model paper. Additionally, the work lacks deeper analysis of complex and domain-specific visualizations, dataset coverage, fidelity assessment without access to source data, and real-world applicability, which limits the strength of the conclusions drawn from the benchmark.

**Reviewer Concerns:**

Two reviewers clearly stated that their most concners got clarifed by the review (Reviewer iwcV's and Reviewer jQs1's). Particularly, jQs1 raised the score. While among the two no-reply responses, Reviewer dX6V major questions lie on the benchmark curation details, such as annotator quality, cross-architecture generalization validation. With the provided rebuttal, i believe the reviewer would satisfy with the response, and keep the score as the positive. While Reviewer w8Z6 major criticize on the potential model overfitting to the small scale curated data, and the generliability of the finentued model. In the rebuttal, the authors provided additional experiments in two additional scenarios including MatPlotAgent visualization generation system and HAIChart visualiation recommendation system. It to some extend ease the concern. However, in the added results, the authors only compare with the baseline no-finetuned model as feedback, instead of other stronger baselines such as GPT5. Also, i hypothsize this reviewer may want to see the results on general VLM benchmarks, with the purpose to see if the finetuned model actually degrade a lot after finetuning on this small scale datasets.

**Reviewer Scores:**

Two reviewers clearly stated that their most concerns got clarified by the rebuttal (Reviewer iwcV’s and Reviewer jQs1’s). Particularly, jQs1 raised the score. Among the two no-reply responses, Reviewer dX6V’s major questions lie in the benchmark curation details, such as annotator quality and cross-architecture generalization validation. With the provided rebuttal, I believe the reviewer would be satisfied with the response and keep the score positive. Meanwhile, Reviewer w8Z6 mainly criticized the potential model overfitting to the small-scale curated data and the generalizability of the fine-tuned model. In the rebuttal, the authors provided additional experiments in two scenarios, including the MatPlotAgent visualization generation system and the HAIChart visualization recommendation system, which to some extent ease the concern. However, in the added results, the authors only compare against the baseline non-fine-tuned model, instead of stronger baselines such as GPT-5. Also, I hypothesize that this reviewer may want to see results on general VLM benchmarks, to assess whether the fine-tuned model degrades significantly after fine-tuning on this small-scale dataset.

---

### Decision · Program_Chairs · 2026-01-26

Accept (Poster)